

# Investigating processes influencing simulation of local Arctic wintertime anthropogenic pollution in Fairbanks, Alaska during ALPACA-2022

Natalie Brett[1,2], Kathy S. Law[1], Steve R. Arnold[2], Javier G. Fochesatto[3], Jean-Christophe Raut[1], Tatsuo Onishi[1], Robert Gilliam[4], Kathleen Fahey[4], Deanna Huff[5], George Pouliot[4], Brice Barret[6], Elsa Dieudonné[7], Roman Pohorsky[8], Julia Schmale[8], Andrea Baccarini[8,9], Slimane Bekki[1], Gianluca Pappaccogli[10], Federico Scoto[10], Stefano Decesari[11], Antonio Donateo[10], Meeta Cesler-Maloney[12], William Simpson[12], Patrice Medina[6], Barbara D'Anna[13], Brice Temime-Roussel[13], Joel Savarino[14], Sarah Albertin[14,1], Jingqiu Mao[12], Becky Alexander[15], Allison Moon[15], Peter F. DeCarlo[16], Vanessa Selimovic[17], Robert Yokelson[18], and Ellis S. Robinson[16]

[1]Sorbonne Université, UVSQ, CNRS, LATMOS, 75252 Paris, France
[2]Institute for Climate and Atmospheric Science, School of Earth & Environment, University of Leeds, UK
[3]Department of Atmospheric Sciences, College of Natural Science and Mathematics, University of Alaska Fairanks, Fairbanks, AK 99775, United States
[4]Center for Environmental Measurement and Modeling, Office of Research and Development, US EPA, Research Triangle Park NC 27709, United States
[5]Alaska Department of Environmental Conservation, P.O. Box 111800, Juneau, 99811-1800, United States
[6]Université Paul Sabatier, Université de Toulouse - CNRS, Toulouse, France
[7]Laboratoire de Physico-Chimie de l'Atmosphère (LPCA), Université du Littoral Côte d'Opale (ULCO): 59140 Dunkerque, Hauts-de-France, FR
[8]Extreme Environments Research Laboratory, École Polytechnique Fédérale de Lausanne, EPFL Valais Wallis, Sion, Switzerland
[9]Laboratory of Atmospheric Processes and their Impacts, Ecole Polytechnique Fédérale de Lausanne, Switzerland
[10]Institute of Atmospheric Sciences and Climate (ISAC) of the National Research Council of Italy (CNR), Lecce 73100, Italy
[11]Institute of Atmospheric Sciences and Climate (ISAC) of the National Research Council of Italy (CNR), Bologna 40121, Italy
[12]Geophysical Institute and Department of Chemistry and Biochemistry, University of Alaska Fairbanks, Fairbanks, AK 99775, United States
[13]Aix Marseille Univ, CNRS, LCE, Marseille, France
[14]IGE, Univ. Grenoble Alpes, CNRS, INRAE, IRD, Grenoble INP, 38000 Grenoble, France
[15]Department of Atmospheric Sciences, University of Washington, Seattle, Washington 98195, United States
[16]Department of Environmental Health and Engineering, Johns Hopkins University, Baltimore, MD 21218, United States
[17]Department of Chemistry, University of Michigan Ann Arbor, Michigan 48109, US
[18]Department of Chemistry and Biochemistry, University of Montana, Missoula, Montana 59812, United States

**Correspondence:** Natalie Brett (natalie.brett@latmos.ipsl.fr) and Steve R. Arnold (S.Arnold@leeds.ac.uk)

**Abstract.** Lagrangian tracer simulations are deployed to investigate processes influencing vertical and horizontal dispersion of anthropogenic pollution in Fairbanks, Alaska, during the ALPACA-2022 field campaign. Simulations of carbon monoxide (CO), sulphur dioxide ($SO_2$) and nitrogen oxides ($NO_x$), including surface and elevated emissions, are highest at the surface under very cold stable conditions. Regional enhancements, simulated up to 200 m, are due to elevated power plant emissions



above 50 m, with south-westerly pollutant outflow. Fairbanks regional pollution may be contributing to wintertime Arctic haze. Inclusion of a novel power plant plume rise treatment that considers the presence of surface and elevated temperature inversion layers leads to improved agreement with observed CO and $NO_x$ plumes with discrepancies attributed to, for example, displacement of plumes by modelled winds. At the surface, model results show that observed CO variability is largely driven by meteorology and to a lesser extent by emissions, although simulated tracers are sensitive to modelled vertical dis-

persion. Modelled underestimation of surface $NO_x$ during very cold polluted conditions is considerably improved following the inclusion of substantial increases in diesel vehicle $NO_x$ emissions at cold temperatures (e.g. a factor of 6 at -30 °C). In contrast, overestimation of surface $SO_2$ is attributed to issues related to the vertical dispersion of elevated space heating emissions during strongly and weakly stable conditions. This study highlights the need for improvements to local wintertime Arctic anthropogenic surface and elevated emissions and improved simulation of Arctic stable boundary layers.

## 1   Introduction

Arctic haze, with enhanced aerosols and trace gases, is formed in the lower troposphere during winter and early springtime (Shaw, 1975), and is predominantly caused by low-level transport of pollution, driven by low pressure weather systems, originating from Northern Eurasia (Stohl, 2006; Bourgeois and Bey, 2011; Law et al., 2014). Declining trends in aerosol mass

concentrations of Arctic haze constituents, including sulfate aerosols and black carbon (BC), since the early 1990s, across many stations including Barrow, Alaska and Alert, Canada, correlate with reductions in anthropogenic emissions in northern mid-latitudes (Bodhaine and Dutton, 1993; Sharma et al., 2019; Schmale et al., 2022). However, increases in Arctic urbanisation and industrial activities, that are anticipated to continue rising due to the warming climate and socio-economic development, also contribute to Arctic haze and to local air quality, highlighting their importance for Arctic urban areas and local

communities (Andrew, 2014; Schmale et al., 2018). Local sources of air pollution in the Arctic include gas flaring, mining, shipping, domestic heating and power generation (Stohl et al., 2013; Schmale et al., 2018). In the wintertime, energy demands are considerable due to the harsh cold climates endured by residents, however, significant challenges arise implementing sustainable transportation and energy infrastructure (de Witt et al., 2021; Kolker et al., 2022) due to remote and sparsely populated communities and cities (Schmale et al., 2018). This has led to substantial investment in fossil fuel power generation, e.g. in

Alaska and Canada (Mortensen et al., 2017; Kolker et al., 2022). The release of harmful air pollutants from surface emission sources and elevated power plant stacks contributes to poor air quality and adverse effects on human health during Arctic winter (Rosenthal and Watson, 2011; Schmale et al., 2018). These effects are exacerbated by snow covered surfaces and low solar radiation at this time of year, which create favourable conditions for reduced atmospheric boundary layer (ABL) heights and the formation of surface-based temperature inversions (SBIs). Such strong stratification near the surface inhibits pollution

dispersion leading to a build-up of pollutants at breathing level (Bradley et al., 1992; Shaw, 1995). However, the contribution



of local Arctic emissions to air quality, and its possible contribution to background Arctic haze remains poorly quantified. This is due to uncertainties in emissions and in the ability of models to capture wintertime processes such as aerosol formation, deposition and the complex boundary layer meteorology (Emerson et al., 2020; AMAP, 2021; Donateo et al., 2023).

Fairbanks, a sub-Arctic city in Interior of Alaska (64.8 N, -147.7 W), is an example of a polluted urban area. Despite the relatively low population ($\sim$ 33 000 inhabitants in Fairbanks and 100 000 in Fairbanks North Star Borough (FNSB) aglomeration), the 24h-average National Ambient Air Quality Standard (NAAQS) of 35 $\mu$g/m$^3$ of particulate matter below 2.5 $\mu$m diameter ($PM_{2.5}$) set by the United States Environmental Protection Agency (US EPA) is regularly exceeded during wintertime (Simpson et al., 2019). Primary emissions in Fairbanks in winter are produced from domestic home heating systems, transportation, and power plant combustion sources (ADEC, 2019a) with enhanced demands due to frequent extreme cold episodes. Fairbanks is situated in a semi-open basin, surrounded by hills and valleys to the north, east and west. This topography, coupled with the regular occurrence of anticyclonic meteorological conditions sets up strong SBIs induced by strong surface radiative cooling (surface temperatures reaching -40 °C), and near-surface temperature gradients often exceeding 0.5 °C/m (Mayfield and Fochesatto, 2013; Malingowski et al., 2014; Ye and Wang, 2020), contributing to very stable meteorological ABL conditions. This favours regional atmospheric blocking (low wind speeds) and hinders pollutant dispersion leading to elevated surface concentrations (Mölders et al., 2011; Cesler-Maloney et al., 2022). Trapping of pollutants occurs, not only at the near-surface, but also in laminar layers aloft due to the presence of elevated temperature inversion (EI) layers that can form above SBIs (Angevine et al., 2001; Fochesatto et al., 2001; Mayfield and Fochesatto, 2013). Thus, pollutant emissions from elevated sources, such as power plant chimney stacks, can be influenced by the presence of stably stratified layers (Pasquill and Smith, 1983; Briggs, 1984; Tran and Mölders, 2011; Akingunola et al., 2018). Less stable conditions, with weak surface temperature inversions, can be induced by transient or cyclonic synoptic conditions or local sub-mesoscale flows under anti-cyclonic conditions (Maillard et al., 2022).

The Alaskan Layered Pollution and Chemical Analysis (ALPACA) project aims to improve understanding about wintertime Arctic air pollution including attribution of local pollution sources, chemical formation pathways of aerosols under cold and low photochemistry regimes, and pollution transport in the stratified ABL (Simpson et al., 2019). To study these issues, the international ALPACA field campaign took place in Fairbanks in January and February 2022 (ALPACA-2022). The campaign design, measurements, and first results are described in Simpson et al. (2024). Vertical profiles of trace gases and particles collected on a tethered balloon (Helikite) on the western edge of the city showed the regular presence of pollution layers close to the surface and aloft which emission tracer forecasts during the campaign attributed to power plant emissions (Simpson et al., 2024).

Here we aim to understand processes influencing the vertical and spatial distributions of air pollutants during the ALPACA-2022 field campaign. We use the FLEXible PARTicle-Weather Research and Forecasting (FLEXPART-WRF) Lagrangian particle dispersion model, driven by meteorological fields from WRF simulations generated by Alaska Department of Environmental Conservation (ADEC)/EPA. Transport of emission tracers of carbon monoxide (CO), sulfur dioxide ($SO_2$), nitric oxide (NO) and nitrogen dioxide ($NO_2$) are simulated in the Fairbanks area and their dependence on ABL structure and stability is investigated. Three of the selected trace gases (CO, $SO_2$ and $NO_2$) are defined as 'criteria pollutants' for human health by





the US EPA. Simulations include hourly-varying surface and non-surface emissions from ADEC/EPA for the campaign period (ADEC, 2023). This includes hourly power plant emissions based on data provided by the power plant operating companies. Buoyancy flux calculations using stack characteristics for each power plant, are used to calculate emission injection heights. The presence of temperature inversion layers in the ABL, which can trap power plant plumes, is also taken into account in a novel approach designed to cap injection heights. Variability in modelled tracers at different altitudes is linked to ABL stability including the presence of SBIs and EIs. Results are compared to vertical profile data and used to evaluate the power plant plume emission treatments, including plume rise and capping of plumes in multi-layered stratified temperature regimes (Mayfield and Fochesatto, 2013). Simulations are also evaluated against surface data and the sensitivity of the results to selected processes is explored, including meteorology and emission treatments. This is one of the first studies investigating the role of ABL meteorology on dispersion of elevated and surface emissions in the Arctic wintertime.

The methodology is described in Section 2 including details about the emissions, power plant plume rise parameterisation, FLEXPART-WRF model configuration and observations used for the model evaluation. Section 3 provides a brief overview of the ALPACA-2022 campaign including observations of trace gases and meteorology. Spatial and vertical distributions of modelled emission tracers over the Fairbanks area are presented in Section 4. Model results are evaluated against selected vertical profile data in Section 5 and surface observations in Section 6. The results of the sensitivity runs are also discussed in Sections 5 and 6. The main findings are presented in Section 7 together with wider implications and potential future research avenues.

## 2 Methodology

FLEXPART-WRF is run from 18 January to 25 February 2022 to explore the transport of local pollution during ALPACA-2022 using high temporal and spatial resolution emissions for surface and elevated sources, including emissions from five power plants within the Fairbanks region. Figure 1 shows the power plant and measurement/analysis locations discussed, together with the areas denoted by FNSB for air quality regulation (aqfairbanks.com). For the purposes of this study, the Fairbanks area encompasses Fairbanks and the adjoining town of North Pole. Section 2.1 describes the power plant and surface emissions used in this study. The injection altitude for the power plant releases is estimated according to a plume rise parameterisation, as described in Section 2.2. The WRF and FLEXPART-WRF model configurations and control simulations are described in Section 2.3 and the observations used for model validation are described in Section 2.4. All dates refer to the year 2022.

### 2.1 Emissions

Selected trace gases from power plant and surface sector emissions provided by the power plant companies and ADEC/EPA, respectively, are included in the FLEXPART-WRF simulations. Gridded hourly emission fluxes for CO, $SO_2$, NO and $NO_2$ were developed with the Sparse Matrix Operator Kernel Emissions (SMOKE) Processing System and data provided by ADEC/EPA for the duration of the campaign (CMAS Center, 2023). Tracers of CO, $SO_2$ and $NO_x$ (NO + $NO_2$) emissions are released from point sources and the near-surface sources with masses based on their respective emissions. These trace gases are chosen





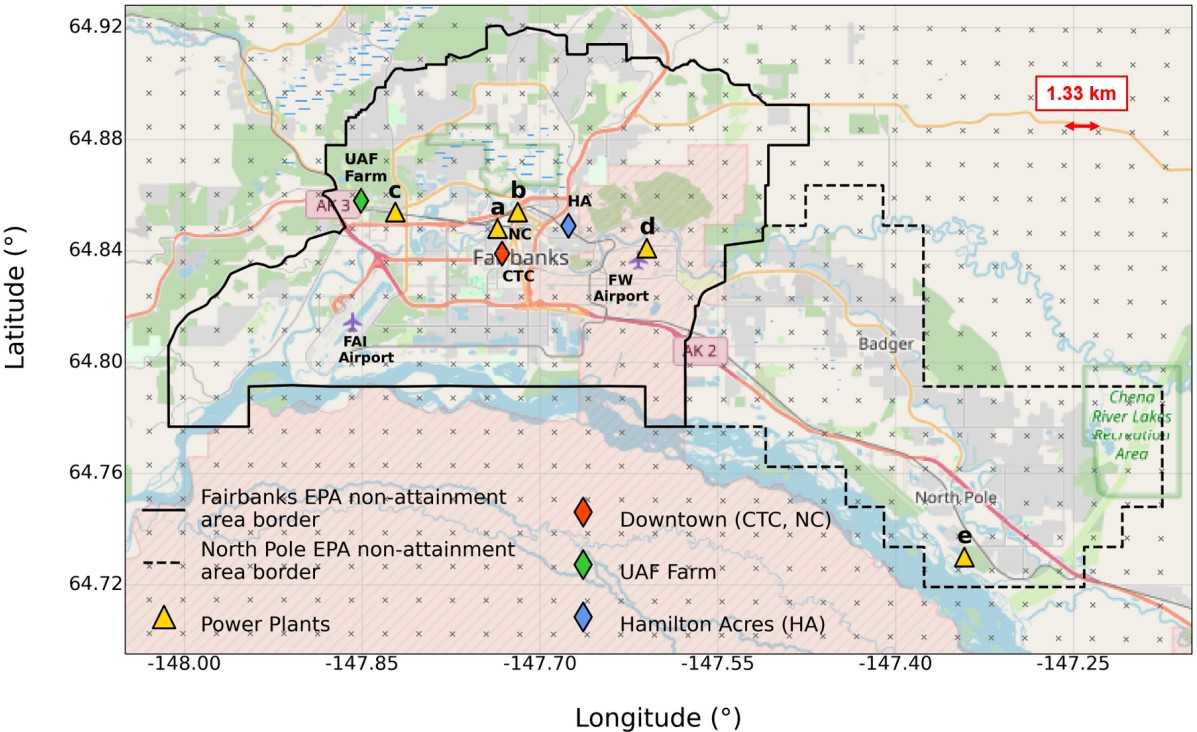

**Figure 1.** Map of Fairbanks and North Pole. Solid and dashed lines indicate the Fairbanks and North Pole EPA non-attainment areas (aq-fairbanks.com). The power plant locations (yellow triangles) correspond to the following power plants: a) Aurora, b) Zehnder, c) University Alaska Fairbanks (UAF), d) Doyon (Fort Wainwright), e) North Pole. Measurement sites at which trace gas measurements are available for model evaluation are indicated (see Section 2.3.1 for details). The two airports Fairbanks International Airport (FAI) and Fort Wainwright (military base) are also indicated. The grid cells for surface-emitted emissions, 1.33 km apart, are shown as small grey crosses. ©OpenStreetMap contributors 2024. Distributed under the Open Data Commons Open Database License (ODbL) v1.0.

based on the availability of emission data, and vertical profile and surface observations for model validation. Additionally, it is informative to compare reactive trace gases ($SO_2$ and $NO_x$) with CO which is a good tracer of transport and dispersion due to its long photochemical lifetime.

### 2.1.1 Power Plant Emissions

The power plants included in the model simulations are listed in Table 1 together with key stack parameter information including stack heights, fuel types, flue gas exit temperatures and velocities. For the 5 power plant facilities, there are 8 stacks included as separate point source releases in FLEXPART-WRF because the UAF and North Pole facilities have more than one power plant stack with variable characteristics that influence the plume buoyancy calculations (Section 2.2). Each power plant provided temporal emission information throughout the ALPACA-2022 campaign (see Fig. A2), with the exception of Doyon



| Power Plant | Stack Height (m) | Fuel Type | Flue Gas Exit Temperature (°C) | Flue Gas Exit Velocity ($ms^{-1}$) |
|---|---|---|---|---|
| Aurora | 48 | Coal | 149 | 78.5 |
| Zehnder | 18 | Diesel | 480 | 146 |
| UAF A | 20 | Diesel | 149 | 18.9 |
| UAF B | 20 | Diesel | 177 | 60.1 |
| UAF C | 64 | Coal | 129 | 23.4 |
| Doyon | 26 | Coal | 186 | 38.4 |
| North Pole A | 34 | Naphtha | 202 | 70.6 |
| North Pole B | 19 | Diesel | 292 | 176 |

**Table 1.** Power plant key characteristics. 'A', 'B' and 'C' denote separate burners and stacks at the same power plant facility. Locations of power plants are shown in Fig. 1.

(coal power plant at Fort Wainwright army base) where hourly 2020 data is used instead. Emissions for each power plant stack are provided at hourly time resolution except for UAF A and B for which only daily variability is available. Due to operational issues, the newer more efficient coal UAF C stack (64 m height, Table 1), was only running from 4 February (0900 AKST) onwards with hourly emissions provided. Prior to this, UAF A and B diesel generators (20 m heights) ran from 17 January, but with very low emissions from 1 February. Zehnder was operating only during cold polluted conditions in January and from 10 to 25 February, the operating periods were more frequent.

Figure 2 shows average hourly emissions of CO, $SO_2$, NO and $NO_2$ during ALPACA-2022 for each stack. Overall, Doyon and Aurora contribute most to $SO_2$ emissions, UAF C, Doyon and Aurora (coal-fired plants) to CO emissions and North Pole A, Doyon and Aurora to NO emissions. North Pole A has notably high $NO_x$ emissions because naphtha fuel has high nitrogen content and high $NO_x$ emission potential. Appendix A1 provides information about emission control strategies contributing to these differences. However, temporal emission variations and differences in stack characteristics also affect whether a particular power plant influences trace gas distributions.

### 2.1.2 Surface Emissions

Surface emissions on 1.33 km horizontal grid spacing are provided by ADEC/EPA for different sectors. Space heating emissions include commercial and residential sources using coal, distillate oil, gas and wood and industrial waste oil. The emissions are distributed over the first 4 WRF model layers with the fractions used by EPA: 15% 0-4 m; 69% 4-8 m; 15% 8-12 m and 0.01% 12-18 m and are processed by SMOKE according to ALPACA-2022 ambient temperatures. All other emissions are based on 2020 surrogates. On-road and non-road mobile sources take into account week-day and weekend differences and are



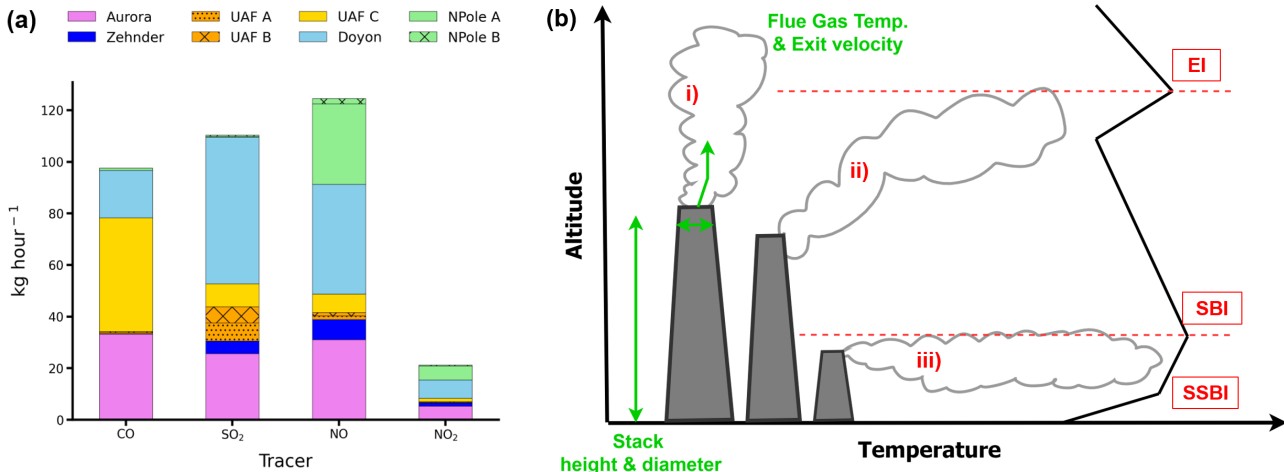

**Figure 2.** a) Average power plant emissions (kg/hour) during ALPACA-2022 for CO, $SO_2$, NO and $NO_2$. b) Schematic to illustrate the plume rise parameterisation used to simulate power plant injection altitudes. Examples of surface-based inversion (SBI), stratified SBI (SSBI) and elevated inversion (EI) layers are given, and i) corresponds to a plume with no inversion capping, ii) a plume which has been capped at an EI top, and iii) a plume which has been capped at an SBI top.

emitted at the surface (0-4 m). Non-point sources include stationary fuel combustion, commercial cooking and solvent use, and are also emitted at the surface. Airport emissions are available on the 38 WRF model levels, but are included from 0 to 18 m (first 4 levels), for the purpose of this study.

Further details about the surface emissions can be found in the ADEC emissions manual (ADEC, 2019a). Average emissions for CO, $SO_2$, NO and $NO_2$ for each sector, summed over the Downtown, HA, UAF Farm and Fairbanks non-attainment areas between 0-18 m are shown in Figure A1.

### 2.2 Plume Rise Parameterisation

Air pollutants released from a power plant stack have a buoyancy flux which is dependent on stack parameters (height, radius, flue gas exit temperature and velocity), along with ambient winds and temperatures in the proximity of the stack (Pasquill and Smith, 1983; Briggs, 1984; Akingunola et al., 2018). This information is required to more realistically predict plume injection altitudes (Bieser et al., 2011; Mailler et al., 2013; Guevara et al., 2014). Thus, power plant plume rise varies temporally depending on power plant operations and local meteorology, in particular related to atmospheric stability and the presence of temperature inversion layers in the Arctic winter. The plume rise parameterisation used here is summarised by the schematic in Fig. 2b, and is based on Briggs (1984) plume rise equations in stable conditions (Equations 1 and 2) where the buoyancy flux, denoted $F_b$ (units = $m^4\ s^{-3}$), is given by:





$$F_b = \frac{g}{\pi} \times (V \times \frac{T_s - T_a}{T_s}) \tag{1}$$

Where g = acceleration due to gravity, $T_s$ = effluent temperature, $T_a$ = ambient temperature at stack height, $V$ = volume flow rate of the effluent which is equivalent to: $v \times r^2$, where $v$ = exit velocity and $r$ = stack radius. The estimated plume rise height, $d_h$, is then given by:

$$d_h = 2.61 \times (\frac{F_b}{U_s})^{\frac{1}{3}} \tag{2}$$

Where $U_s$ corresponds to wind speed at closest altitude of radiosonde profile to the power plant stack height.

Stack parameters, combined with ambient temperatures and winds at the closest altitude to the stack height, interpolated from the Fairbanks International Airport (FAI) radiosonde profiles are used to calculate new injection altitudes above the stack height every 12 hours at 0300 and 1500 Alaskan Standard Time (AKST). For three missing radiosonde profiles, the assumption is made that the atmospheric profile has similar characteristics to the previous profile. The airport is 2 to 12 km from the Fairbanks power plant facilities and $\sim$ 25 km from the North Pole facility.

Diagnosis of plume rise injection heights is further complicated by the vertically stratified ABL in Fairbanks wintertime since the presence of SBIs or EIs can inhibit plume rise and cap the emissions. Although the plume rise calculation given in Equations 1 and 2 is generally appropriate for stable conditions, it is not necessarily representative of the extremely stable conditions that occur in winter in Fairbanks with a high latitude continental climate. SBIs are extremely shallow throughout ALPACA-2022 where the 25$^{th}$, 50$^{th}$ and 75$^{th}$ percentiles of SBI top heights determined from FAI radiosondes are 21 m, 46 m and 89 m, respectively. Stratified SBI (SSBI) layers within the SBIs can also develop close to the surface with even steeper positive temperature gradients, as well as EIs aloft, as depicted in Fig. 2 and discussed further in Section 3.

In order to take into account possible capping of power plant emissions at the injection (stack) location, the occurrence of SSBIs, SBIs and/or EIs is diagnosed from the FAI radiosondes every 12 hours. A layer fit routine is applied to the radiosonde profiles up to 3000 m to smooth the temperature profiles and assign temperature gradients (dT), according to Fochesatto (2015). Once this is applied, the inversion layer diagnosis is performed, based on the following conditions for each profile:

– Negative temperature gradients throughout the profile are removed as no inversions are observed.

– SBIs are assigned at the first change in sign of dT away from the surface (from positive to negative).

– EIs are assigned based on layers above the SBI, again when sign of dT of the next layer changes from positive to negative.

– SSBIs are assigned if there is at least one layer below the SBI, and the gradient changes between the SBI top and the surface but remains positive. If there is more than one layer using this description, the sub-layer with the steepest gradient is assigned as the SSBI.



For each diagnosed inversion layer, the temperature and altitudes at the top of the layer is assigned. Derived 12-hourly SBI
and SSBI altitudes are shown in Figure 3d and discussed further in Section 3. Injection heights for the power plant emissions
are capped at the top of the diagnosed inversion layers in the CTRL simulation, only when the inversion height exceeds the
stack height. Otherwise, the EI top aloft is used, if diagnosed. Emission tracers are released between +/-8 % of the calculated
plume rise height, to represent the plume thickness and, in the case of plume capping, that a small fraction of the emissions
penetrates the temperature inversion. Modelled power plant tracers are compared to available vertical profile observations in
Section 5. The sensitivity of the results to plume rise injection height and capping is also examined.

## 2.3 Model Simulations

This section provides details about the WRF and FLEXPART-WRF model configurations and the tracer simulations.

### 2.3.1 WRF Configuration

The dispersion of emission tracers released in the FLEXPART-WRF simulations are driven by hourly meteorology fields from
WRF model simulations provided by EPA for the ALPACA-2022 campaign (EPA-WRF from now on) at 1.33 km horizontal
resolution with 38 vertical levels. Twelve levels are in the lowest 555 m with three below 10 m (Gilliam et al., 2023). The
physics parameterisations used are the Rapid Update Cycle Land Surface Model (Benjamin et al., 2004), the Mellor-Yamada
Nakanishi Niino (MYNN) planetary boundary layer scheme (Nakanishi and Niino, 2009), the Rapid Radiative Transfer Model
(RRTM) short wave (SW) and long wave (LW) radiation (Iacono et al., 2008) scheme and explicit grid scale hydrometeors
using the Morrison micro physics scheme (Morrison et al., 2009). Observational nudging is applied using all available near-
surface measurements of temperature, humidity and winds and vertical profiles at a few key measurements sites for the duration
of ALPACA-2022. The near-surface observations include University of Alaska Fairbanks (UAF) Community Training College
(CTC), which was the main ground-based measurement site during ALPACA-2022, ADEC/EPA sites including NCORE (NC),
A-Street and Hurst Rd, and standard U.S. weather sites and local measurements from the MADIS database (MADIS, 2023).
CTC, NC and UAF locations are shown in Fig. 1. Above the surface, hourly Doppler Wind Light Detection And Ranging
(LiDAR) measurements at CTC (18 January - 7 February) and the UAF Farm (8-25 February) (Fochesatto et al., 2024), as well
as FAI radiosonde data, are assimilated into the EPA-WRF simulations. Finally, nudging to National Centers for Environmental
Prediction (NCEP) Global Forecast System (GFS) analyses are included above 300 m every 3 hours.

Evaluation of the EPA-WRF simulation generally found that, when surface measurements are assimilated, the errors at sites
not included in the nudging also decrease. For example, root mean square error (RMSE) temperature profile errors, compared
to FAI radiosondes, are as low as can be expected for a two-month simulation with errors at or below 1 K throughout the
troposphere. RMSEs of near surface temperatures (2, 3, 6, 11 and 23 m) are 2 K or less over the full ALPACA-2022 campaign
with multiple sites having RMSEs of 1.5 K or less. Given the difficulties simulating the winter climate of the Fairbanks area,
the model performs well (Gilliam et al., 2023). Statistical evaluation of temperature profiles and SBI/EI diagnosis performed
by Fochesatto et al. (2023) confirms the good model performance in terms of vertical temperature profiles. Wind speed and
direction biases are larger below 150 m (up to 2.5 m$s^{-1}$) than above the inversion layer (up to 1.5 m$s^{-1}$). 10 m wind speed



RMSEs at observation sites, including FAI and CTC, are closer to 1.5 m$s^{-1}$ with direction errors around 35° (Gilliam et al., 2023). Near-surface wind errors are important considerations for this study when evaluating the transport of surface emission tracers, and discussed in the next section.

### 2.3.2 FLEXPART-WRF Configuration

FLEXPART-WRF is a Lagrangian particle dispersion model used to simulate the transport of atmospheric trace constituents. FLEXPART is often run in backwards mode to identify key source areas, in particular for long-range transport studies e.g. Stohl et al. (2013). Forward simulations are used here to evaluate dispersion of emission tracers, and the relation to local and synoptic scale meteorology, over the Fairbanks area during ALPACA-2022.

The land use and topography data for the simulations are taken from EPA-WRF together with hourly winds and temperatures that drive horizontal and vertical transport of the tracers. The turbulent wind parameterisation in the ABL is either calculated internally using the Hanna scheme based on ABL parameters including ABL height, Obukhov Length and friction velocity (Hanna, 1984), or using external prognostic turbulent kinetic energy (TKE) from WRF that includes internal partitioning of TKE into horizontal and vertical components based on the Hanna scheme surface-layer scaling and local stability (Brioude et al., 2013). Brioude et al. (2013) suggested using the Hanna turbulence scheme in typical mid-latitude environments to ensure a well-mixed ABL, but this is not applicable in conditions where the ABL is stably stratified, as is predominantly the case in Fairbanks during winter. Simulations comparing Hanna (not shown here) and WRF-TKE schemes showed that WRF-TKE better captures differences in stability regimes around Fairbanks, for instance changes from stable to less stable conditions during the campaign, and is used here in the control simulation (CTRL). The ABL height (hmix), sensible heat flux and friction velocity are calculated in FLEXPART-WRF based on EPA-WRF input fields. Hmix has a default minimum height (hmin) of 100 m. If hmix is calculated to be lower than hmin, it is set equal to hmin. However, FLEXPART-WRF is generally used in conditions where strong stratification is not a distinct feature with more sunlight, turbulence, and stronger ABL mixing. Since FLEXPART-WRF is not currently configured for use in strongly stable conditions, hmin is used here as a proxy to investigate the sensitivity of tracer dispersion to the SBI layer height since it has a strong influence on trapping emissions at or close to the surface. Model simulations are sensitive to hmin due to the difficulties simulating shallow wintertime SSBIs or SBIs in WRF-EPA. Hmin is set to 20 m in the CTRL configuration rather than 100 m due to better agreement during stable conditions, and the sensitivity to different hmin values is explored in Section 6.

### 2.3.3 Tracer Simulations

Tracers of CO, $SO_2$, NO and $NO_2$ are released in each simulation based on emissions in the FNSB region and background concentrations from further afield are not included. Therefore, modelled mixing ratios are enhancements due to Fairbanks local emissions above background concentrations. All tracers are assigned masses according to their emission mass at hourly time resolution. For each power plant facility, 5000 particles are released hourly for each tracer and diurnal variability is calculated from the diurnal cycle for each power plant stack. Every 12 hours, a new injection height is assigned at the point of emission according to the plume rise parameterisation (Section 2.2). For the surface sources all emission sectors are summed





and an hourly emission variability is assigned to each tracer according to the total sectors, for example the diurnal cycle of CO is comparable to the on-road mobile sector. 80,000 particles are released hourly and weighted according to the emission mass in each 1.33 x 1.33 $km^2$ grid cell, extending over the wider Fairbanks and North Pole area (Fig. 1). Mobile and Non-point source sectors are released between 0-4 m only, while the space heating and airport emissions are released between 0-4, 4-8 and 8-12 m (layers 3 and 4, 8-12 and 12-18 m, respectively are summed due to negligible contributions in layer 4). The airport

emissions occurring higher than 18 m are not included in this study as they are generally transported to the south-west of the city (see Section 4). Modelled tracer concentrations are calculated in volume mixing ratios allowing comparison with observed CO, $SO_2$ and $NO_x$ (NO + $NO_2$) mixing ratios. In CTRL, emitted CO, $SO_2$ and $NO_x$ are treated as tracers and atmospheric lifetimes are not included. The influence of meteorology and emission treatments are explored in Section 6, together with atmospheric lifetimes (Appendix E5). Dry and wet deposition are included in CTRL only for $SO_2$ (see Appendix E3 for more

details) since these losses are not important for CO, and considered to be very small for $NO_x$ (Liu et al., 1987). Runs with and without dry and wet deposition of $SO_2$ only had a very small influence on the results (not shown). CTRL includes power plant plume rise and capping of plume injection heights, as described in Section 2.2. The NOCAP sensitivity includes power plant plume rise without capping at inversion heights and, in the NO-RISE sensitivity, emission tracers are released at the height of the stack. Results are discussed in Section 5. The CTRL setup and power plant sensitivities are summarised in Table 2.

| Simulation Name | Air tracers | Description |
|---|---|---|
| CTRL | CO | Surface and power plant tracers |
| | $NO_x$ | Power plant simulation includes plume rise parameterisation |
| | $SO_2$ + deposition | plus capping at diagnosed inversion heights (Section 2.2). |
| NOCAP | CO | Power plants only |
| | $NO_x$ | Plume rise parameterisation without capping |
| | $SO_2$ + deposition | |
| NORISE | CO | Power plants only |
| | $NO_x$ | No plume rise parameterisation - emission tracers released at |
| | $SO_2$ + deposition | stack height |

**Table 2.** Summary of the CTRL simulation setup and power plant plume rise sensitivity tests.

## 2.4 Observations

Model simulations are evaluated against surface and vertical profile observations from ALPACA-2022 sites shown in Fig. 1. Further details about measurement techniques, sites and the observations are given in Simpson et al. (2024). Hourly averaged surface observations of CO, $SO_2$ at CTC and NC, $NO_x$ at CTC, as well as wind speed and direction and temperatures at 3, 11 and 23 m at CTC and 3 and 11 m at NC are used to evaluate FLEXPART-WRF tracer concentrations and EPA-WRF

meteorology, respectively, in urban Fairbanks (Downtown in Fig. 1). Surface observations of CO and $NO_x$ at the ALPACA-





2022 house site in Hamilton Acres (HA), in the east residential area of Fairbanks, together with surface CO and meteorological parameters (2.5 m winds and 2 m and 11 m temperatures) at the UAF Farm site in the west of the city, are also used.

In-situ vertical profiles were measured at the UAF Farm site using the École Polytechnique Fédérale de Lausanne (EPFL) Helikite, a tethered balloon stabilised by a kite, from the surface up to 350 m (Pohorsky et al., 2024). Here, profiles of tem-
265 perature and $NO_x$ and CO, measured using the Micromegas low-cost sensor package at 15 s time resolution and calibrated using machine learning algorithms are used as well as EPFL mid-infrared absorption (MIRA) Pico CO data. EPFL $CO_2$ profiles measured by Vaisala GMP343 are used to check pollution presence observed in the trace gas profiles, since we expect $CO_2$ profile measurements to be highly reliable due to stability of the Vaisala instrument. More details about the EPFL instruments are provided in Pohorsky et al. (2024). All Helikite observations are averaged over 15 s time resolution for consistency.
Temperature profiles from the FAI radiosondes at 1500 and 0300 AKST are used to complement the analysis. Wind LiDAR attenuated backscatter data are also used to detect pollution (aerosol) plume presence between 40 m and 290 m (see Appendix D2 for details).

At the surface, strongly stable (SS) and weakly stable (WS) meteorological regimes are diagnosed based on observed temperature gradients ($\frac{dT}{dZ}$) per 100 m calculated using the 12-hourly FAI radiosonde data. To improve the temporal resolution,
temperature gradients (dT 23-3 m) at CTC, with hourly resolution (shown in Fig. 3c), are also used to account for variability not captured in the 12 hourly data. Criteria based on previous studies, including Cesler-Maloney et al. (2022) and Malingowski et al. (2014), are used to determine SS or WS regimes:

| Strongly Stable (SS) | Weakly Stable (WS) |
| --- | --- |
| $\frac{dT}{dZ}$ per 100 m $\geq$ 10 °C | $\frac{dT}{dZ}$ per 100 m < 10°C **and** dT 23-3 m < 2 °C |
| **or** | |
| $\frac{dT}{dZ}$ per 100 m < 10 °C **and** dT 23-3 m $\geq$ 2 °C | |

## 3 Meteorological Variability during the ALPACA-2022 Campaign

Figure 3 shows time series of observed surface $NO_x$, wind speeds, temperature gradients and stability analysis at the CTC
site in central Fairbanks and surface pressure at the UAF Farm, during ALPACA-2022. Overall, anticyclonic conditions were frequent during the campaign, resulting in cold, calm and generally clear-sky conditions (Simpson et al., 2024). This coincides with the presence of SBIs, high $NO_x$ concentrations, and generally lower wind speeds near the surface (Fig. 3, panels a-c). Due to a large scale synoptic variability during the campaign, anticyclonic conditions were interspersed with less stable conditions. This was due to the intrusion of low-pressure weather systems over central Alaska, notably during February. During these
conditions, weaker SBIs, lower $NO_x$ and higher surface wind speeds were observed. Fig. 3 variables (panels a-c) are coloured according to SS or WS regimes. Most notably, SS conditions prevailed in periods with strong positive surface temperature gradients resulting in higher $NO_x$. The presence of SBIs, SSBIs and EIs are also diagnosed from FAI radiosonde profiles as described in Section 2.2 and SBI and SSBI top heights are shown in Fig. 3d. SBI top heights range between 7 to > 200 m, and





**Figure 3.** Observations of a) surface NO$_x$ mixing ratios (ppb), b) wind speeds (3 m, m$s^{-1}$) and surface pressure (hPa) at the UAF Farm and c) temperature inversions (dT 23m-3 m, °C/m) at CTC during the ALPACA-2022 campaign (1 hour averages), coloured by strongly stable (SS) and weakly stable (WS) regimes, d) SBI and SSBI top heights (in meters) derived from the FAI radio-sondes (12-hourly), and e) stability strengths, $\frac{dT}{dZ}$ per 100 m (°C/ 100 m) derived from 12-hourly radiosonde data over given altitude bins. The meteorological periods used in the analysis are also indicated in panels a)-d). See text for details.

are lowest during SS conditions (often below 30 m). The presence of stable layers aloft is also diagnosed from radiosonde data
up to 300 m providing information about ABL stability (Fig. 3e) and used in the evaluation of the model results. For instance,
days with strong stability in the surface layer (0-25 m) and weaker stability aloft (> 25 m), such as 25 January, indicate a
decoupling of the surface layer from EIs that are linked to large scale meteorology. The range of stability strengths shown for
the surface layer also enables weaker and stronger SBI and SSBIs to be distinguished as shown in Fig. 3d.



In addition to stability regimes, the results are discussed in relation to three periods representative of the dominant meteorological situations that occurred during ALPACA-2022. The first period from 29 January to 3 February occurred when SS conditions dominated at the surface, and EIs were present aloft (Figs. 3d,e). Cold anticyclonic conditions persisted from 29 January to 1 February (named anticyclonic-cold (AC-C)), followed by a transition from AC to cyclonic conditions or transient-cold (T-C) from 2-3 February, as shown by a decrease in surface pressure in Fig. 3b. The T-C period corresponds to the formation of a high-low pressure gradient disrupting anticyclonic conditions. The surface layer was decoupled from aloft with SS conditions persisting at the surface as shown by strong surface stability strengths (30 - 60 °C per 100m, Fig. 3e), and SBI or SSBIs often below 30 m (Fig. 3d). The second period from 23 to 25 February, encompassed a transition from anticyclonic to cyclonic conditions with warmer temperatures compared to T-C (named Transition-Warm, T-W). Competing high and low-pressure weather systems, combined with a reduction in radiative cooling with respect to January, and the presence of high altitude clouds, contributed to warmer temperatures at this time. Intrusion of a warm air mass warmed the layers above the surface layer and increased the temperature gradients at the surface, as shown by the increased inversion strength at the surface between 24 to 25 February (Figs. 3d,e). These SS surface conditions resulted in $NO_x$ exceeding 250 ppb (Fig. 3a). The third period, from 5 to 21 February, is denoted the Mixed period, with transient, cyclonic and anticyclonic large scale meteorological conditions. SS conditions occurred at the surface but did not persist for longer than 24 hours and were interspersed with WS conditions. Enhanced surface pollution coincides with SBI presence as shown in Fig. 3d.

## 4 Vertical and Horizontal Dispersion of Emission Tracers

Figure 4 shows total surface-emitted plus power plant tracers of CO and $SO_2$ from CTRL near to the surface (0-10 m) and for $SO_2$ aloft (50-100 m, 200-300 m) averaged over the whole campaign for SS and WS conditions. Winds are also shown and provide an indication of average wind patterns. Below 10 m, simulated tracers are primarily localised in the main urban centres of Fairbanks and North Pole (non-attainment areas) with concentrations under SS conditions about two times higher than under WS conditions (SS CO > 500 ppb, WS CO > 200 ppb). This is due to weaker surface winds during SS conditions with no prevalent wind direction (see also observed and EPA-WRF winds at CTC (Fig. B1)). Tracers below 10 m include surface-emitted sources and elevated sources from space heating, airports and power plants. In particular, power plant emissions with low stack heights, such as Zehnder (18 m) if capping at a shallow SBI occurs while plants with higher stacks may be transported downwards more intermittently as discussed in the next section. Spatial differences in CO and $SO_2$ occur because of differences in the dominant surface emission sectors. Two hot spots with enhanced $SO_2$ correspond to airport emissions located to the south-west of central Fairbanks (FAI) and the east of downtown Fairbanks (Fort Wainwright army base), as shown in Fig. 1. Simulated $SO_2$ in downtown Fairbanks is primarily influenced by residential and commercial distillate oil heating sectors contributing > 90% of surface $SO_2$ emissions. This is reduced in the wider Fairbanks non-attainment area ($\sim$ 65% ) where airport emissions also contribute $\sim$ 30% (Fig. A1). $SO_2$ is smaller in North Pole which is mainly influenced by residential heating emissions. CO at 0-10 m is primarily influenced by on-road mobile emissions sector (Fig. A1).







**Figure 4.** a) Total power plant and surface-emitted tracers (enhancements above background in ppb) from CTRL for CO and $SO_2$ at 0-10m, and b) $SO_2$ at 50-100 m (i, ii) and 200-300 m (iii, iv) for strongly stable (SS) (left) and weakly stable (WS) (right) meteorological conditions. Wind vectors (black arrows) indicating average wind direction (degs.) and speeds (m$s^{-1}$) from EPA-WRF are shown and correspond to respective altitudes. The Fairbanks and North Pole non-attainment area borders are marked with black and white circles, power plants as white triangles and analysis locations as coloured diamonds as in Fig. 1.

**Figure 5.** a) Modelled (CTRL) $SO_2$ tracer as a function of altitude (m) and local time (AKST, hours) for i) total power plant emissions and (ii) total surface emissions at a) Downtown and b) UAF Farm. The WS and SS surface stability regimes indicated every 12 hours.

$SO_2$ is also simulated more substantially between 50-100 m under SS compared to WS conditions due to the stratification of the ABL and to a stronger north-easterly flow, possibly contributing to a wider regional influence. Above 50 m, enhanced





concentrations are found around the power plants suggesting that power plant emissions are the main contributors to $SO_2$ aloft (50-100 m, 200-300 m) (see Fig. C1 for results at 100-200 m). Modelled values are in agreement with DOAS $SO_2$

measurements ranging from 5-15 ppb collected between 73-191 m to the north-east of the Downtown area during polluted periods (Simpson et al., 2024). $SO_2$ is also influenced by power plant emissions at 200-300 m with enhancements up to 1-2 ppb. Interestingly, concentrations are enhanced during WS compared to SS conditions above 200 m, when winds are east-northeast (WS) as opposed to south-east (SS). These results may also be due to stronger upward transport during WS conditions. However, tracer enhancements are considerably smaller above 200 m and the bulk of pollution tracers are transported at lower

altitudes in dominant north-easterly outflow (to the south-west).

Additional results for CO and $NO_x$ are shown in Figs. C1 and C2, respectively. CO concentrations above 50 m relative to 0-10 m are inappreciable compared to $SO_2$ because the fraction of power plant to surface sector emissions of CO is much smaller. For instance, average power plant emissions (kg hour$^{-1}$) are a factor of 135 and 14 larger than surface emissions in the Fairbanks non-attainment area (kg hour$^{-1}$ km$^{-2}$) for $SO_2$ and CO, respectively, an order of magnitude difference (Fig. 2a

and Fig. A1c). Simulated $NO_x$ at 0-10 m and 50-100 m show similar spatial patterns to CO with surface concentrations of > 50 ppb, on average. Emissions are mainly from on-road mobile, and to a lesser extent from residential distillate oil. Power plant emissions also contribute aloft (> 10 ppb at 100-200 m) especially around the North Pole A stack which runs on naphtha, a fuel high in $NO_x$ emissions (see Fig. C2b).

The vertical distributions of $SO_2$ from power plant and surface-emitted sources at Downtown and the UAF Farm during

the campaign are shown in Fig. 5 and SS or WS conditions are indicated. As shown in Fig. 4, simulated near-surface mixing ratios are enhanced during SS compared to WS conditions. Emission tracers are concentrated in the lowest 20 m, in particular in the Downtown area due to strong vertical stratification. This capping at 20 m is related to running the model with hmin = 20 m in the FLEXPART-WRF turbulence scheme. Sensitivity of the model results to this parameter is examined further in Section 6. In contrast, lower surface concentrations are simulated during WS conditions. They are sometimes linked to stronger

vertical transport when a higher proportion of $SO_2$ is lofted upwards up to 300 m, for example on 6-7 and 9-10 February over Downtown. In other cases, reduced near-surface $SO_2$ mixing ratios are explained by enhanced horizontal dispersion e.g. on 24-25 January and 3-4 February (Downtown) due to stronger wind speeds between 2-6 m$s^{-1}$ (see also Fig. 3b). At the UAF Farm, the model simulates stronger vertical dispersion of both surface and power plant tracers (Fig. 5b), likely induced by stronger turbulence and wind speeds at this site (see also Fig. B2 showing stronger modelled and observed winds compared

to the Downtown sites). At HA, surface-emitted tracers are also maintained near the surface during SS conditions. Vertical transport appears larger than Downtown but smaller than the UAF Farm, markedly in February, when mixing heights greater than 20 m are depicted (see Fig. C3a).

Power plant tracers of $SO_2$ are generally simulated between 50-250 m over Downtown (Fig. 5a) with some dispersion towards the surface in both SS and WS conditions (e.g. 30 January to 1 February) and enhanced vertical transport in WS

conditions. The results also show that power plant $SO_2$ tracers are simulated at higher altitudes from 4-25 February over the UAF Farm (Fig. 5b). This is due to a change in operations from UAF A and UAF B to the UAF C facility which has a higher stack height (64 m) and runs on coal instead of diesel, also resulting in higher CO concentrations from power plant emissions



during this period (Fig. C3b). Power plant tracers also have a substantial impact at HA (e.g. $SO_2$, Fig. C3a), and are attributed predominantly to the Doyon stack to the south-east of the site (Fig. 1).

Overall, these results show that pollution is enhanced at the surface. Surface enhancements are considerable under SS conditions while aloft enhancements can be greater under WS conditions due to more vertical transport. In both cases, the results suggest that background pollution levels are being influenced by local air pollution sources from Fairbanks and North Pole. This regional pollution could be contributing to wintertime Arctic haze which has appreciably smaller concentrations of trace gases and aerosols. For example, 0.012-0.1 $\mu g/m^3$ (less than 0.1 ppb) of $SO_2$ and <0.01 $\mu g/m^3$ sulfate $SO_4^{2-}$ aerosols

were observed at the Alaskan remote sites Denali and Poker Flat in January 2000 (Tran et al., 2011).

## 5   Simulated Vertical Distributions and Power Plant Plumes

Pollution plumes were regularly intercepted by the Helikite at the UAF Farm above the surface layer (Simpson et al., 2024), and are used here to evaluate simulated vertical transport of tracers and, in particular, the power plant plume rise parameterisation. Selected cases with different meteorological regimes are investigated in more detail. As noted earlier, surface mixing ratios at

the UAF Farm are generally reduced compared to central Fairbanks. Differences in synoptic and local-scale meteorological conditions are influencing horizontal and vertical transport at this site together with lower emission magnitudes. However, above about 80 m, there is less influence of local valley flows at the UAF Farm and wind speeds/directions are more similar to central Fairbanks (Fochesatto et al. (2024)). Periods with east or north-easterly winds favoured transport of power plant pollution from Fairbanks to the UAF Farm.

In this analysis, CO and $NO_x$ pollution plumes are identified in each of the Helikite flights when elevated concentrations are observed above the 90[th] percentile in the data of the flight. To compare to the model results, that are enhancements above background, a polluted background is assigned in each flight using the modal concentration of the observed concentration distribution, and subtracted from observed plume mixing ratios. The resulting $\delta CO$ and $\delta NO_x$ enhancements are compared with the model results. In order to evaluate power plant plumes only, this comparison only uses observations above 30 m, away

from the influence of surface emissions. Some profiles of CO on 30 January and 10 February are removed due to issues with the CO sensor when the power was switched off to replace batteries and switched back on during the flight.

     Figure 6 shows the comparison of model results from CTRL and observed enhancements for each of the identified plumes for CO and $NO_x$ during the campaign when flights took place. The model is also run without the plume capping at the point of emission injection (run NO-CAP). CTRL generally captures plume presence aloft when compared with observed $\delta NO_x$

and $\delta CO$ above 30 m (Fig. 6a), although there are some displacements that could be due to temporal biases in modelled wind speeds and directions or in the diagnosed injection height. This could be due to using 12-hourly radiosonde data or due to spatial differences, for example using observed profiles at FAI rather than at each power plant location. In addition, the model is run with an hourly time resolution using EPA-WRF fields whilst the Helikite observations are collected at very high temporal resolution. The model is likely to have difficulties capturing this variability on small spatial scales. To examine the influence of

the model treatment of power plant emissions, the model is run without plume capping at temperature inversions at the point





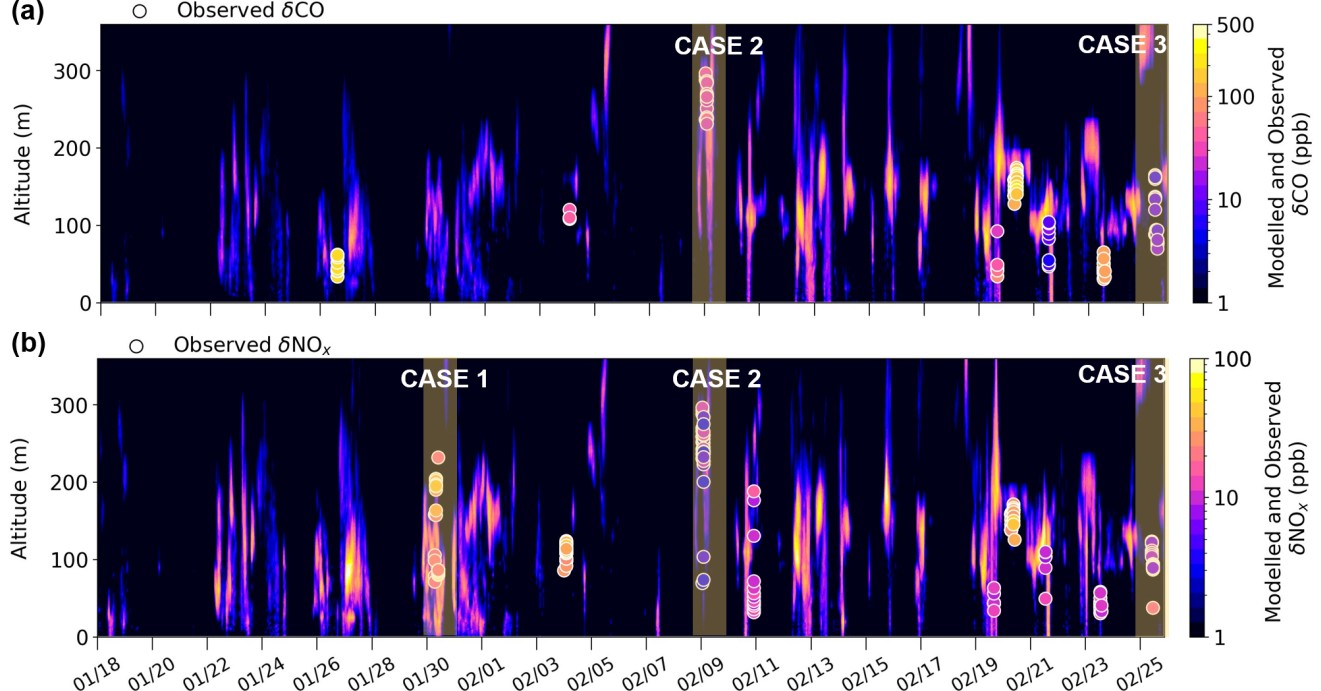

**Figure 6.** Comparison of modelled (CTRL) power plant and observed trace gas enhancements above background (> 30 m) for A) $\delta$CO (ppb), and B) $\delta$NO$_x$ (ppb) at the closest grid-cell to the UAF Farm. Periods with available observations are shown. Observed plume enhancements are shown as circles (ppb). Cases discussed in the main text are highlighted.

of emission (run NO-CAP) and without plume injection due to plume buoyancy, i.e. emissions at stack height (run NO-RISE). Results are shown in Fig. D1. Results are generally improved in CTRL compared to NO-CAP, or otherwise comparable. Results in NO-RISE are worse with tracers generally concentrated in the lowest 100 m and plume enhancements are overestimated compared to observations.

To evaluate model performance further, specific cases during the different meteorological situations discussed earlier are examined in more detail. They are selected to illustrate model behaviour after examination of all cases shown in Fig. 6. The first case on 30 January is during the cold stable polluted AC-C period. The second case from 8-9 February is during the Mixed period with lower surface concentrations and the third case on 25 February is at the end of T-W when temperatures were warmer but stable surface conditions resulted in high surface pollution levels. Results are shown in Figures 7 and 8. For each

case, observed plume enhancements of $\delta$NO$_x$ and/or $\delta$CO are shown together with model results from CTRL and NO-CAP or NO-RISE. Results are binned over altitude and averaged over the 4 grid cells surrounding the UAF Farm (Figs 6). Observed Helikite temperature profiles are shown together with radiosonde temperature profiles at 1500 and 0300 AKST for the days in question, for each case in Fig. 7a. Modelled vertical cross sections (total power plant tracer) for a period that extends several hours before and after the flight are also shown together with observed plume altitudes and concentrations (Figs. 8a). In addi-



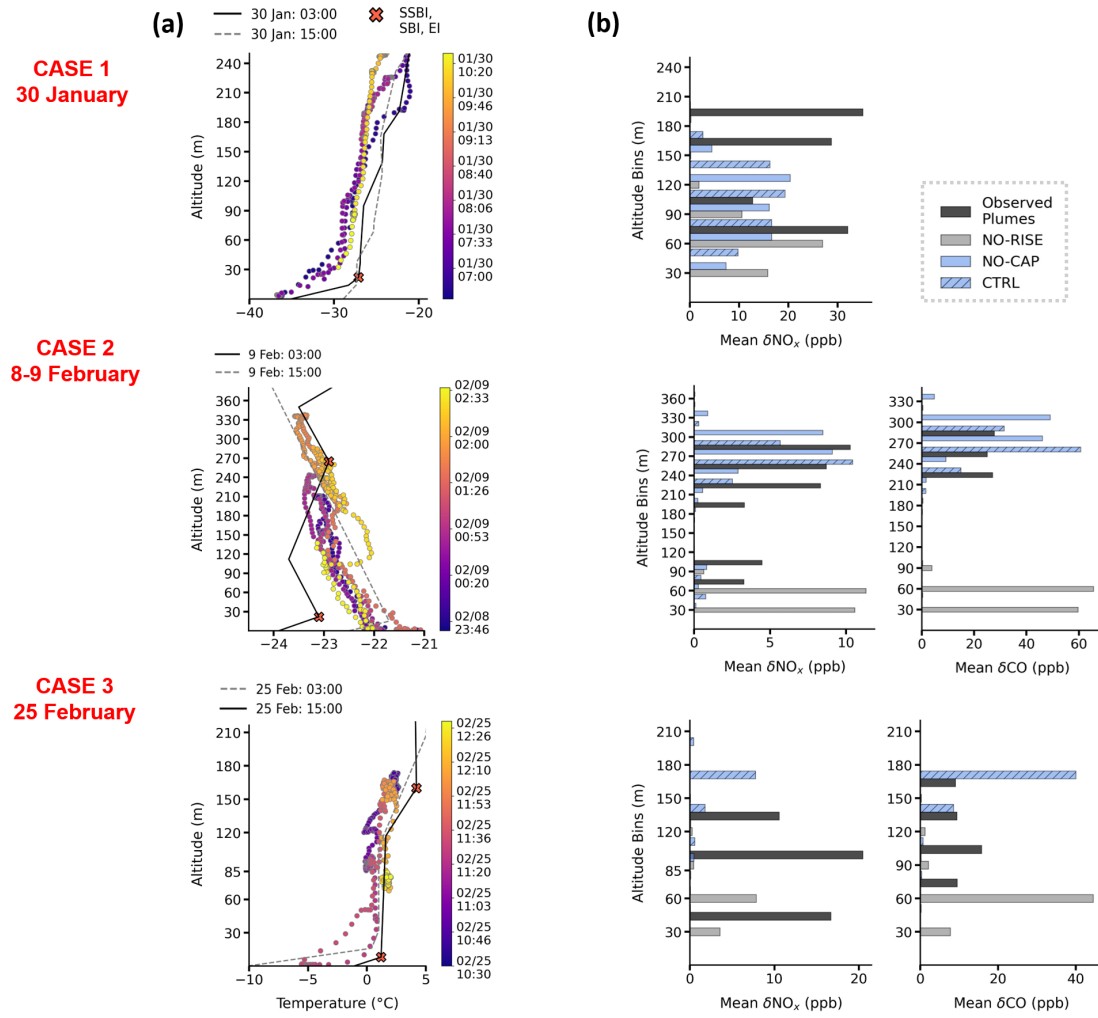

**Figure 7.** a) Temperature profiles (°C) recorded during the Helikite flight for the cases highlighted in a) and b). Coloured circles correspond to the time during the flight. Radiosonde temperature profiles (°C) used for the calculation of plume rise are also shown (solid black lines) and 12 hours before or after the flight (dashed grey line). Derived temperature inversion (temperature (°C), height (m)) are indicated as red crosses. b) Modelled (CTRL) power plant tracers compared to observations for CO and/or $NO_x$ (ppb) averaged over altitude bins every 30 m (indicated on the y-axis) at the time of the Helikite flight. For panels a) and b), Cases 1-3 are shown from top to bottom. See text for more details.

tion, hourly power plant contributions (%) (summed over all altitudes) are provided in Fig. 8b and the altitude corresponding to the 95[th] percentile for all contributing power plants are shown in Fig. 8c, allowing identification of the origin of different plumes.



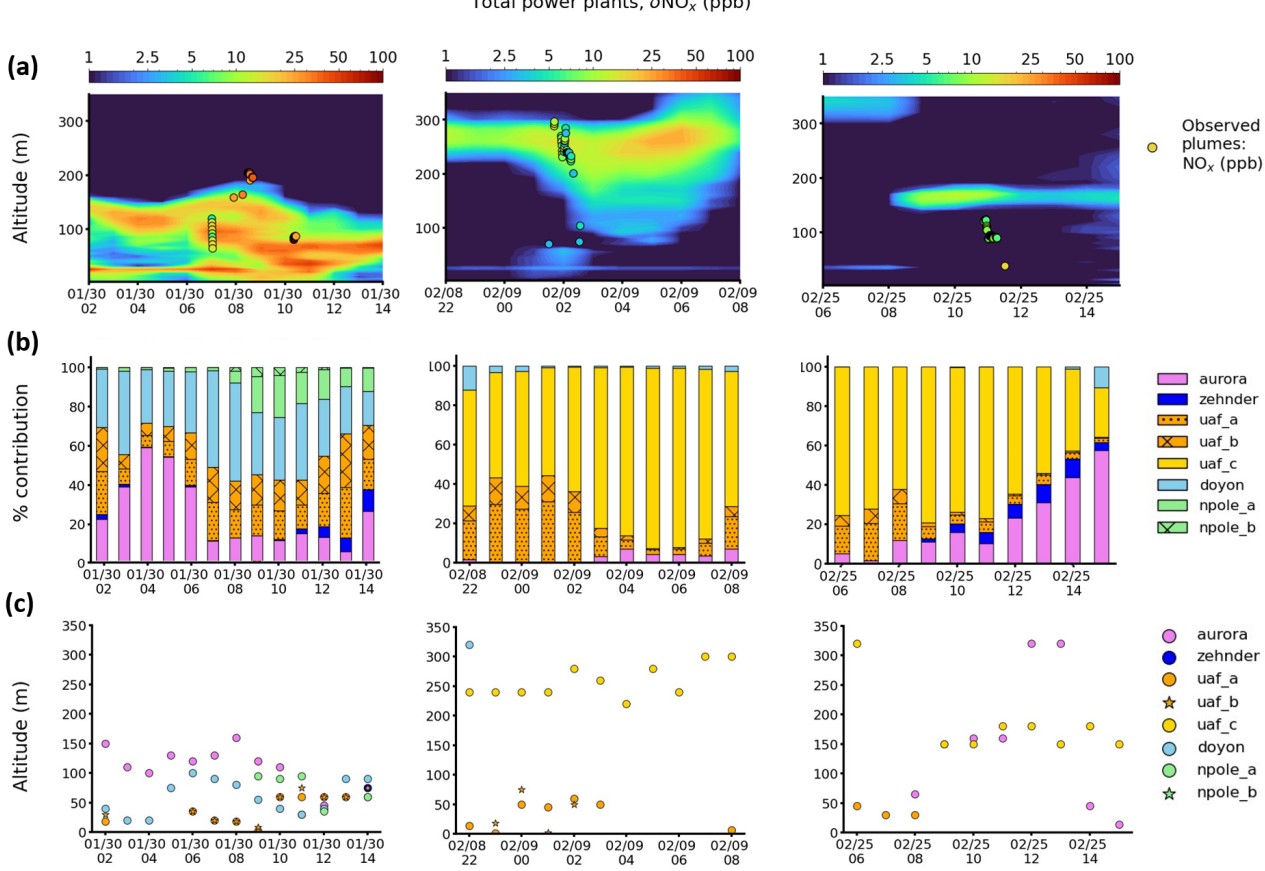

**Figure 8.** a) Vertical cross section of total simulated (CTRL) power plant tracer over several hours before, during and after each flight, with observations included as scatter points (as in Fig. 6). b) Hourly % contributions from different power plant stack throughout the vertical profile. c) Time series of 95th percentile for each contributing power plant stack (hourly). For panels a) to c), Cases 1-3 are shown from left to right. See text for more details.

**Case 1 - 30 January 2022, Fig. 7 top panel & Fig. 8 left panels:** This case during AC-C is characteristic of SS surface
conditions with low wind speeds (<1 m$s^{-1}$) from the east or north-east ($\frac{dT}{dZ}$ up to 30 °C per 100 m at 0-25 m) and some stratification in the layers aloft ($\frac{dT}{dZ}$ up to 10 °C per 100 m, at 100 - 300 m, see Fig. 3e). Only NO$_x$ observations are available with two plumes identified between 70-110 m and 160-210 m altitude, just below elevated inversions observed in the Helikite temperature profile data (Fig. 7a). Modelled plumes are between 30-150 m and attributed predominantly to Doyon and UAF A and B. Aurora contributes most at 120-150 m, notably between 0700 and 0900 AKST with some downward transport to around
100 m between 0900 and 1030 AKST (Fig. 8a,c). The EI in the Helikite temperature profile occurs around 210 m, indicating trapping of the upper observed plume. However, no capping is applied in CTRL for the Aurora emissions because the predicted plume rise is lower than the radiosonde EI (398 m). Therefore, the calculated emission injection height for Aurora until 0900



AKST is 150 m (midpoint) and is the same in CTRL and NO-CAP. Also, at 0300 AKST on 30 January (time of radiosonde), the observed LiDAR wind speeds at CTC (900 m south-east of Aurora) were up to 4 m$s^{-1}$ at the Aurora stack height (48

m), while the radiosonde wind speeds were lower than 1 m$s^{-1}$ (> 5 km south-west of Aurora). This suggests that the Aurora plume is not simulated over the UAF Farm due to spatial discrepancies in the wind speeds. This may also contribute to an underestimation of the plume injection height and explain why the model does not capture the observed plume at 160-210 m.

    **Case 2 - 8-9 February 2022, Fig. 7 top panel & Fig. 8 middle panels:** This case, during the Mixed period, contrasts to the previous SS case and is characteristic of WS conditions. Wind directions from the south to south-west transport pollution to the

north (0-500 m altitude). At the time of this local nighttime Helikite flight, conditions were more stable than the daytime with pollution trapped at the surface due to a drop in wind speeds and an increase in SBI strength (Fig. 3). A weak EI was observed aloft at 260 m at 0300 AKST as shown in Fig. 7a, resulting in dispersed plumes of NO$_x$ and CO aloft over the UAF Farm. In this case, the radiosonde-derived EI agrees with the observed Helikite EI and, even if the stratification is rather weak, a layer of trapped emissions, with observed CO and NO$_x$ enhancements, is evident. Modelled plume enhancements from CTRL

compare well with the observed plume aloft between 250-300 m with some downward transport (to 200 m) towards the end of the flight which is also observed. This plume is attributed to UAF C. In this case, EI capping is applied and improves the modelled plume altitude compared to NO-CAP. Simulated plumes are much too low (30-60 m) in NO-RISE highlighting the need to include plume buoyancy calculations (Fig. 7b) as shown in previous studies (e.g., Briggs, 1984; Akingunola et al., 2018). A lower altitude plume between 50-100 m is only observed in the NO$_x$ data. Only small enhancements (< 1 ppb) are

simulated in CTRL, and also in NO-CAP, and attributed to UAF A and B stacks. They have lower stack heights and run on diesel which may explain the lack of observed CO plume enhancements. The model may be underestimating NO$_x$ in this case, or surface-emitted tracers may be lofted vertically and contribute to the observed plume at 50-100 m.

    **Case 3 - 25 February 2022, Fig. 7 top panel & Fig. 8 right panels:** This case is at the end of T-W. An elevated plume is observed between 85-120 m in NO$_x$ and 120-160 m in CO. The Helikite temperatures indicate EIs near 85 and 120 m (Fig. 7a).

The plume aloft, which encompasses most of the data points for both $\delta$CO and $\delta$NO$_x$, is captured in CTRL but not in NO-CAP. Ths is due to the EI observed by the 1500 AKST radiosonde (160 m) that is used to calculate the plume injection height in CTRL, while in NO-CAP the injection altitude is approximately 500 m, demonstrating the importance of the capping parameterisation. However, the modelled plume altitude is likely overestimated by approximately 30-50 m due to the EIs occurring at lower altitudes at the UAF Farm (Fig. 7a). There is better agreement of modelled $\delta$CO with the observed enhancements than

for $\delta$NO$_x$ (Fig. 7b and 8a), which can be explained by contributions from different power plants. The UAF C stack contributes to $\delta$CO directly at the UAF farm as shown in Fig. D3a. UAF C, Aurora and Zehnder contribute to modelled $\delta$NO$_x$ (Fig. 8c) but UAF C NO$_x$ emissions are low compared to CO because the stack has more NO$_x$ emission controls (ADEC, 2019b) (see Appendix A1). Aurora and Zehnder plumes are displaced to the south of the UAF Farm due to a displacement in modelled wind direction (north-east vs east). This results in stronger transport to the south, displacing the simulated plumes slightly

south of the UAF Farm (Fig. D3a), most likely explaining the underestimated modelled NO$_x$ enhancement. NO$_x$ plumes are also displaced southward in a supplementary case on 3-4 February from the Doyon power plant between 120 - 180 m (Fig.





D2c in Appendix D1).

Appendix Figure D3 shows doppler wind LiDAR observations for cases 1 (CTC) and 2 (UAF Farm). In each case, plumes
are identified by the wind LiDAR at a comparable altitude to the identified plumes at the Farm. Although the wind LiDAR
is sensitive to aerosols, and not trace gases, the results suggest that power plants are also a source of aerosols over Fairbanks
(more details in Appendix D2).

Overall, based on the evaluation of these cases, the CTRL run including plume rise and capping using information on
the ABL structure often performs best compared to available profile observations. Therefore, CTRL is used in the following
examination of processes influencing surface pollution during ALPACA-2022. Evidently, plume rise and capping has to be
taken into account but ideally using vertical profile information at the point of injection would be required to improve the
plume rise calculations. Discrepancies in modelled winds sometimes lead to displacement in modelled plumes, as shown by
case 3 and the supplementary case on 3-4 February (Appendix D1). This is important for power plant facilities located away
from the UAF Farm e.g. Aurora and Doyon.

**6   Processes Influencing Simulated Surface Trace Gases**

Model results from the CTRL run are initially evaluated against surface observations. To understand model behavior during
different meteorological conditions, and to examine possible causes of model discrepancies, the sensitivity of model results
to various processes is then explored. This analysis is not exhaustive in terms of the processes considered and other possible
processes are highlighted in the discussion of the results.

**6.1   Evaluation against surface observations**

Total modelled CO, $SO_2$ and $NO_x$ from surface-emitted and power plant sources in the surface layer between 0-5 m compared
to available surface observations as a function of time, Downtown, are shown in Figure 9a. Note that $SO_2$ results include wet
and dry deposition but their influence is small as noted earlier (also Appendix E3). Downtown observations correspond to CTC
and NC data averaged for CO and $SO_2$ and compared with the closest grid cell to the Downtown area, while $NO_x$ observations
are only available at the CTC site. Diurnal cycles of the observations and model results during the entire campaign (all data)
and events AC-C, T-C, Mixed and T-W are shown in Figure 9b. Results for the HA site in eastern residential Fairbanks and
the UAF Farm are provided in Figures E1 and E2. Normalised mean biases (NMBs) and normalised mean errors (NMEs) for
Downtown using hourly results are provided in Table 4. Both metrics are shown as fractions with no units and Equations are
given in Appendix E1. Tables E1 and E2 correspond to HA and UAF Farm in Appendix E2.
As discussed earlier, observed CO, $SO_2$ and $NO_x$ are enhanced during stable conditions. Observed variability with larger
concentrations in SS compared to WS conditions is generally captured. CO concentrations and variability are reasonably well
simulated. However, while the NMB is 0.02 over the entire campaign (all data), the NME is 0.52 (Table 4). There are also
negative biases during the stable transient events T-C and T-W (NMBs = -0.34 and -0.55) and a strong positive bias during the



| Sensitivity Simulation | Tracer | Description |
|---|---|---|
| CONST-EM | CO | CTRL with constant emissions |
| | $SO_2$ | |
| | $NO_x$ | |
| NOx_Emissions | $NO_x$ | CTRL + temperature dependent diesel vehicle emissions |
| NOx_Emissions_LT | | NOx_Emissions + variable photochemical lifetime |
| SO2_SOR | $SO_2$ | SO2_CTRL + sensitivity to oxidation ratio |
| MixH_100_CO | CO | CTRL + hmin=100 m |
| MixH_100_SO2 | $SO_2$ | SO2_SOR + hmin=100 m |
| MixH_100_NOx | $NO_x$ | NOx_Emissions_LT + hmin=100 m |
| MixH_10_CO | CO | CTRL + hmin=10 m |
| MixH_10_SO2 | $SO_2$ | SO2_SOR + hmin=10 m |
| MixH_10_NOx | $NO_x$ | NOx_Emissions_LT + hmin=10 m |

**Table 3.** Surface sensitivity simulations. See text for details.

Mixed period (NMB = +0.5). Since CO has a long photochemical lifetime in winter of the order of months, discrepancies may
be caused by meteorology.

$SO_2$ and $NO_x$ tracer variability in CTRL is comparable to that of CO. However, for $SO_2$ there are large overestimates, in
particular during the Mixed event (NMB = +1.26, NME = 1.37, Table 4). The main source of $SO_2$ in Downtown is residential
distillate oil in the space heating sector emissions (Fig. A1). This source is released up to 12 m with 85% of the emissions
released above 5 m. Therefore, these emissions can be transported to the surface as well as higher in altitude. Modelled $SO_2$
appears to be sensitive to the vertical transport of these emissions and is explored in the sensitivity analysis. Although the
photochemical loss of $SO_2$ by OH is not considered to be important during the winter (e.g., Green et al., 2019), oxidation by
other reactions may be important. In contrast, the model significantly underestimates observed $NO_x$ notably in SS conditions
(NMB = -0.65 and -0.8 events AC-C and T-C with comparable NMEs, Table 4). Moreover, an overestimate might be expected
because the lifetime of $NO_x$ is not included in CTRL. The sensitvity of modelled $SO_2$ and $NO_x$ to processes governing their
lifetimes are considered in Appendix E5.

Observations at HA in the East Residential area of Fairbanks follow the same general variability as Downtown but differ
during the strongly stable events AC-C, T-C and T-W, as highlighted by the diurnal variations (Fig. 9b and Fig. E1b). The
Downtown sites are located close to main roads leading to higher observed $NO_x$ mixing ratios than at the HA site. CO magni-
tudes are more comparable because of higher contributions from residential wood burning at the HA site, as supported by the
strong peak around 0600 AKST in the diurnal cycle of CO at HA (Fig. E1b). However, Downtown, the diurnal cycle follows





the on-road mobile sector (Fig. 9b). The agreement between model and observations is weaker at HA, for instance NME = 0.56 (Table E1) in contrast to 0.37 Downtown (Table 4) for CTRL_CO because the horizontal resolution of the surface emissions (1.33 km grid spacing) may be too coarse to sufficiently capture small spatial differences within the city. Emission source contributions for CO and $NO_x$ in the Downtown and HA areas are comparable (Fig. A1), supporting this argument. It should also

be noted that the model results shown in Fig. 9 are interpolated onto the same grid as the emissions (1.33 km). Furthermore, the locations of meteorological data assimilated in EPA-WRF are biased towards the Downtown area, potentially leading to more realistic simulated meteorology. Moreover, during SS conditions, horizontal transport is hindered in Fairbanks, leading to a large variability in the observations at different locations. This was demonstrated by Robinson et al. (2023) during multiple mobile sniffer drives of $PM_{2.5}$ around Fairbanks.

At the UAF Farm site, smaller surface CO mixing ratios are observed. Over the entire campaign, NMBs and NMEs are comparable to Downtown but biases are higher when stable conditions influence Downtown more than the UAF Farm, notably during the AC-C period. A local flow that originates from large-scale north-easterly winds intermittently descends the Goldstream valley to the north-west resulting in a dominant north-westerly flow at the UAF Farm towards the surface (Maillard et al., 2022; Fochesatto et al., 2024). The wind direction of the local flow is captured by EPA-WRF at 10 m due to data

assimilation, however, underestimations in horizontal wind speeds can occur when strong static stability is observed (strong temperature gradients) due to difficulties simulating dynamic instability (turbulence/wind shear) induced by the local flow (e.g. Fig. B2 during AC-C).

## 6.2 Sensitivity Simulations

Following the initial evaluation, the sensitivity of modelled tracers to meteorology, emissions and vertical mixing are explored.

Description of the sensitivity to $NO_x$ trace gas lifetimes are included in Appendix E5. The series of sensitivity simulations, carried out to better understand processes influencing modelled surface tracers and which may help explain model biases, are summarised in Table 3.

### 6.2.1 Sensitivity to Meteorology

As noted earlier, model biases can be induced by errors in EPA-WRF or treatments in FLEXPART-WRF of vertical or horizontal

transport. Of particular interest are discrepancies during cold stable periods with poor air quality. For example, the NMB and NME of CO during T-C are -0.34 and 0.39, respectively. Temperature gradients at CTC (dT 23-3 m) are generally well captured by EPA-WRF since the model is nudged with these temperatures. However, dT 23-3 m is not well reproduced in EPA-WRF during T-C on 2-3 February when the very large observed dT (up to 8 °C) is underestimated by 3 °C (Fig. B1b). 23 m wind speeds measured at CTC are also overestimated resulting in stronger horizontal transport at the surface than observed (Fig. B1).

There is also more upward vertical transport of tracers on 1 February (Fig. 5). Consequentially, modelled CO is underestimated during T-C. This could be explained by a transient synoptic condition (i.e., a low-pressure weather system) in upper layers above the surface layer from 2-3 February (T-C), disrupting the vertical stratification provided by the stable anticyclonic conditions that occurred from 29 January to 1 February (AC-C). Yet at the surface, local-scale radiative cooling persisted and







**Figure 9.** a) Total modelled surface and power plant tracers as a function of time between 0-5 m for CTRL and selected sensitivity simulations described in Table 2 compared to available surface observations, Downtown for i) CO, ii) $NO_x$ and iii) $SO_2$. b) Diurnal cycles, Downtown, for observations (black) and model simulations (colours as in (a)), averaged over all data (left) and over events A-C, T-C, Mixed and T-W.

strong temperature gradients were maintained and strengthened due to the arrival of the warm air mass aloft, as also observed
by Mayfield and Fochesatto (2013).

During the warm polluted period T-W at the end of the campaign (23-25 Feb) under AC to cyclonic transient conditions, CO and $NO_x$ are underestimated compared to observations. This was an unusual event when the SBI was very strong but temperatures were warmer (-10 to 5 °C) than in AC-C and T-C, for example. Whilst observed temperature gradients at CTC are captured well by EPA-WRF during this period, horizontal transport appears to be overestimated because the EPA-WRF
wind speeds are slightly higher than observations close to the surface e.g. at 10 m (Fig. B1). This may partly explain the low model NMB during this period (CO = -0.55 and $NO_x$ = -0.74, respectively, Table 4). During the Mixed period, the dT (23-3 m) is often too high compared to observations, for example on 16 February (Fig. B1). This leads to overestimates in modelled mixing ratios, notably for $SO_2$ (Fig. 9).

In order to explore the influence of meteorological variability on simulated tracers at the surface, the model is run with
constant emissions averaged over the entire campaign (run CONST-EM). Results are examined for CO, since, due to its long lifetime, CO is more influenced by meteorology. Note that $NO_x$ and $SO_2$ with constant emissions were also simulated (not shown), and are more comparable to the CTRL simulation. Differences in diurnal cycles for CTRL and CONST-EM CO are shown in Fig. 9b). CTRL shows better agreement compared to the observations for the whole campaign, AC-C and T-C. Simulated CO mixing ratios are enhanced during the daytime due to the diurnal variability in the emissions, which are
dominated by the on-road sector Downtown (Fig. A1). These results suggest that modelled CO biases can be explained partly by emission variability and by differences in modelled and observed meteorology influencing tracer transport and mixing as well as ABL stability. Discrepancies are linked in part to the EPA-WRF simulation as discussed above and also to treatments in FLEXPART-WRF. The sensitivity of results to the mixing height parameter in the FLEXPART-WRF BL scheme is examined further in Section 6.2.4.

**6.2.2   Sensitivity to Vehicle $NO_x$ Emissions at Cold Temperatures**

The underestimate of $NO_x$ during SS periods, such as AC-C, is more significant than CO and $SO_2$ and may indicate a missing source of $NO_x$. The on-road sector is an important source of $NO_x$ in the Downtown area (Fig. A1) for which diesel is the largest contributor, even if the fraction of diesel vehicles is rather low (9% diesel versus 90% gasoline vehicles in Fairbanks non-attainment area, EPA 2022). The diesel fleet in the area is predominantly made up of heavy duty trucks. In 2022, EPA used
MOtor Vehicle Emission Simulator 3 (MOVES3) (U.S. EPA, 2021) to calculate on-road emissions, which were subsequently processed with the SMOKE model. MOVES3 includes a higher incremental temperature dependence of CO compared to $NO_x$ gasoline emissions, which is important because CO emissions are much higher than $NO_x$ emissions for gasoline. In



| Simulation Name | NMB | | | | | NME | | | | |
|---|---|---|---|---|---|---|---|---|---|---|
| | All Data | AC-C | T-C | Mixed | T-W | All Data | AC-C | T-C | Mixed | T-W |
| CTRL CO | 0.02 | 0.02 | -0.34 | 0.5 | -0.55 | 0.52 | 0.39 | 0.37 | 0.68 | 0.56 |
| CONST-EM CO | 0.03 | -0.17 | -0.46 | 0.54 | -0.45 | 0.54 | 0.37 | 0.5 | 0.73 | 0.5 |
| MixH_100_CO | -0.19 | -0.17 | -0.55 | 0.21 | -0.64 | 0.47 | 0.34 | 0.55 | 0.48 | 0.64 |
| MixH_10_CO | 0.3 | 0.45 | -0.04 | 0.79 | -0.45 | 0.66 | 0.66 | 0.35 | 0.92 | 0.51 |
| CTRL NOx | -0.46 | -0.65 | -0.8 | -0.03 | -0.74 | 0.69 | 0.66 | 0.8 | 0.66 | 0.74 |
| NOx Emissions | -0.07 | -0.05 | -0.47 | 0.59 | -0.67 | 0.68 | 0.45 | 0.5 | 0.93 | 0.67 |
| NOx Emissions_LT | -0.23 | -0.17 | -0.5 | -0.21 | 0.7 | 0.61 | 0.41 | 0.52 | 0.7 | 0.7 |
| MixH_100_NOx | -0.25 | -0.32 | -0.67 | -0.01 | -0.77 | 0.61 | 0.42 | 0.67 | 0.65 | 0.77 |
| MixH_10_NOx | -0.2 | 0.2 | -0.25 | 0.44 | -0.62 | 0.61 | 0.53 | 0.44 | 0.76 | 0.62 |
| CTRL SO2 | 0.6 | 0.03 | -0.3 | 1.26 | 0.6 | 0.94 | 0.49 | 0.4 | 1.37 | 0.95 |
| SO2 SOR | 0.47 | -0.02 | -0.29 | 1.08 | 0.32 | 0.84 | 0.48 | 0.37 | 1.22 | 0.74 |
| MixH_100_SO2 | 0.03 | -0.26 | -0.57 | 0.43 | -0.08 | 0.62 | 0.42 | 0.63 | 0.7 | 0.54 |
| MixH_10_SO2 | 1.08 | 0.5 | 0.1 | 1.71 | 1.0 | 1.31 | 0.79 | 0.55 | 1.8 | 1.18 |

**Table 4.** Normalised mean biases (NMBs) and normalised mean errors (NMEs) of model simulations (total tracers) at the surface, Downtown, compared to surface observations Downtown (CTC and NC averaged), at hourly time resolution. NMBs and NMEs are given for all data and the meteorological events AC-C, T-C, Mixed and T-W. The values highlighted in green correlate to optimal or improved simulations for each period, while red colours correspond to large positive or negative biases or errors.

addition, cold-temperature dependencies for diesel vehicle cold starts for both CO and $NO_x$ are set to zero, however, data was only collected down to +1.5 °C in that study (U.S. EPA, 2015). More details are provided in Appendix E4. Several studies
have shown that $NO_x$ emissions from diesel vehicles are higher at cold temperatures, in particular in modern vehicles with selective catalyst reduction (SCR) units that have been introduced following more stringent emission regulations. Failure to heat the diesel exhaust fluid (DEF) injection to the required temperature to initiate the SCR units is considered to contribute to enhanced emissions (Weber et al., 2019; Selleri et al., 2022; Seo et al., 2022; Wærsted et al., 2022). Ambient temperatures in Fairbanks reach -40 °C, up to 2 °C lower than the lowest temperatures in these studies. Hence the cold temperature dependence
for diesel $NO_x$ emissions may be too weak in MOVES3, resulting in a substantial underestimate in modelled $NO_x$ during cold stable conditions. Other emission inventories, such as CAMS, also have a weaker temperature dependence for $NO_x$ vehicle emissions than CO at low temperatures (Guevara et al., 2021). This may be because the current emissions inventories are based on older vehicles without SCR units that are associated to newer diesel vehicles or due to limited research on this topic in very cold environments.

The possible contribution of temperature dependent diesel emissions to CO and $NO_x$ in Fairbanks is investigated first by summing the EPA emissions for each sector over the volume of the box covering 4 grid cells in Downtown and up to 10 m to estimate hourly concentrations that are compared to CTC observations. The results are shown in Fig. 10 averaged over 3 °C





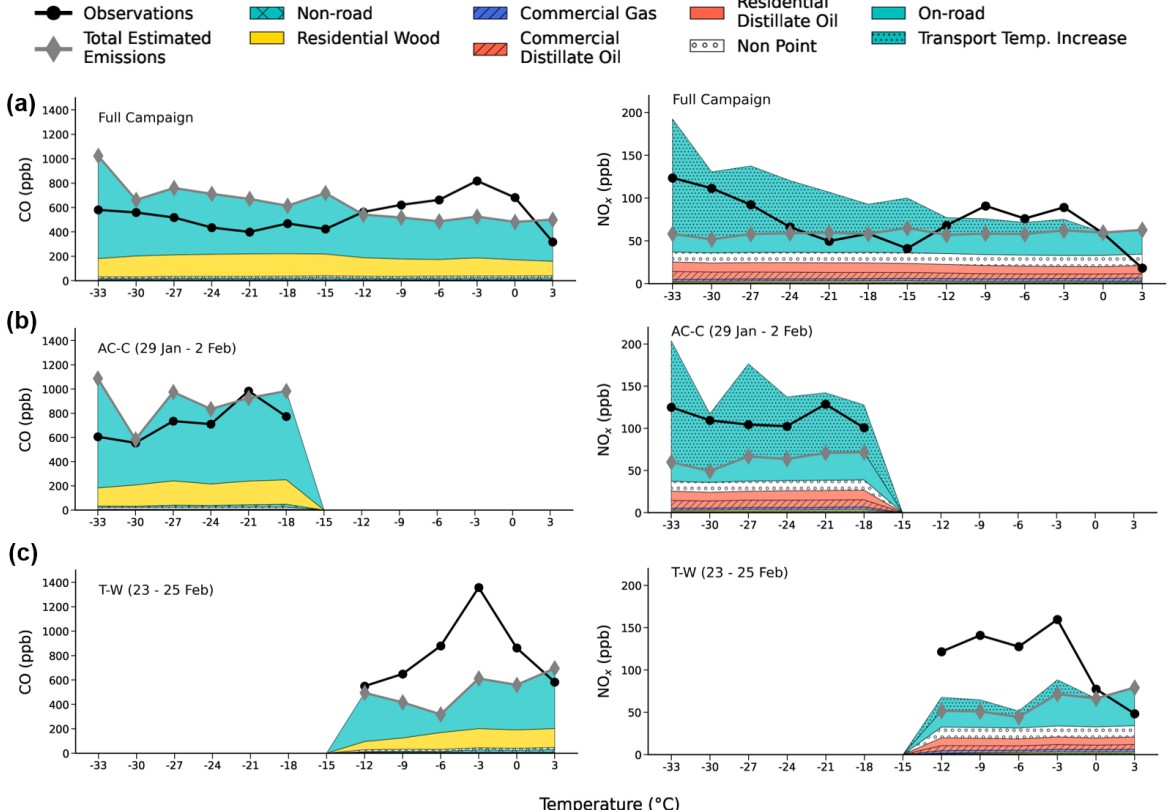

**Figure 10.** EPA surface emission mixing ratios calculated over the Downtown area for contributing emission sectors (grey lines) compared to observations at the CTC site (black lines) in ppb averaged over 3 °C temperature bins for (a) full campaign, b) 29 January to 2 February and c) 23 to 25 February, for CO (left) and $NO_x$ (right). Different emission sectors coloured are shown and given in the legend. The $NO_x$ increment from the linear temperature dependence is shown with '..' hatching. The mid point of the 3 °C temperature bin is shown on the x-axis. The bold black lines correspond to the total surface-emitted emissions listed in the legend with the $NO_x$ vehicle temperature increment. See text for details.

temperature bins over the full campaign, the cold polluted period (AC-C, 29 January to 2 February), and the warm polluted period (T-W, 24 to 25 February). The observations show a clear increase in $NO_x$ at colder temperatures, especially below -23
°C, but is much less distinct for observed CO. For CO, as noted earlier, a cold temperature dependence is already included for mobile (on-road and non-road) gasoline emissions in MOVES3, and there is good agreement between the CO observations and estimated mixing ratios during the cold polluted period (Fig. 10b). The poor agreement between $NO_x$ observations and estimated $NO_x$ supports the hypothesis that an increase in diesel $NO_x$ vehicle emissions due to a cold temperature dependence may be required. Estimated mixing ratios are underestimated in the warm polluted period compared to observations for both
CO and $NO_x$ (Fig. 10c), indicating that a cold-temperature effect is not driving the discrepancy in this period.





Temperature-dependent $NO_x$ emissions are revisited based on a study of diesel vehicles in Norway that found a factor of 3 increase was required at -13 °C with a linear increment from 2.9 to 1.0 between -13 °C to +14 °C, respectively (Wærsted et al., 2022). Here, emission enhancements for total mobile emissions using a log-linear function of between a factor of 1.5 to factor of 10 are calculated for daily average temperatures between 0 °C to -40 °C. For example, the increment is ×3 at -20 °C and a factor of 6 at -30 °C (see Fig. E3). A log-linear function is also used in MOVES3 for the temperature-dependent increase in gasoline emissions (U.S. EPA, 2015). Estimated mixing ratios including the cold temperature dependence are also shown in Fig. 10. Inclusion of this $NO_x$ emission enhancement significantly reduces discrepancies compared with observations during AC-C with very cold temperatures, and biases for the cold and warm polluted events are now comparable to CO (Fig. 10). However, observed $NO_x$ at intermediate temperatures between -22 and -13 °C is now overestimated. This corresponds to temperatures during the Mixed period when surface conditions varied between SS and WS conditions and the discrepancy between observed and estimated CO and $NO_x$ is expected to be influenced more by meteorology and BL stability, as discussed previously.

The log-linear $NO_x$ temperature dependence is applied to modelled mobile emissions tracers in the NOx_Emissions run leading to significant improvements compared to the observations, notably during cold stable conditions e.g. the NME is reduced from 0.8 to 0.5 during T-C. The results suggest an increase in $NO_x$ emissions from diesel vehicles is needed during stable periods with very cold temperatures, notably below -20 °C. The modelled $NO_x$ diurnal cycle also shows a clear improvement during the daytime although differences compared to the observations remain at nighttime. This can be partly explained by difficulties in modelling extremely stable conditions that are enhanced at nighttime. For example, during T-C, there is also an underestimate in CO and $SO_2$ between 0000 and 0600 AKST. However, the large nighttime underestimate in $NO_x$ with respect to CO (e.g. for all data), may indicate an underestimation in $NO_x$ from residential distillate oil emissions (Fig. A1). These emissions dominate at night when mobile emissions are low and warrants investigation in future studies. In event T-W, the bias reduction is small and NMB remains strongly negative at -0.67 because only a small increment is applied to the mobile $NO_x$ emissions at warmer ambient temperatures. The fact that both CO and $NO_x$ are underestimated during this period suggests that these biases are unlikely to be due to the cold temperature dependence, but potentially by uncertainty in the mobile emissions on these days and/or overestimated horizontal transport induced by modelled surface stability as discussed in the previous section.

Discrepancies in modelled $NO_x$ could also be explained by inclusion of atmospheric lifetimes and is explored in Appendix E5 (run NOx_EMissions_LT, shown in Fig. 9 for Mixed period). Notably, inclusion of a shorter atmospheric lifetime during WS conditions improves agreement compared to observations during the Mixed period because $O_3$ transported from aloft leads to titration of NO by reaction with $O_3$ (NME is reduced from 0.93 to 0.7). This has little effect during SS conditions when a longer lifetime is expected due to $O_3$ titration by excess NO and limited $O_3$ production or transport from aloft. Although, assumptions about $NO_x$ lifetimes in this study are simple and a more sophisticated investigation into $NO_x$ chemical processing may be required moving forwards.



### 6.2.3  Sensitivity to SO$_2$ Oxidation

Dry and wet deposition are included in the CTRL SO$_2$ simulation and a photochemical lifetime is not considered because it is too long during Arctic wintertime (Yu et al., 2018; Green et al., 2019) (Appendix E5). Appendix E3 explains the impacts of deposition on SO$_2$. However, SO$_2$ can be oxidised and form secondary sulfate species through other reactions e.g. by oxidation with hydrogen peroxide (H$_2$O$_2$) (Alexander et al., 2012; Moon et al., 2023). Based on isotope observations used in Moon et al. (2023), it is shown that secondary sulfate aerosol formation increased in February (average 44.4 % secondary sulfate) with

respect to January (average 27.5 % secondary sulfate) during ALPACA-2022, consistent with higher observed sulfur oxidation ratio (SOR), an indicator of secondary aerosol formation. Increased secondary sulfate formation in February was due to more WS conditions with higher O$_3$ concentrations at the surface, higher humidity and more clouds, promoting oxidation through aqueous and heterogeneous chemistry.

Here, the SO$_2$ SOR sensitivity explores an effective reduction in SO$_2$ by reducing SO$_2$ emissions using daily SOR values

calculated in Moon et al. (2023) (Table 3). Modelled SO$_2$ overestimates are reduced for the entire campaign, and notably, in late February for the reasons given (NMEs reduced from 0.94 to 0.84 entire campaign, 0.95 to 0.74 during T-W). Remaining overestimates during T-W may be due to residential heating emissions being too high during the warm polluted period. However, since a temperature dependence is already applied in the residential heating emissions, this appears to be unlikely as the controlling factor. Another possible reason could be that SO$_2$ oxidation was enhanced due to the presence of aerosol haze that

occurred during this period. Such pollution haze has previously been shown to promote oxidation of SO$_2$ (e.g., Wang et al., 2014). Overestimate of SO$_2$ may also be influenced by modelled vertical mixing and is explored in the following section.

### 6.2.4  Sensitivity to Vertical Mixing

In a final set of sensitivities, vertical mixing near the surface is explored. The model is run including all previous updates (Table 3). As described earlier, hmin, which is used as a proxy for the height of surface stable layers in this study, is set to 20 m in

CTRL. Thus, tracers, that are emitted from sources at or below 20 m, can be mixed up to this height if the FLEXPART-WRF ABL height is less than this, as depicted in Fig. 5. Since the structure of stable layers in the ABL is complex, sometimes with very shallow SBIs or SSBIs within SBI layers, and difficult for models to reproduce, a sensitivity run is performed with hmin equal to 10 m (MixH_10). However, during the Mixed period (WS conditions), the ABL is less stable with more vertical mixing. To explore this, a sensitivity with hmin equal to 100 m is also performed (run MixH_100). Results from MixH_100

and MixH_10 for selected periods are shown in Fig. 9 with NMBs and NMEs for all runs in Table 4.

CO and NO$_x$ are overestimated compared to the observations during AC-C in the runs with hmin = 10 m, also seen in the diurnal comparisons, leading to poorer agreement compared to CTRL CO and NOx_EMissions_LT (Fig. 9). Results are worse for SO$_2$ due to excessive trapping of space heating emissions below 10 m. In contrast, on-road emissions for NO$_x$ and CO are released only at the near-surface (0-4 m). Negative biases are reduced during T-C in runs using hmin = 10 m, in particular

during the daytime, and also during event T-W but only slightly and NMBs remain high (-0.62 for MixH_10_NOx, -0.45 for MixH_10_CO, Table 4). As discussed in Section 4.3.2, this may be explained by meteorology, although the surface inversion



strength is reproduced quite well by EPA-WRF, simulated horizontal transport is too strong during T-W below 20 m. Runs using hmin = 100 m lead to improvements (reduced positive biases, improved NMEs) during the Mixed period for CO and $NO_x$ due to more vertical dispersion in less stable conditions. Model biases in $SO_2$ are generally improved when hmin = 100 m,

notably in the Mixed period and T-W. The results indicate that the modelled tracers are sensitive to the vertical distribution of emissions, such as those from space heating, as well as the treatment of vertical dispersion and turbulence in FLEXPART-WRF.

In general, these results indicate that runs using a mixing height of 20 m (CTRL) performs well for all tracers under stable conditions but, during periods with strong SBI or SSBIs, such as during the T-C and T-W events, vertical mixing of surface-based emission tracers (i.e. CO and $NO_x$), is even more suppressed and runs with hmin = 10 m perform better. On the other

hand, runs with MixH_100 improve simulated tracer concentrations during WS conditions with enhanced vertical transport. $SO_2$ is more complex since space heating emissions are also emitted above the surface which can be mixed down to the surface or aloft. Overall, this suggests that improvements are needed to the treatment of vertical mixing in FLEXPART-WRF during wintertime Arctic conditions.

## 7 Conclusions and Future Perspectives

This study presents a detailed investigation of processes influencing wintertime pollution from surface urban and elevated point sources in Fairbanks, a sub-Arctic city in Alaska, exploiting Larangian particle dispersion modelling and comprehensive surface and vertical profile measurements made during the ALPACA campaign in January-February 2022 (Simpson et al., 2024). To evaluate the dispersion and vertical distribution of different pollution sources in the Fairbanks area, high temporal and spatial resolution surface and power plant emission tracers of CO, $SO_2$, NO and $NO_2$ have been included in the FLEXPART-WRF

model. To account for the presence of stable layers, at the surface and aloft, a scheme for estimating power plant emission injection heights in FLEXPART-WRF was implemented using detailed information about the power plant stack emissions, building on the previous work of Briggs (1984) in stable conditions. Comparison of simulated tracer distributions with observations, and sensitivities to switching off power plant plume rise and plume capping by stable layers, show that accounting for plume buoyancy and capping emission injection are critically important for accurate simulation of power plant plume injection

heights and their transport downwind. In particular, the use of detailed stack parameters (stack height and radius, flue gas exit temperature and velocity), and temperature profile measurements to diagnose the presence of inversions that trap pollution plumes, are required.

Model results were evaluated depending on different meteorological conditions. Notably, analysis of surface temperature gradients identified strongly stable (SS) and weakly stable (WS) conditions close to the surface, following Maillard et al.

(2022) and Simpson et al. (2024). Simulated trace gas concentrations, which are enhancements above background, emitted from surface and elevated sources, including the power plants, are larger during SS compared to WS conditions over the Fairbanks area. Vertical transport is more limited in SS conditions and by the presence of elevated inversion layers. Near surface pollution is reduced and pollution concentrations above 200 m are enhanced during WS conditions owing to stronger horizontal and vertical transport, likely due to enhanced turbulent mixing. Pollution outflow to the south-west, due to dominating north-



easterly winds up to 200 m suggests a possible regional influence from anthropogenic emissions over the wider Fairbanks area, including North Pole which requires further investigation. Modelled tracer concentrations are larger than those typically found in wintertime Arctic haze.

Pollution plumes observed by the Helikite aloft are generally well simulated in terms of timing and vertical distributions. These plumes are attributed to particular power plant stacks following transport by north-easterly or easterly winds to the
UAF Farm site in the west of Fairbanks. In some cases, small discrepancies in EPA-WRF winds, used to drive the tracer simulations, results in displacement of the plumes, for example to the south of the measurement location. The plume rise calculations could be improved further by using WRF temperatures and winds at the location of the power plant stacks, rather than using radiosondes at Fairbanks airport, allowing spatial differences to be better captured. Acquisition of more vertical profile observations (e.g. using drones) at, and downwind of, the power plant stacks would also be valuable.

At the surface, modelled CO compares well to observations in downtown Fairbanks with variability driven by changes in surface stability. Discrepancies are mostly explained by differences in modelled meteorology or ABL stability on short time-scales. Agreement at other sites is less good. At the Hamilton Acres site in the eastern residential area, model discrepancies could be explained by the horizontal resolution of the emissions (1.33 km) being too coarse to capture the larger residential wood burning emissions at this site. Surface pollution is reduced at the UAF Farm in western Fairbanks, a site also influenced
by a local valley flow, that frequently establishes during anticyclonic conditions, induces turbulence and clears out surface pollution. This flow is underestimated by EPA-WRF in situations when strong surface stability is observed, and thus in the tracer simulations. This is due to misrepresentation of dynamic instability (turbulence/shear) induced by the local flow in the WRF simulations. These results highlight the complexities of dispersion modelling in a region influenced by strongly stable ABL conditions and local-scale phenomena linked to orography. Improvements to WRF simulations based on Maillard et al.
(2022) who examined surface effects of the local valley flow at the UAF Farm site, or using higher resolution model simulations, such as Large Eddy Simulations, may also improve results.

In contrast to CO, surface $NO_x$ is significantly underestimated in the CTRL simulation, notably in very cold, stable conditions. A possible cause is underestimation of $NO_x$ emissions from diesel vehicles, already shown to be important down to -13 °C (e.g. Wærsted et al. (2022)). Inclusion of a log-linear temperature dependence for $NO_x$ emissions from the mobile (on-road
and non-road) sector, by a factor of x1.5 at 0°C to x6 at -30 °C (average daily temperatures), considerably improves the model results (during daytime). Previous studies have not considered such large increases at very low temperatures below -15 °C and warrants further investigation. Such dependencies may be due to inefficient, or even failure of, selective catalytic reduction units implemented in vehicles to reduce $NO_x$ emissions (Seo et al., 2022), and should be considered in emission inventories in cold wintertime environments similar to Fairbanks. Inclusion of photochemical lifetimes for NO and $NO_2$ also improves
simulated surface $NO_x$ notably during WS conditions, when $O_3$ concentrations are higher. Future work investigating chemical processing of $NO_x$ and $O_3$ at the surface, and in power plant plumes, will help to better constrain $NO_x$ lifetimes in the polluted Arctic wintertime.

Surface $SO_2$ is generally overestimated, despite inclusion of simplified treatments of wet and dry deposition and an estimation of the fraction of $SO_2$ converted to secondary sulfate species. Discrepancies appear to be mostly driven by the vertical



transport of space heating emissions which are distributed between 5 and 12 m in the EPA-ALPACA emission inventory. This is explored by varying the minimum mixing height (hmin) in FLEXPART-WRF which, in this study, influences the altitude to which surface tracers are mixed vertically. Increasing hmin from 20 m to 100 m improves the comparison to observed $SO_2$ at the surface due to enhanced vertical transport of the space heating emissions. In contrast, the on-road mobile sector dominates surface emissions of CO and $NO_x$ in central Fairbanks and they are often trapped near to the surface by very shallow SBI or

SSBIs. For these tracers, runs with hmin equal to 10 m limits vertical mixing and leads to further improvement in the model results compared to surface observations. Model sensitivity to the hmin parameter suggests that improvements are needed to the treatment of turbulent mixing in wintertime conditions with very stable boundary layers.

Overall, the findings of this study illustrate the complexity of simulating surface and elevated pollution sources in cold stable Arctic wintertime conditions. The tracer simulations, while simplified in some aspects, provide important insights into possible

processes affecting trace gas pollution at the surface and aloft in the boundary layer. They form a basis for regional 3D chemical and aerosol modelling of pollution due to anthropogenic emissions over the Fairbanks region and its potential contribution to background Arctic haze during winter-spring. As the Arctic becomes more developed in the future, due to increasing human activity and climate warming, higher energy demands in Arctic communities are expected. This may lead to increases in poor air quality during Arctic winter, notably if poor energy infrastructure persists. More stringent emissions standards for surface

and elevated sources, and an accelerated transition towards renewable energies needs to be considered at the policy level in the Arctic moving forwards.

*Data availability.* Final data from the study will be available to the scientific community two years after the conclusion of the study. Arctic-data.io (https://arcticdata.io/catalog/portals/ALPACA) provides a portal to archival repositories of the field study's data.



## Appendix A: Emissions

In this section, additional figures related to the emissions used in this study are provided. Appendix A1 provides details on emission controls used for the power plant stacks running during ALPACA-2022. Figure A1 shows surface emissions for each sector averaged over the non-attainment areas, Downtown, Hamilton Acres (HA) and UAF Farm areas. In the Fairbanks non-attainment area, airport emissions are large for $SO_2$. The emission contributions at the East Residential and Downtown sites are comparable but magnitudes are greater Downtown (Fig. A1c,d). The UAF Farm site is dominated by the mobile sector but

the magnitude of emissions are small compared to the other locations. Figure A2 shows a time series of power plant emission data for each power plant stack for the trace gases (CO, $SO_2$, NO and $NO_2$). Differences in trace gas emissions according to fuel type are evident in Fig. A2. For example, North Pole A emits large $NO_2$ emissions due to running on Naphtha fuel, while $NO_x$ and $SO_2$ emissions from UAF C are small compared to the other coal-fired stacks (Aurora and Doyon) owing to more stringent emission controls (Appendix A1).

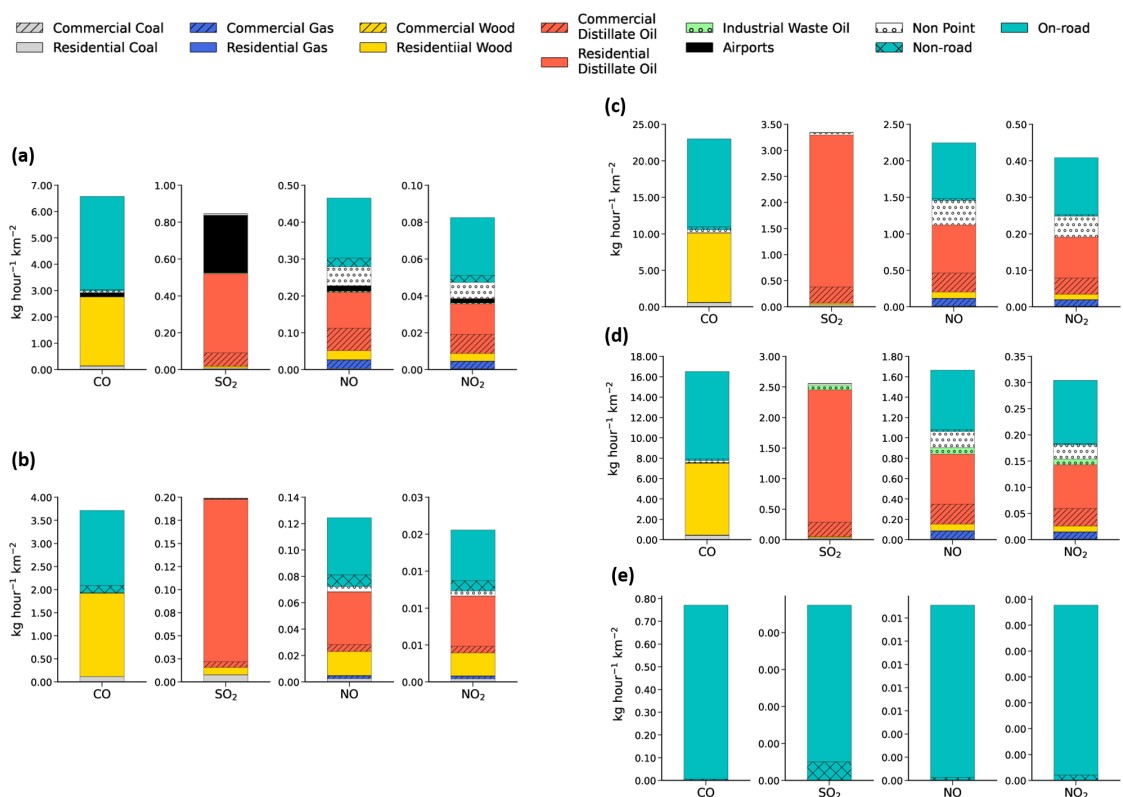

**Figure A1.** CO, $SO_2$, NO and $NO_2$ emissions (kg hour$^{-1}$ km$^{-2}$) averaged over the campaign, summed between 0-18 m altitude, and averaged per 1.33 km grid cell for a) Fairbanks Non-attainment area, b) North Pole Non-attainment area, c) Downtown, d) Hamilton Acres and e) UAF Farm. Panels c to e are for the EPA emissions grid cell (1.33 km) closest to the location. See Fig. 1 for details.





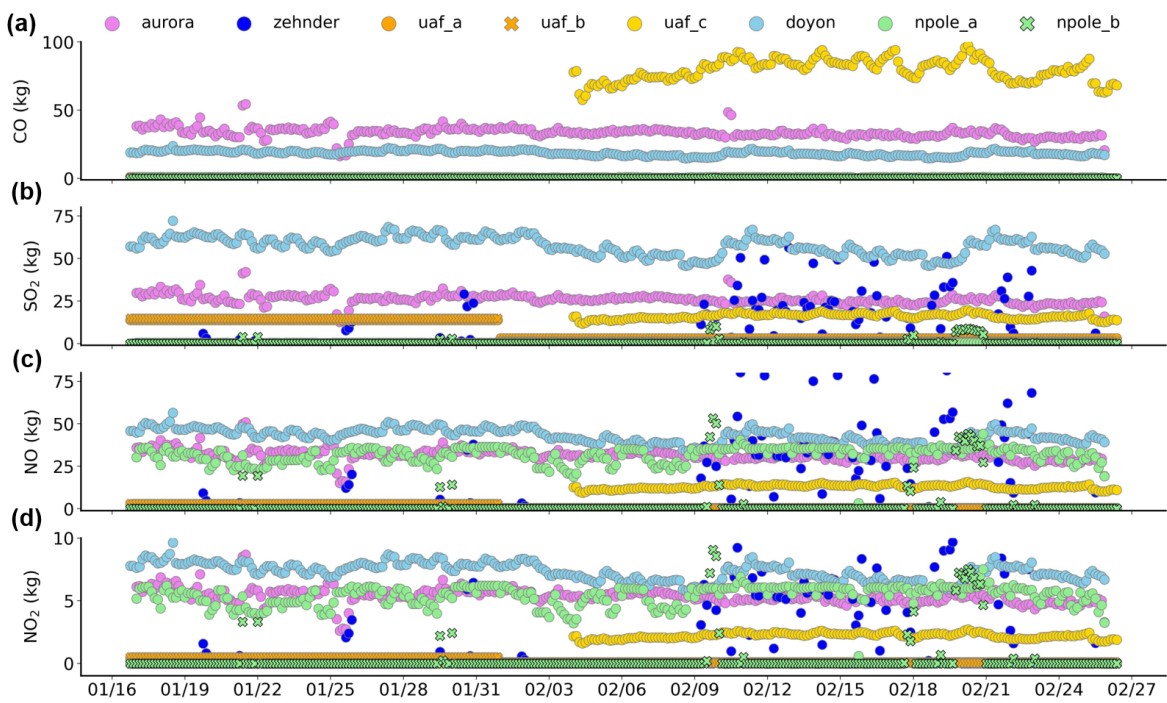

**Figure A2.** Emissions of a) CO, b) $SO_2$, c) NO and d) $NO_2$ (kg) for each power plant stack (indicated above panel A) as a function of time during ALPACA-2022. 3-hour averages are shown for clarity of the data points, but 1-hour data is used in the modelling study.

## A1  Power Plant Emissions Control Strategies

The power plant emissions used in this study were provided by each of the power plant facilities for the campaign period. The emissions vary depending on fuel type and emission reduction controls (ADEC, 2019b). The UAF C coal stack uses low $NO_x$ burners (40-60% efficiency) and staged combustion to reduce $NO_x$ emissions. However, Aurora and Doyon (also coal) do not have $NO_x$ emission controls such as selective catalyst reduction (SCR) units. Diesel or fuel oil power plants (Zehnder, UAF A, B and North Pole A) do not have $NO_x$ emission reduction strategies, while North Pole A uses water injection (70% efficiency in $NO_x$ reduction). $SO_2$ control strategies include 0.25% sulfur by weight for each coal power plant and UAF C also uses limestone injection. North Pole A uses 20 parts per million (ppm) sulfur, while UAF A, B and North Pole B are limited to 15 ppm sulfur. The limit is as high as 1000 ppm sulfur for Zehnder but operations are limited to < 70 tons per year. Zehnder and North Pole B stacks ran intermittently (non-continuous) during the campaign, and more frequently in February than January (Fig. A2), due to having 'limited operation' controls. ADEC (2019b) provides more information on control strategies for each of the power plant facilities.

 

## Appendix B: Evaluation of EPA-WRF model results against meteorological observations

Figures B1 and B2 compares modelled and observed meteorology (temperatures, wind speeds and directions) as a function of time for available altitudes up to 23 m, at the Downtown and UAF Farm sites, respectively. The general variability is well
captured by EPA-WRF, notably Downtown. For temperature gradients (23-3 m) Downtown, however there are some days in which high temperature gradients are underestimated (e.g. 2 February) or overestimated (e.g. 24 January). Wind direction agreement is poor when wind speeds are very low, but this is expected owing to higher uncertainties at low wind speeds (Fig. B1f, B2d). At the UAF Farm, the very high observed temperature gradients during AC-C and T-C are underestimated by EPA-WRF. Wind speeds at the UAF Farm are in poorer agreement when temperature gradients are high, notably during AC-C
and T-C. Effectively modelling local flows, as experienced at the UAF Farm, is challenging (see discussion in Section 6.1). Discrepancies in winds and temperatures may contribute to differences between the FLEXPART-WRF tracer concentrations and observations. This is considered in the main text.

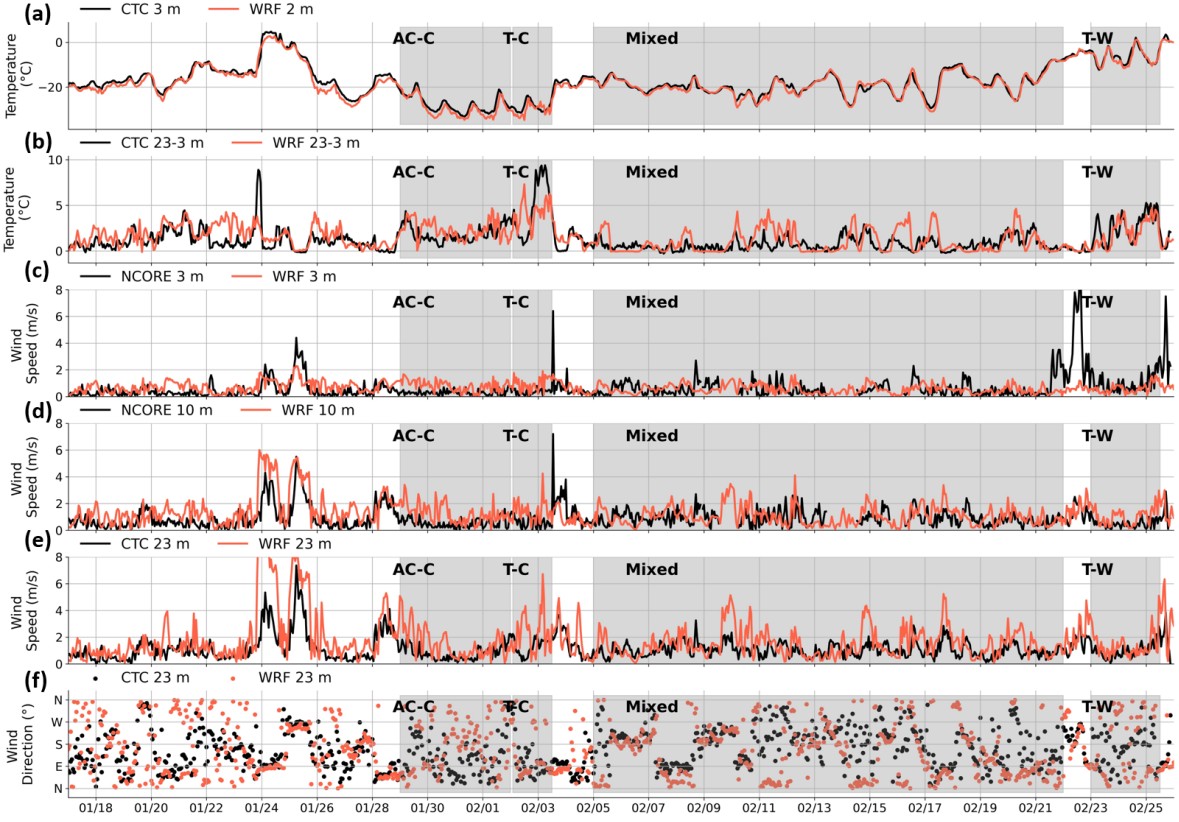

**Figure B1.** Time series of surface and near-surface temperatures (°C), temperature gradients (dT) (°C), wind speeds ($\mathrm{m}s^{-1}$) and directions (degs.), at Downtown, compared to EPA-WRF (red) for available observations (black) up to 23 m altitude during ALPACA-2022.



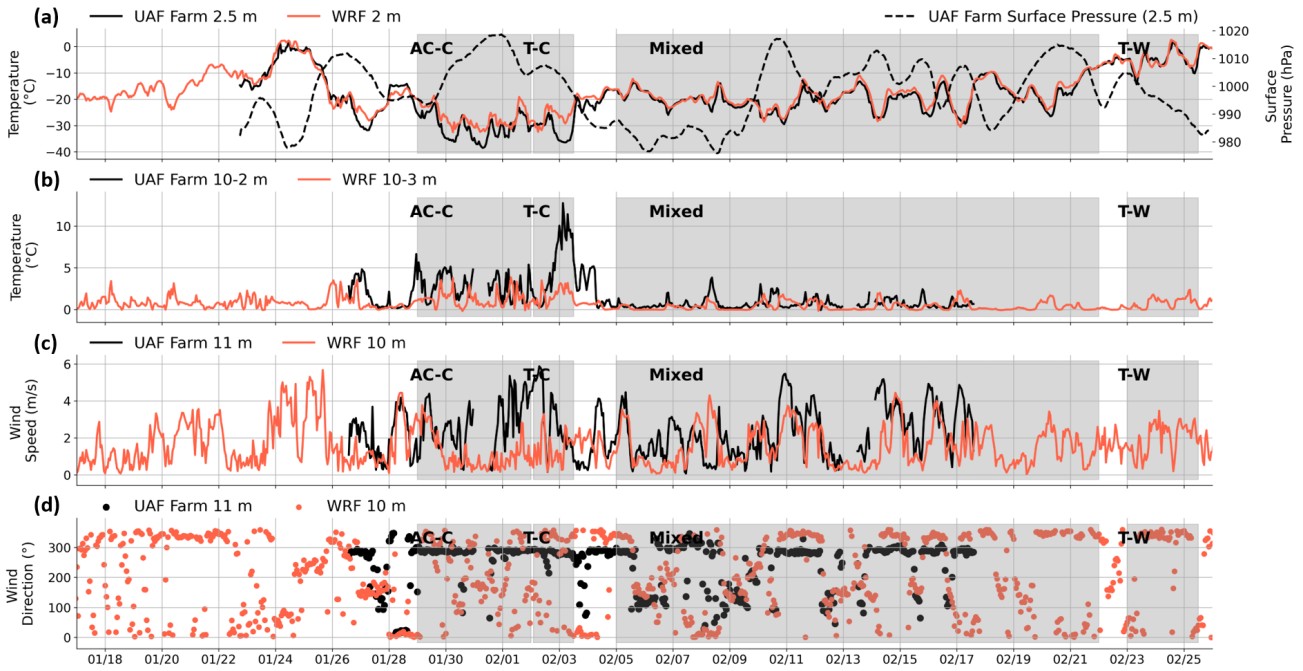

**Figure B2.** Time series of surface and near-surface temperatures (°C), temperature gradients (dT) (°C), wind speeds (m$s^{-1}$) and directions (degs.), at UAF Farm, compared to EPA-WRF (red) for available observations (black) up to 11 m altitude during ALPACA-2022.

## Appendix C: Vertical and Horizontal Dispersion of Emission Tracers

Figures C1 and C2 show spatial distributions in total (surface-emitted plus power plant) tracer enhancements at various altitude
levels for $SO_2$, CO and $NO_x$. In Fig. C1, above 50 m, power plant influences are less evident for CO than for $SO_2$ and $NO_x$, with respect to the total tracer because CO emissions from power plants are smaller relative to surface-emitted tracers. For example, there is an order of magnitude difference for surface-emitted and power plant modelled CO tracers, that is not seen for $SO_2$ and $NO_x$ (depicted in Fig. C3b, see the following). At 0-10 m, $NO_x$ spatial variability is comparable to that of CO due to similarities in emission sources (see also Fig. A1). $NO_x$ mixing ratios are larger from 0-100 m in SS compared to WS
conditions, while above 100 m, mixing ratios are larger in WS conditions and influences from power plants are evident (Fig. C2). The North Pole power plant stacks have larger influences for $NO_x$ than CO and $SO_2$ due to differences in fuel types. Fig. C3 shows simulated power plant (i) and surface-emitted (ii) tracers as a function of altitude and time for $SO_2$ at Hamilton Acres and CO at the UAF Farm. Power plant contributions vary at HA and the UAF Farm, indicating influences from different power plants at each location. Vertical transport is stronger at the UAF Farm site, notably for the surface-emitted tracers (Fig. C3b).



**Figure C1.** Spatial maps of total power plants and surface-emitted tracer enhancements (CTRL simulation) for i) SS and ii) WS for conditions a) SO$_2$ (ppb) at 100 - 200 m, b) CO at 50 - 100 m and c) CO at 100-200 m and 200 - 300m (iii, iv). Wind vectors (black arrows) indicate average wind direction (degs.) and wind speeds (m$s^{-1}$) simulated by EPA-WRF. [PLACEHOLDER: MAY REMOVE]





**Figure C2.** Spatial maps of total power plants and surface-emitted tracer enhancements (CTRL simulation) for i) SS and ii) WS conditions for CO (ppb) at a) 0-10 m (top), 50-100 m (lower) and b) 100 - 200 m (top) and 200 - 300 m (lower). Wind vectors (black arrows) indicate average wind direction (degs.) and wind speeds (m$s^{-1}$) simulated by EPA-WRF.



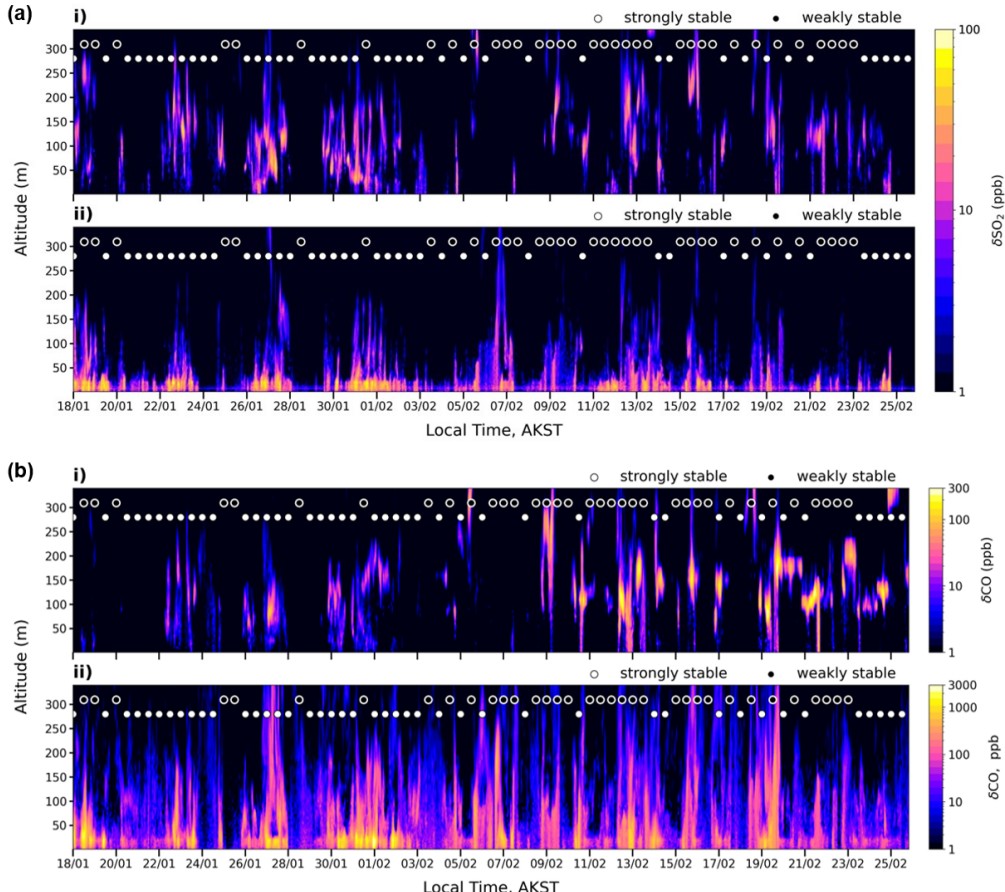

**Figure C3.** a) Total power plants (i) and surface-emitted (ii) tracer enhancements (CTRL simulation) as a function of altitude and time for a) $SO_2$ (ppb) at Hamilton Acres and b) CO (ppb) at the UAF Farm. The WS and SS stability regimes indicated every 12 hours.

**Appendix D: Simulated Vertical Distributions and Power Plant Plumes**

Vertical distribution of CO and $NO_x$ power plant tracers as a function of time at the UAF Farm (1 grid cell) of $\delta CO$ and $\delta NO_x$ are shown for NO-RISE and NO-CAP simulations in Figure D1. The altitude and concentration of the model simulated tracers compared to observed plumes is significantly improved in NOCAP than NORISE, highlighting the importance of accounting for plume buoyancy. CAP (Fig. 6, main text) vs NOCAP differences are evaluated in the main text in more detail for individual cases. In some cases (e.g. CASE 3, main text and CASE 4, D1), the observed plume is not simulated at the grid cell closest to the UAF Farm due to displacement induced by wind direction discrepancies (see Appendix D1).



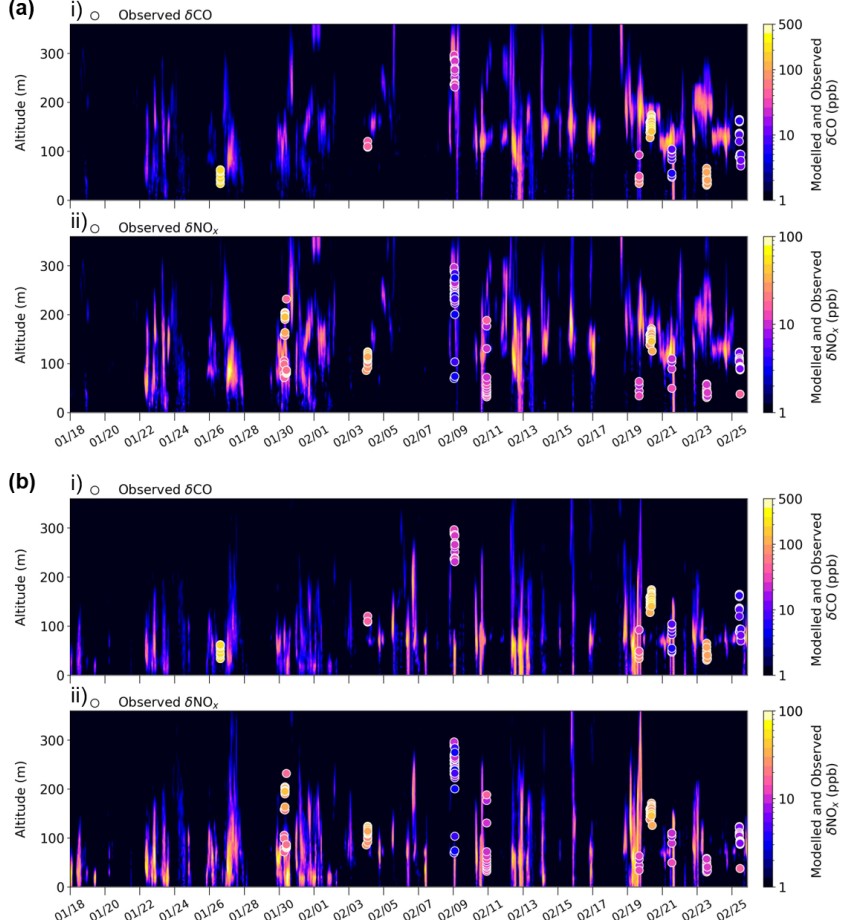

**Figure D1.** Comparison of modelled (CTRL) power plant and observed trace gas enhancements above background (> 30 m) fora) NOCAP and b) NORISE and i) $\delta$CO (ppb) and ii) $\delta$NO$_x$ (ppb) at the closest grid-cell to the UAF Farm. Periods with available observations are shown. Observed plume enhancements are shown as circles (ppb).

## D1  Power plant plume model displacement

Figure D2a shows the spatial distribution of power plant tracers during case 3 for CO and NO$_x$ to support the discussion in section 5. Here, the larger modelled CO enhancements are supported by the influence of the UAF C stack in close proximity to the UAF Farm and a displacement of the plume from the Aurora and Zehnder stacks in the east of Fairbanks, with larger NO$_x$ concentrations. Figure D2b shows results for an additional case study (Case 4) on 3-4 February which supports displacement of power plant tracers due to EPA-WRF model against observation discrepancies. In Fig. D2b, there is a large underestimate in NO$_x$ compared to observations, averaged over altitude bins. This can be explained by a discrepancy in the modelled wind direction (model: north-east, observed: east) leading to displacement of the modelled plume, as depicted in Fig. D2c. A sim-




ulated plume is transported from the Doyon stack and south of the UAF Farm site between 120 - 180 m at 0200 AKST. The UAF C stack was not running during this flight (operations started from 0900 AKST on 4 February, Section 2.1.1).

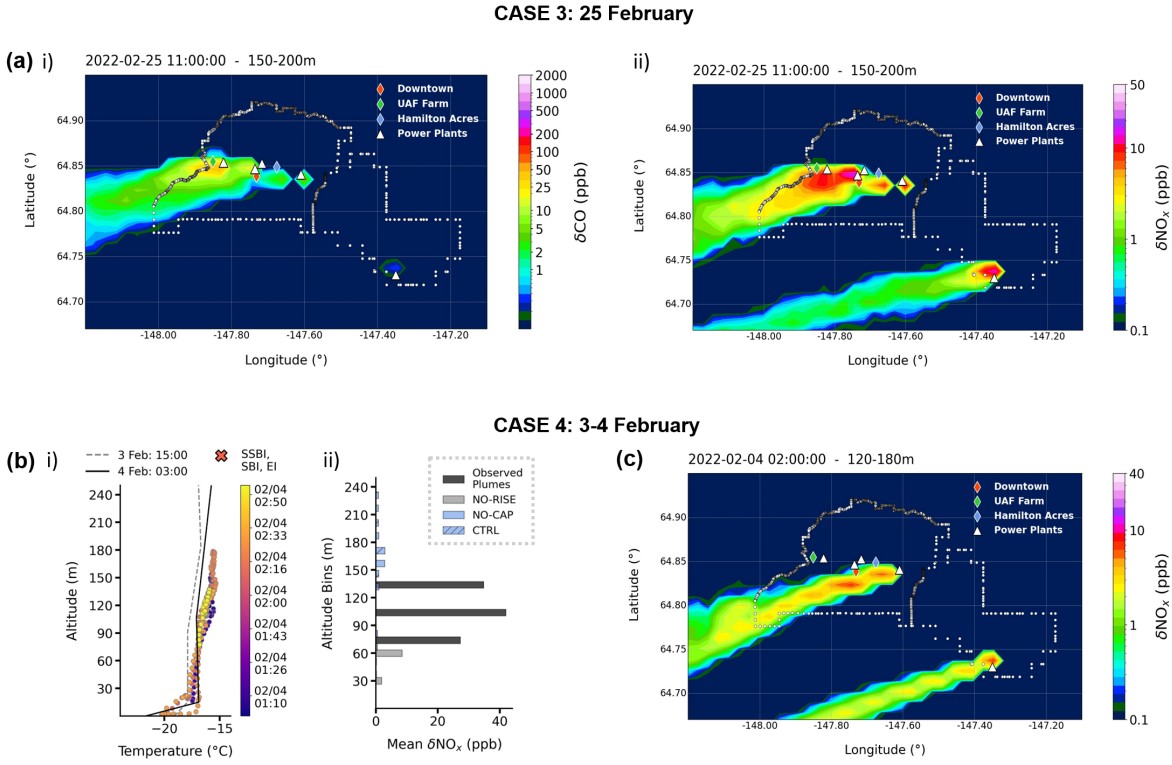

**Figure D2.** a) Spatial distribution of power plant tracers for i) CO (ppb) and ii) $NO_x$ (ppb) between 150-200 m at 1100 AKST on 25 February (Case 3, Section 5), highlighting the Aurora plume displacement to the south of the UAF Farm. b) Observed temperature profiles and modelled $NO_x$ tracer (ppb) enhancements against observations averaged over 30 m altitude bins for Case 4 on 3-4 February (see Fig. 7 for details). c) Spatial distribution of power plant $NO_x$ tracer (ppb) between (120 - 180 m) at 0200 AKST on 4 February, highlighting the Doyon plume displacement to the south of the UAF Farm.

## D2   Model evaluation against Wind LiDAR observations

Figure D3 shows modelled power plant $NO_x$ tracer enhancements compared to wind LiDAR observations measured at the CTC site (CASE 1 on 30 January to 1 February) and the UAF Farm (CASE 2 on 8 to 9 February). The wind LiDAR measures the

3 wind components using 5 beams (1 vertical and 4 slanted) of infrared light to record the attenuated backscatter of particles in the air (aerosol and and water droplets) between 40 m and 290 m (20 m depth layers), referred to here as the RCS (range corrected signal) (Fig. D2, panel iii)) (Dieudonné et al., 2023). The quality of the signal to noise ratio depends on the presence of particles in the atmosphere, i.e. higher pollution or precipitation (snow) produces a stronger signal. A wind LiDAR plume





mask has been developed to distinguish between water droplets and aerosols based on RCS and is used here to explore the

presence of elevated plumes containing aerosols compared to the model results. This comparison is qualitative since the model

simulations are enhancements in trace gas mixing ratios above background and wind lidar RCS provides an indication about

the presence of particles between than 0.5 - 1 μm with a peak of sensitivity around 0.7 - 0.8 μm (Dieudonné et al., 2019),

which could be either primary or secondary aerosols.

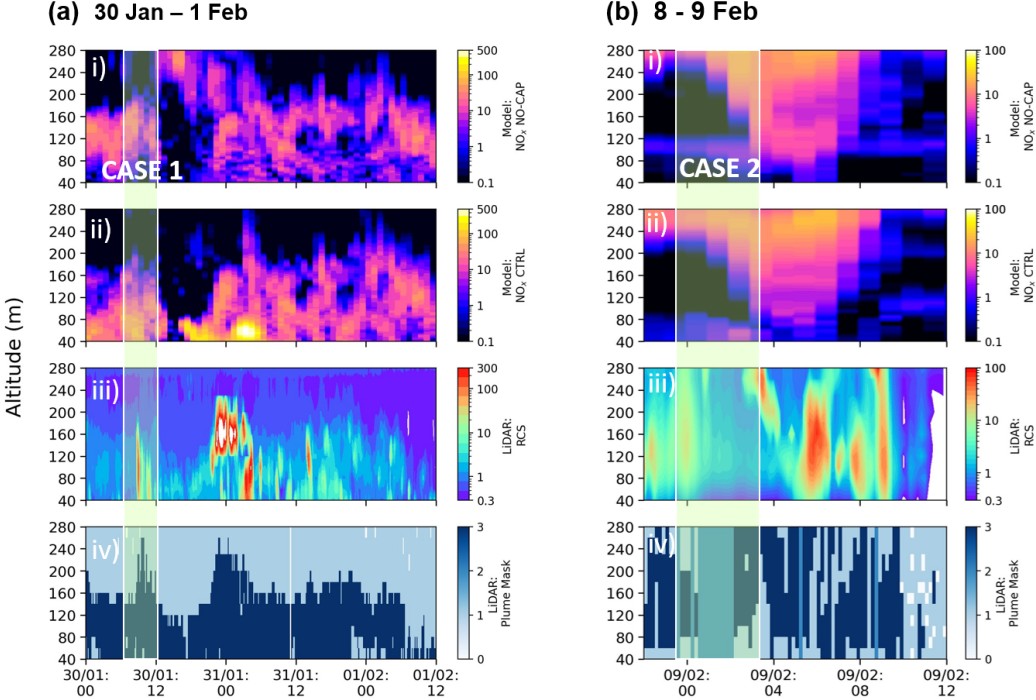

**Figure D3.** i, ii) Simulated power plant NO$_x$ tracers (ppb) as a function of altitude and time for NOCAP and CTRL simulations, respectively, iii) aerosol back-scatter coefficient observed by wind LiDAR observations and iv) LiDAR-plume mask (described in Appendix D), for a) 30 January to 1 February at Downtown (CASE 1 highlighted by white box) and b) 8 to 9 February at the UAF Farm (CASE 2 highlighted by white box).

## Appendix E:  Evaluation of modelled surface tracers

### E1  Statistical Metrics


To evaluate model performance Normalised Mean Biases (NMBs) and Normalised Mean Errors (NMEs) are calculated using the following equations :

$$NMB = \frac{\sum_{i=1}^{n}(M_i - X_i)}{\sum_{i=1}^{n} X_i} \tag{E1}$$



$$NME = \frac{\sum_{i=1}^{n} |M_i - X_i|}{\sum_{i=1}^{n} X_i} \qquad\qquad (E2)$$

**E2   UAF Farm and Hamilton Acres**

NMBs and NMEs for model performance compared to surface observations are provided for Downtown (Table 4) and here for the HA and UAF Farm sites (Tables E1 and E2). Figures E1 and E2 show the time series and diurnal cycles for the different meteorological regimes as described in Section 6.1 for HA and the UAF Farm, respectively.

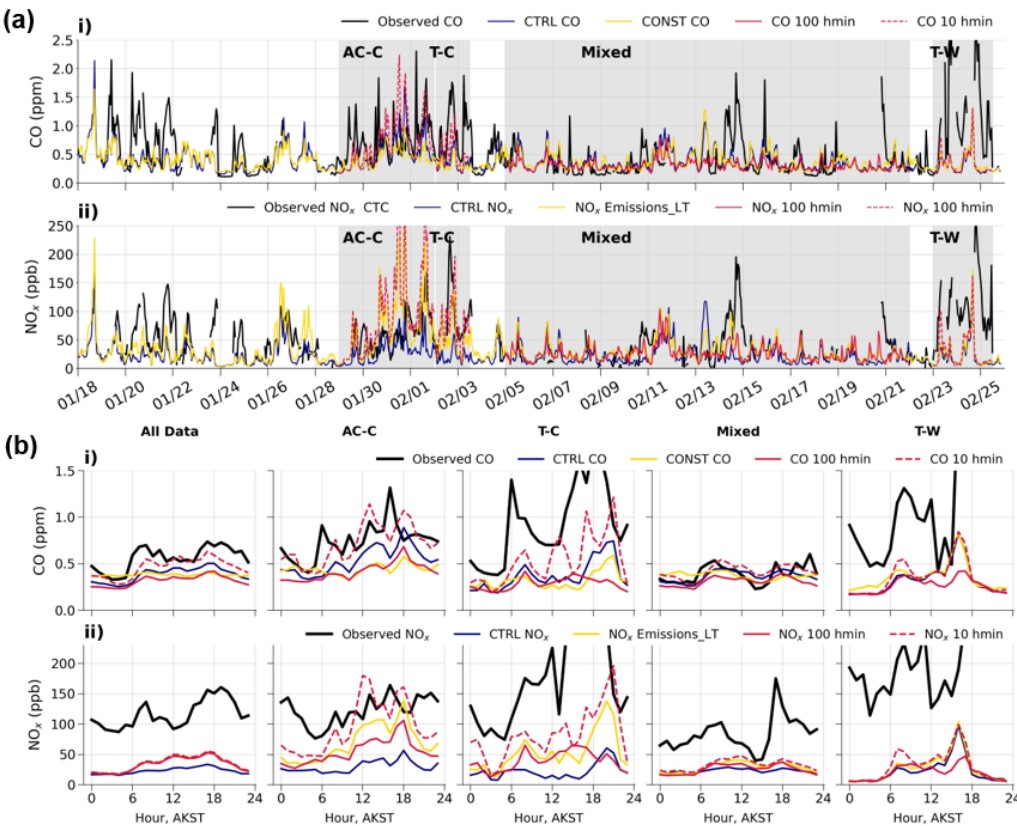

**Figure E1.** a) Total modelled surface and power plant tracers as a function of time between 0-5 m for CTRL and selected sensitivity simulations described in Table 2 compared to available surface observations, at HA for i) CO and ii) NO$_x$. b) Average diurnal cycles at HA for observations (black) and model simulations (colours as in (a)) for i) CO and ii) NO$_x$, averaged over all data and for events A-C, T-C, Mixed and T-W.




| Simulation Name | NMB | | | | | NME | | | | |
|---|---|---|---|---|---|---|---|---|---|---|
| | All Data | AC-C | T-C | Mixed | T-W | All Data | AC-C | T-C | Mixed | T-W |
| CTRL CO | -0.34 | -0.27 | -0.59 | -0.06 | -0.78 | 0.56 | 0.44 | 0.6 | 0.55 | 0.78 |
| CONST CO | -0.35 | -0.4 | -0.64 | -0.05 | -0.74 | 0.56 | 0.47 | 0.64 | 0.54 | 0.74 |
| MixH_100_CO | -0.44 | -0.43 | -0.68 | -0.21 | -0.78 | 0.57 | 0.49 | 0.68 | 0.49 | 0.78 |
| MixH_10_CO | -0.21 | -0.03 | -0.42 | 0.06 | -0.74 | 0.55 | 0.46 | 0.5 | 0.61 | 0.75 |
| CTRL NOx | -0.79 | -0.73 | -0.88 | -0.69 | -0.91 | 0.82 | 0.73 | 0.88 | 0.82 | 0.91 |
| NOx Emissions | -0.58 | -0.27 | -0.71 | -0.44 | -0.89 | 0.72 | 0.51 | 0.72 | 0.77 | 0.89 |
| NOx Emissions_LT | -0.65 | -0.37 | -0.72 | -0.6 | -0.9 | 0.73 | 0.53 | 0.73 | 0.73 | 0.9 |
| MixH_100_NOx | -0.66 | -0.51 | -0.81 | -0.67 | -0.91 | 0.73 | 0.59 | 0.81 | 0.75 | 0.91 |
| MixH_10_NOx | -0.64 | -0.13 | -0.57 | -0.57 | -0.87 | 0.72 | 0.48 | 0.6 | 0.72 | 0.87 |

**Table E1.** Comparison of Normalised mean biases (NMBs) and normalised mean errors (NMEs) of model simulations (total tracers) between 0-5 m, compared to surface observations at Hamilton Acres for CO and NO$_x$ at hourly time resolution. NMB and NMEs are given for all data and the meteorological events A-C, T-C, Mixed and T-W.

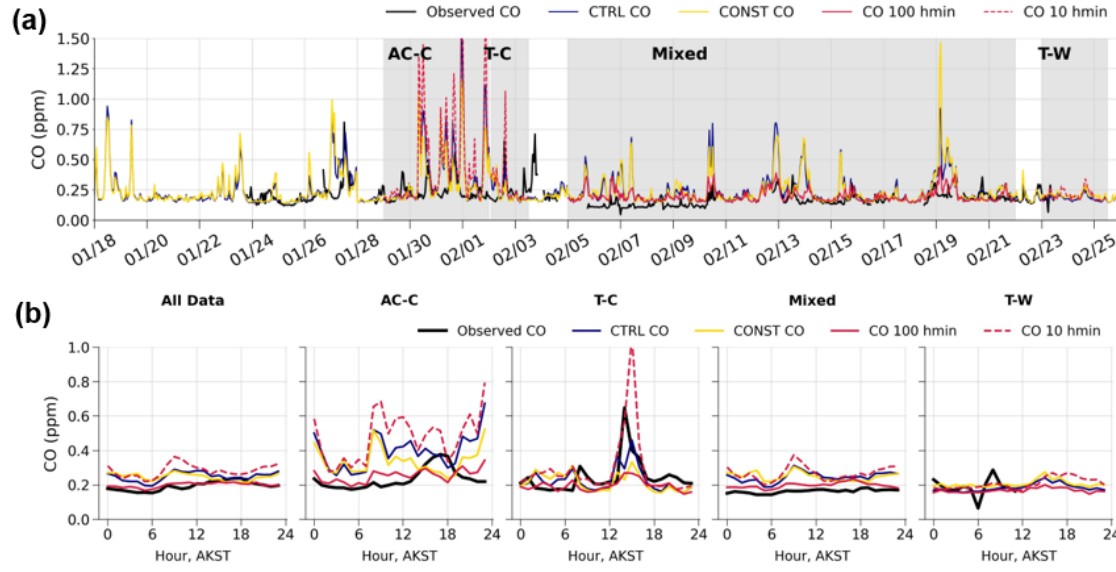

**Figure E2.** a) Total modelled surface and power plant tracers as a function of time between 0-5 m for CTRL and selected sensitivity simulations described in Table 2 compared to available surface observations, at UAF Farm for CO. b) Average diurnal cycles at UAF Farm for observations (black) and model simulations (colours as in (a)) for CO averaged over all data and for events A-C, T-C, Mixed and T-W.





| Simulation Name | NMB | | | | | NME | | | | |
|---|---|---|---|---|---|---|---|---|---|---|
| | All Data | AC-C | T-C | Mixed | T-W | All Data | AC-C | T-C | Mixed | T-W |
| CTRL CO | 0.39 | 0.68 | -0.04 | 0.58 | -0.06 | 0.56 | 0.82 | 0.33 | 0.64 | 0.18 |
| CONST CO | 0.37 | 0.44 | -0.06 | 0.61 | 0.09 | 0.56 | 0.66 | 0.38 | 0.67 | 0.19 |
| MixH_100_CO | 0.07 | 0.08 | -0.17 | 0.22 | -0.09 | 0.31 | 0.34 | 0.29 | 0.33 | 0.19 |
| MixH_10_CO | 0.6 | 1.04 | 0.19 | 0.71 | 0.04 | 0.73 | 1.14 | 0.43 | 0.76 | 0.17 |

**Table E2.** Comparison of Normalised mean biases (NMBs) and normalised mean errors (NMEs) of model simulations (total tracers) between 0-5 m, compared to surface observations at UAF Farm for CO at hourly time resolution. NMB and NMEs are given for all data and the meteorological events A-C, T-C, Mixed and T-W.

### E3  SO$_2$ Wet and Dry Deposition

Dry or wet deposition were explored initially and included in the CTRL simulation for SO$_2$. With regard to dry deposition of SO$_2$, fluxes are expected to be low in Arctic winter due to lower temperatures, less exposure to moist surfaces and low levels of oxidants, as found in the Athabasca Oil Sands region in Canada (Hsu et al., 2016). However, Hsu et al. (2016) recorded higher deposition fluxes close to emission sources such as power plants. In this study, a dry deposition velocity of 0.1 cm$s^{-1}$ is used, based on values recorded over snow in the wintertime Arctic, ranging from 0.06 and 0.082 cm$s^{-1}$ (Dasch and Cadle,
1986; Valdez et al., 1987) and 0.2 cm$s^{-1}$ in northern Canada between February and March (Barrie and Walmsley, 1978). A simplified treatment for wet deposition of SO$_2$ is also included in the CTRL run using FLEXPART-WRF. A wet deposition velocity, or scavenging coefficient is prescribed $1 \times 10^{-4}$, together with a Henry's law constant of $3 \times 10^{-4}$, based on values used in other studies (e.g., Valdez et al., 1987; Choi et al., 2000; Elperin et al., 2013). In general most of the precipitation occurred during the Mixed period in February and during WS conditions. Biases were reduced when deposition is considered
but had minor effect compared to the other sensitivities in this study, hence deposition was included in the CTRL simulation.

### E4  Sensitivity to Vehicle NO$_x$ Emissions at Cold Temperatures

MOVES3 (U.S. EPA, 2021) includes an incremental temperature dependence with higher emissions at colder temperatures for gasoline and diesel vehicles based on MOVES2014b (U.S. EPA, 2015). Updates in MOVES3 compared to MOVES2014b include reduced NO$_x$ emissions due to the diesel fleet turnover but not to the temperature adjustments for the trace gas species in
this study. For start energy combustion emissions (from engine fuel ignition), there is a higher increment for gasoline emissions at colder temperatures (up to a factor of 4.8 at -30 °C) than diesel (up to a factor of 2.7 at -30 °C). A multiplicative adjustment using a log-linear fit based on ambient temperatures is applied to CO gasoline emissions. However, for NO$_x$ gasoline emissions, at -18 °C a 1.227 additive temperature adjustment is reduced only to 1.201 at -30 °C, so that the adjustment does not exceed 1.2 at colder temperatures. Since NO$_x$ emissions from gasoline vehicles are much lower than for CO, this results in a much
higher increment for CO emissions at cold temperatures. In addition, for diesel vehicle cold starts, no statistical relationship was found for both NO$_x$ and CO, and the temperature adjustments are set to zero in MOVES3 following U.S. EPA (2015).





However, data was only collected down to +1.5 °C in that study. While diesel CO emissions were not statistically significant, $NO_x$ diesel emissions were a factor of 2.6 higher at this temperature compared to a factor of 0.32 at 7 °C. This, together with other studies discussed in Section 6.2.2, suggests that a much higher increment may be required for temperatures below 0 °C.

Figure E3 shows the log-linear function used to increase mobile $NO_x$ emissions based on decreases in daily average ambient temperatures.

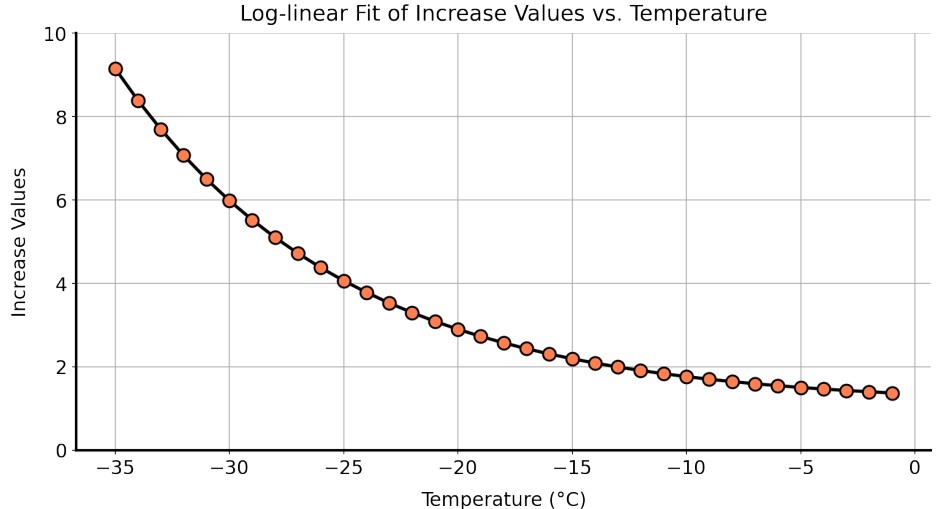

**Figure E3.** Increment applied to $NO_x$ mobile (on-road+non-road) emissions as a function of daily average temperature (°C) using a log-linear function.

### E5   Sensitivity to Photochemical Lifetimes

This section investigates the influence of photochemical lifetimes on simulated tracers. The photochemical lifetime of $SO_2$ is considered, but is estimated to be long, around 10 to 20 days since hydroxyl radical (OH) concentrations, one of the main loss

pathways for $SO_2$, are very low in Arctic winter (Yu et al., 2018; Green et al., 2019). Therefore, it is considered that transport and emissions of $SO_2$ are more important than photochemical loss during winter. The lifetime of CO is of the order of months during Arctic winter (AMAP, 2021) and not considered further.

However, a photochemical lifetime for $NO_x$ is considered and included in the run with temperature-dependent diesel vehicle emissions (run NOx_Emissions_LT). NO is lost by reaction with $O_3$ and reformed following photolysis of $NO_2$ but in winter

$O_3$ is fully or almost fully titrated since photolysis rates are very low, especially under conditions with strong surface-based temperature inversions with little vertical mixing of $O_3$ from aloft. Here, the lifetimes of NO and $NO_2$ are included and assumed to have a longer lifetime in SS compared to WS conditions. During polluted SS conditions $O_3$ concentrations are very low or even zero at the surface and $NO_x$ levels are high, whilst during WS conditions $O_3$ is higher and mixed down from aloft contributing to reduced $NO_x$ (Simpson et al., 2024). Kenagy et al. (2018) found that winter nighttime lifetimes were shorter (6.3





hours) than daytime (29 hours) due to the occurrence of nocturnal chemistry such as nitric acid and $N_2O_5$ production. However, if $O_3$ concentrations are very low, $N_2O_5$ formation is limited (Fibiger et al., 2018). In our study, lifetimes are assumed to be 8 and 12 hours for NO and $NO_2$, respectively, in WS conditions, and 48 hours for both species in SS conditions, in line with typical winter values and increased in SS conditions to account for titrated $O_3$. Inclusion of $NO_x$ lifetimes reduces the NMB and NME arising from inclusion of the temperature-dependent vehicle emissions, notably during the Mixed period (NMB and

NME reduced from 0.59 and 0.93 to -0.21 and 0.7, respectively, Table 4).

*Author contributions.* EPA-WRF simulations were provided by R.G. (EPA). Helikite data was provided by the EPFL team and Micromegas Helikite data by B.B. Downtown surface observations were provided by M.S.-M. and W.S. FLEXPART-WRF simulations were conducted by NB with contributions from T.O. and J.-C.R. Key contributions to the manuscript include N.B., K.S.L. and S.A., assisted by contributions from all co-authors. The views expressed in this article are those of the author(s) and do not necessarily represent the views or policies of the

U.S. Environmental Protection Agency.

*Competing interests.* The authors declare that they have no conflict of interest.

*Acknowledgements.* We thank the entire ALPACA science team of researchers for designing the experiment, acquiring funding, making measurements, and ongoing analysis of the results. The ALPACA project is organized as a part of the International Global Atmospheric Chemistry (IGAC) project under the Air Pollution in the Arctic: Climate, Environment and Societies (PACES) initiative with support from

the International Arctic Science Committee (IASC), the National Science Foundation (NSF), and the National Oceanic and Atmospheric Administration (NOAA). We thank University of Alaska Fairbanks and the Geophysical Institute for logistical support, and we thank Fairbanks for welcoming and engaging with this research. We also thank each of the power plant facilities for providing the power plant emissions for the duration of the campaign. N.B. K.S.L., B.D'A., J.S., S.A., B.B., S.B., H.D., E.D., A.I., T.O., J.-C.R., F.R., T.R., and B.T.-R. acknowledge support from the Agence National de Recherche (ANR) CASPA (Climate-relevant Aerosol Sources and Processes in the Arctic) project (grant

no. ANR-21-CE01-0017), and the Institut polaire français Paul-Émile Victor (IPEV) (grant no. 1215) and CNRS-INSU programme LEFE (Les Enveloppes Fluides et l'Environnement) ALPACA-France projects. We also acknowledge access to IDRIS HPC resources (GENCI allocation A013017141) for the FLEXPART-WRF simulations. S.R.A. and N.B. acknowledge support from the UK Natural Environment Research Council (grant ref. NE/W00609X/1). G.J.F. acknowledges support from NSF grants 2117971, 2146929, and 2232282. R.P. and J.S. received funding from the Swiss National Science Foundation grant no. 200021_212101. J.S. holds the Ingvar Kamprad Chair for Extreme

Environments Research. S.D., A.D., G.P., F.S. acknowledge support from the PRA ("Programma di Ricerche in Artico") 2019 programme (project "A-PAW") and from the ENI-CNR Research Center "Aldo Pontremoli". W.R.S. and M.C.-M. acknowledge support from NSF grants NNA-1927750 and AGS-2109134. J.M. acknowledges support from NSF grants NNA-1927750 and AGS-2029747. V.S. acknowledges support from NSF grants RISE-1927831 and AGS-2037091. P.F.D. and E.S.R. acknowledge funding support from NSF award 2012905.



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
