# Peer review of "Investigating processes influencing simulation of local Arctic wintertime anthropogenic pollution in Fairbanks, Alaska during ALPACA-2022"

_EGUsphere, 2024_

## Referee Comment (RC3)

**Reviewer's report on the manuscript by Brett et al. "Investigating processes influencing simulation of local Arctic wintertime anthropogenic pollution in Fairbanks, Alaska during ALPACA-2022", Atmospheric Chemistry and Physics, Manuscript ID: egusphere-2024-1450**

The manuscript presents an investigation of the processes influencing dispersion of local anthropogenic pollutants in a sub-Arctic urban setting (Fairbanks, Alaska) during wintertime. Through some detailed analysis of the comparisons between model simulations (using a particle dispersion model) and observations during the ALPACA campaign, the authors explored how the unique local meteorology at Fairbanks in wintertime (dominated by cold, stably stratified conditions) influence the dispersion and transport of pollutants from both surface and elevated sources. Overall, the paper is good and worthy of eventual publication. I do have a few comments (both general and specific) which I hope the authors can address to improve the paper's clarity.

General comments

1. Plume-rise calculation: The use of 12 hourly (03 and 15 local time) radiosonde data at Fairbanks airport for the plume rise (or injection height) calculation is problematic. Why not using the EPA-WRF hourly meteorology at stack locations for this since it is driving FLEXPART?

2. How does FLEXPART-WRF deal with possible plume recirculation (given the topographical feature of Fairbanks) as the model domain (shown in Figure 1) seems to be rather small? I may have missed it – how long is the forward tracer dispersion simulation after particle release?

3. Overprediction of surface SO2: I understand that the authors alluded to vertical mixing being a main factor affecting the model simulation of surface SO2 rather than chemical lifetime (oxidation) in the wintertime at Fairbanks. It might still be interesting to compare modelled SO2 with observed SO2+sulfate (in terms of total sulfur; I believe that sulfate was measured at CTC site as indicated in Simpson et al., 2024). Since the model does not consider sulfur oxidation, this can perhaps provide a check on whether modelled and observed total sulfur is comparable. If they are, this can then lead to questions such as whether oxidation pathways, other than the usual suspects of OH (in clear air) and H2O2, O3 (in aqueous phase), may be at play or whether it could be an emission issue, e.g., a larger portion of sulfur (from space heating, for example) could be emitted as sulfate rather than SO2 perhaps (with reference to Moon et al., ACS EST Air 2024, 1, 139-149). Just a thought.

4. I don't quite see the logic behind the constant emission test being used to illustrate the sensitivity of modelled tracers to meteorology. What the test illustrates is perhaps the impact of emission dependency on meteorology (e.g., dependency of CO emissions from diesel trucks on temperature)? Maybe I misunderstood the CONST-EM run. Were the campaign averaged emissions (or emission rates) applied to surface emissions only or applied to stack emissions also? If applied to stack emissions, were plume-rise calculation done the same way as CTRL run or done differently?

Specific comments

Line 17: Arctic haze occurs primarily during late winter and spring (highest occurrence in April and May) according to Shaw et al. (1995) rather than winter and early spring.

Line 158-160: The problem here is that the buoyancy force is only evaluated at the stack height while the atmospheric stability changes with height as in the case. It is particularly problematic under the strongly stratified situation or with elevated inversion layers. Akingunola et al. (2018) employed a methodology to evaluate the buoyancy force as the plume rises through the vertical column (model levels), which is more computational involved than the capping strategy used here.

Line 168: I don't understand why this is done, by removing negative temperature gradient.

Line 178: The use of +/- 8% of plume height for plume width/thickness – what is the basis for this?

Line 247-251: Were there fog and precipitation events during the ALPACA campaign?

Line 273-277: Are SS and WS regimes determined for the lowest 100 m only?

Figure 5: Are the results shown from the runs conducted with only surface or power plant emissions separately? Also, it may be clearer to denote SS and WS using the notation of filled (solid) and unfilled (open) circles.

Line 368-370: How is wintertime Arctic haze defined here? Are you really considering $0.012 – 0.1$ ug/m3 of $SO_2$ and $< 0.01$ ug/m3 sulfate aerosols as Arctic haze? These are much lower levels compared to the concentration levels (particularly in terms of sulfate) observed during the springtime Arctic haze events at High Arctic sites (e.g., Utqiagvik and Alert).

Line 380-383: The explanation of how the background CO and $NO_x$ concentrations are determined is not very clear.

Figure 8(c): Are these supposed to be time series of modelled plume height or the vertical extend of the power plant plumes?

Line 423-426: The difference between the radiosonde wind observation at Fairbanks airport and the LiDAR wind observation close to Aurora power plant would not necessarily affect the modelled dispersion of the Aurora power plant plume. The issue is whether the WRF simulated wind at the Aurora location is different from the LiDAR (close to Aurora) wind observation, since the dispersion calculation depends on the WRF simulated wind not the radiosonde observation at Fairbanks airport (I assume). Of course, the difference in temperature profiles between the airport and the power plant location could affect the plume injection height in this case. If there is a significant vertical wind shear at the power plant location, that would impact the modelled dispersion if the plume were misplaced in the model.

Line 433: I can't see any agreement between the nighttime radio sounding and the Helikite profiling below 270 m from Fig. 7a.

Line 436-437: The statement on the CTRL (with capping) being better than NO-CAP in this case is somewhat subjective. It could be argued that NO-CAP was a bit better than CTRL in terms of plume centerline (or height of maximum concentration) and vertical distribution in Fig.7b (case 2).

Line 444: what about the NOx plume at ~45 m?

Line 451&455: "Fig.D3a" should be Fig.D2a?

Line 459-462: Could perhaps compare the LiDAR aerosol plume with modelled SO2 plumes (given that the aerosols may be dominated by sulfate)?

Line 599: Fig.10 shows that NOx is modestly overestimated in this temperature range (-13C to -23C) without the inclusion of the adjustment for cold temperature.

Line 605-606: Why should the cold temperature effect on NOx emission from diesel vehicles be limited to stably stratified conditions only? In another word, why would the vehicle emissions be dependent on atmospheric stability?

Line 662-668 and Line 727-731: The rather complicated sensitivity to the vertical mixing (via the hmin setting), i.e., conflicting results amongst different pollutants (with enhanced or supressed vertical mixing), may also be an indication of compensating errors in the system.

---

## Author Comment (AC1)

**Article Review comments for: "Investigating processes influencing simulation of local Arctic wintertime anthropogenic pollution in Fairbanks, Alaska during ALPACA-2022", Atmospheric Chemistry and Physics, Manuscript ID: egusphere-2024-1450**

We thank the reviewers for their careful assessment of our manuscript. We found the comments very useful for improving and clarifying some of the information that was vague or unclear in the submitted manuscript. Below, we respond to the comments in turn, and summarise modifications made to the manuscript. Our responses are formatted as follows:

Reviewer comments **- black text (bold)**
Author responses - black text
Submitted manuscript text - blue text
Revised manuscript text - **blue text (bold)**

Line numbers refer to those in the original submission.

Note that minor corrections are also included in the revised manuscript: sulfate = sulphate, and hmin = $h_{min}$.

**Reviewer #1**

**The authors present a modeling analysis of CO, $NO_x$, and $SO_2$ concentrations in Fairbanks, Alaska to align with the ALPACA-2022 measurement campaign. They use the FLEXPART Langrangian tracer model driven by meteorology from WRF and emissions from local power plants and the local environmental regulator. The model performance is better under some conditions than others, and a number of sensitivity analyses are presented to identify potential reasons for poor performance.**

**Overall, I find the analysis thorough and compelling. I only have minor suggestions and questions to improve clarity.**

**Comments:**

**170: I found this description of the inversion diagnosis confusing. For example, the statement "no inversions are observed" is very peculiar given the importance of inversions noted throughout the rest of the study. I recommend rethinking how this diagnosis is presented.**

Thank you for pointing this out, the phrasing of the sentence is misleading. The sentence has been modified to make clear that we removed profiles when no inversions were observed.

Line 170: Negative temperature gradients throughout the profile are removed as no inversions are observed.

Line 170: **Profiles with negative temperature gradients, i.e. no surface or elevated inversions detected, are removed from the analysis.**

**Step changes introduced by calculating new injection heights every 12 hours is a limitation. The authors may consider the influence of smoothly varying heights between 12 hours calculated ones, e.g., using linear interpolation.**

This is a good suggestion, and it is something we considered. However, the model set up makes this challenging without significant updates to the model code. In addition, since linear interpolation assumes smooth changes in meteorological conditions, this approach also has limitations. Linear interpolation doesn't add any new information, and may not capture meteorological variability happening on timescales shorter than 12 hours. For instance, sometimes the meteorological conditions at the surface change suddenly, and clear out pollution in a step-wise manner, e.g. see Figure 3 on 25 January and 3 Feb where $NO_x$ concentrations are substantially reduced. For this reason, we argue that linear interpolation may not improve the model simulation of temporal variability significantly, without more vertical profile observations for validation. We have updated the text to clarify possible improvements to the model simulations in the future.

The plot below shows an example of the calculated plume rise, here for the Aurora stack, for 12 hourly data (black crosses) included in capping simulations. The black straight lines show the step changes. Hourly values calculated using linear interpolation are also shown (small pink circles). This comparison shows that the range of plume rise heights remains the same, and the overall change to the model results, is likely to be negligible.

[Figure]

**The point made on lines 369-370 need more context. It is unclear how the quoted concentrations of $SO_2$ and $SO_4^{2-}$ relate to the local and regional influence noted in the beginning of the paragraph.**

Thank you for this remark. We agree that the text reads as though Arctic Haze has very low concentrations of $SO_2$ and sulphate, which is not the case. In fact, there was an error in our conversion from µg/m$^3$ to ppb based on $SO_2$ data reported by Tran et al., 2011 (Figure 5 data). We apologise for this oversight. We have instead included updated examples of $SO_2$ concentrations in the Arctic (Skov et al., 2023), and of observed $SO_4^{2-}$ aerosol concentrations

in wintertime Arctic haze (Ioannidis et al., 2023). This point essentially highlights the large influence of local emissions in the Fairbanks area.

Lines 368-370: This regional pollution could be contributing to wintertime Arctic haze which has appreciably smaller concentrations of trace gases and aerosols. For example, 0.012-0.1 µg/m$^3$ (less than 0.1 ppb) of $SO_2$ and <0.01 µg/m$^3$ sulfate $SO_4^{2-}$ aerosols were observed at the Alaskan remote sites Denali and Poker Flat in January 2000 (Tran et al., 2011).

Lines 368-370: This regional pollution could be contributing to wintertime Arctic haze **which has lower concentrations** of trace gases and aerosols. **For example, simulated $SO_2$ concentrations at Villum in north-east Greenland range between 0.1 and 2.2 µg/m$^3$ (approx. 0.1-0.9 ppb) in 2018 and 2019 winter months (Skov et al., 2023). Sulphate concentrations between 0.1 to 0.8 µg/m$^3$ at Alert, Zeppelin and Villum in January and February 2014 were reported in Ioannidis et al. (2023), while sulphate in downtown Fairbanks ranged between 1-5 µg/m$^3$ during ALPACA 2022 (Moon et al., 2023).**

**380-386 is unclear – please clarify what was done to estimate dCO and dNOx**

We have revised these sentences. Reviewer 3 has also asked for clarification on this.

Lines 380-386: In this analysis, CO and $NO_x$ pollution plumes are identified in each of the Helikite flights when elevated concentrations are observed above the 90th percentile in the data of the flight. To compare to the model results, that are enhancements above background, a polluted background is assigned in each flight using the modal concentration of the observed concentration distribution, and subtracted from observed plume mixing ratios. The resulting $\delta CO$ and $\delta NO_x$ enhancements are compared with the model results. In order to evaluate power plant plumes only, this comparison only uses observations above 30 m, away from the influence of surface emissions. Some profiles of CO on 30 January and 10 February are removed due to issues with the CO sensor when the power was switched off to replace batteries and switched back on during the flight.

Lines 380-386: **Since model results are representative of enhancements above background, a polluted background is assigned to the Helikite CO and $NO_x$ measurements to determine observed pollution plume enhancements ($\delta CO$ and $\delta NO_x$) equivalent to the simulated quantities. First, the pollution plumes are assigned using the 90th percentile of the distribution of concentrations observed during each flight. A polluted background is assigned using the modal concentration of each flight, and subtracted from the identified plume to give the observed pollution plume enhancement ($\delta CO$ and $\delta NO_x$). In order to evaluate power plant plumes only, this comparison only uses observations above 30 m, away from the influence of surface emissions. Some profiles of CO on 30 January and 10 February are removed due to issues with the CO sensor.**

**416: note/clarify why only $NO_x$ observations are available**

There were issues with the CO sensors for this flight as mentioned in lines 385-386. This point has been clarified in the text.

 Only NO$_x$ observations are available with two plumes identified between 70-110 m and 160-210 m altitude, just below elevated inversions observed in the Helikite temperature profile data (Fig. 7a).

 Only NO$_x$ observations are available **for this case because of issues with the CO sensors on 30 January. Two plumes are** identified between 70-110 m and 160-210 m altitude, just below elevated inversions observed in the Helikite temperature profile data (Figure 7a).

**Table 4: green and red assignments are unclear – are these designated across simulations? If so, why are multiple values highlighted in each column and row?**

We agree that the classification was confusing and over-complicated. To simplify interpretation of the results presented in Table 4, the colour scheme has been updated to show only the largest and smallest NME/NMB values in red and green, respectively, for each column (i.e. each meteorological event), and for each tracer.

End of Table 4 caption: The values highlighted in green

End of Table 4 caption: The values highlighted **in green and red show the smallest and largest NMBs and NMEs, respectively, for each period and each tracer.**

New table:

| Simulation Name | NMB | | | | | NME | | | | |
|---|---|---|---|---|---|---|---|---|---|---|
| | All Data | AC-C | T-C | Mixed | T-W | All Data | AC-C | T-C | Mixed | T-W |
| CTRL CO | 0.02 | 0.02 | -0.34 | 0.5 | -0.55 | 0.52 | 0.39 | 0.37 | 0.68 | 0.56 |
| CONST-EM CO | 0.03 | -0.17 | -0.46 | 0.54 | -0.45 | 0.54 | 0.37 | 0.5 | 0.73 | 0.5 |
| MixH_100_CO | -0.19 | -0.17 | -0.55 | 0.21 | -0.64 | 0.47 | 0.34 | 0.55 | 0.48 | 0.64 |
| MixH_10_CO | 0.3 | 0.45 | -0.04 | 0.79 | -0.45 | 0.66 | 0.66 | 0.35 | 0.92 | 0.51 |
| CTRL NOx | -0.46 | -0.65 | -0.8 | -0.03 | -0.74 | 0.69 | 0.66 | 0.8 | 0.66 | 0.74 |
| NOx Emissions | -0.07 | -0.05 | -0.47 | 0.59 | -0.67 | 0.68 | 0.45 | 0.5 | 0.93 | 0.67 |
| NOx Emissions_LT | -0.23 | -0.17 | -0.5 | -0.21 | 0.7 | 0.61 | 0.41 | 0.52 | 0.7 | 0.7 |
| MixH_100_NOx | -0.25 | -0.32 | -0.67 | -0.01 | -0.77 | 0.61 | 0.42 | 0.67 | 0.65 | 0.77 |
| MixH_10_NOx | -0.2 | 0.2 | -0.25 | 0.44 | -0.62 | 0.61 | 0.53 | 0.44 | 0.76 | 0.62 |
| CTRL SO2 | 0.6 | 0.03 | -0.3 | 1.26 | 0.6 | 0.94 | 0.49 | 0.4 | 1.37 | 0.95 |
| SO2 SOR | 0.47 | -0.02 | -0.29 | 1.08 | 0.32 | 0.84 | 0.48 | 0.37 | 1.22 | 0.74 |
| MixH_100_SO2 | 0.03 | -0.26 | -0.57 | 0.43 | -0.08 | 0.62 | 0.42 | 0.63 | 0.7 | 0.54 |
| MixH_10_SO2 | 1.08 | 0.5 | 0.1 | 1.71 | 1.0 | 1.31 | 0.79 | 0.55 | 1.8 | 1.18 |

**Figure 10: "surface emission mixing ratios" are unclear. Are these concentration contributions from each sector calculated with sensitivity analyses? The description in lines 580-585 is not sufficient to interpret the figure.**

Figure 10 is showing estimated mixing ratios over the Downtown area calculated using EPA emissions fluxes, using an assumption about a mixed layer depth at the surface. No model results are shown here. During the cold period estimated mixing ratios slightly overestimate CO compared to the observations (+23 %), whereas $NO_x$ is underestimated (-43 %) (Figure 10b). Since traffic is the main source of both CO and $NO_x$, this suggests that the vehicle $NO_x$ emissions are underestimated in the inventory in the coldest conditions. Adding a temperature enhancement for $NO_x$ brings the inventory and observations closer as shown in the right panel of Figure 10b. More information has been added to lines 580-582 to make this clearer. This point was misinterpreted by all 3 reviewers, showing that our initial explanation was lacking clarity. The text and figure caption have been revised. Note this also includes changes based on comments from reviewers #2 and #3 (lines 584-589).

Lines 580-582: The possible contribution of temperature dependent diesel emissions to CO and $NO_x$ in Fairbanks is investigated first by summing the EPA emissions for each sector over the volume of the box covering 4 grid cells in Downtown and up to 10 m to estimate hourly concentrations that are compared to CTC observations.

Lines 580-582: The possible contribution of temperature dependent diesel emissions to CO and $NO_x$ **concentrations in Fairbanks is investigated based on the EPA surface emission fluxes that are used in the model simulations. In order to compare to surface observations at CTC, surface fluxes for each emission sector, in $kg/m^2/s$, are converted into hourly mixing ratios (in ppb) by taking into account the volume of each emission grid cell ($1.3 \ km^2$ x 10 m (AGL) in vertical). These estimates are averaged over the 4 grid cells covering the Downtown area and shown in Figure 10.**

Lines 584-589: The observations show a clear increase in $NO_x$ at colder temperatures, especially below -23°C, but is much less distinct for observed CO. For CO, as noted earlier, a cold temperature dependence is already included for mobile (on-road and non-road) gasoline emissions in MOVES3, and there is good agreement between the CO observations and estimated mixing ratios during the cold polluted period (Fig. 10b). The poor agreement between $NO_x$ observations and estimated $NO_x$ supports the hypothesis that an increase in diesel $NO_x$ vehicle emissions due to a cold temperature dependence may be required. Estimated mixing ratios are underestimated in the warm polluted period compared to observations for both CO and $NO_x$ (Fig. 10c), indicating that a cold-temperature effect is not driving the discrepancy in this period.

Lines 584-589: **At intermediate temperatures (-13°C to -23°C), common during the Mixed period, estimated $NO_x$ mixing ratios are overestimated compared to observations. This is in part because meteorology and mixing are not considered, as also shown for CO. However, t**he observations show a clear increase in $NO_x$ at colder temperatures, especially below -23°C, which is much less distinct for observed CO. For CO, as noted earlier, a cold temperature dependence is already included for mobile (on-road and non-road) gasoline emissions in MOVES3, **and there is better agreement between the CO observations and estimated mixing ratios during the cold polluted period (CPP) (Figure 10b)**. The poor agreement between $NO_x$ observations and estimated $NO_x$ **during the CPP** supports the hypothesis that an increase in diesel $NO_x$ vehicle emissions due to a cold temperature dependence may be required. **Furthermore, estimated CO and $NO_x$ mixing ratios are both**

**underestimated during T-W (Figure 10c),** indicating that a cold-temperature effect is not driving the discrepancy in this period.

Figure 10 caption: EPA surface emission mixing ratios calculated over the Downtown area for contributing emission sectors (grey lines) compared to observations at the CTC site (black lines) in ppb averaged over 3 °C temperature bins for (a) full campaign, b) 29 January to 2 February and c) 23 to 25 February, for CO (left) and $NO_x$ (right). Different emission sectors coloured are shown and given in the legend. The $NO_x$ increment from the linear temperature dependence is shown with '..' hatching. The mid-point of the 3 °C temperature bin is shown on the x-axis. The bold grey lines correspond to the total surface-emitted emissions listed in the legend with the $NO_x$ vehicle temperature increment. See text for details.

**Figure 10: Estimated mixing ratios in ppb, based on EPA surface emission fluxes, averaged over the Downtown area, compared to observed mixing ratios in ppb at the CTC site (black circles), averaged over 3 °C temperature bins for a) full campaign, b) 29 January to 2 February, and c) 23 to 25 February, for CO (left) and $NO_x$ (right). The shaded colours correspond to the contributing emission sectors indicated in the legend (total of all sectors = grey diamonds), The mid-point of the 3 °C temperature bin is shown on the x-axis. The increment in $NO_x$ vehicle emissions at low temperatures, according to the linear temperature dependence is also shown with cyan shading and '..' hatching. See text for details.**

**The manuscript by Brett et al. presents an investigation of the role of the boundary layer in controlling the distribution of locally emitted pollutants in Fairbanks Alaska under, generally, extremely stable conditions found during long winter nights. In addition to widespread, ground-based emissions, the authors pay particular attention to emissions from a number of power plant stacks and the role of plume rise in controlling the dispersion of the emitted species. The investigation is based on a comparison of a Lagrangian particle dispersion model connected to a mesoscale meteorological model (WRF) with observations made as part of the Alaskan Layered Pollution and Chemical Analysis (ALPACA) field campaign.**

**The research on dispersion of pollutants under the very particular meteorology of Fairbanks Alaska in the winter is a challenging topic and a fascinating case study. The work presented here is well thought out and thoroughly executed. My only significant criticism is the tremendous volume of results that are being presented in a single paper. The manuscript discusses the physical layout of Fairbanks and important emission sources, the observations, the plume rise parameterization, FLEXPART, WRF, analysis of the meteorology during the study period, tracer results for the elevated plumes, comparisons with surface observations for the tracers, a sensitivity test with constant emissions, effects of cold temperatures on diesel NOx emissions, sensitivity tests with different minimum mixing layer heights, sensitivity tests with different approaches to plume rise and sensitivities to different assumptions of SO$_2$ oxidation. And there is additional material in the appendix. I would urge the authors to consider paring back the results and move additional material to the appendix and would suggest the discussion of the sensitivity to the minimum mixing layer height as one candidate. While the difficult of simulating boundary layer mixing under very stable conditions is certainly a topic of relevance for the current study the differences with the '100 hmin' and '10 hmin' cases are not anything unexpected and could be dealt with much more compactly in the body of the manuscript. The addition of the results from '100 hmin' and '10 hmin' to Figure 9 has the additional negative effect of making the plot very difficult to read. The results of tests with constant emissions is another topic I could also suggest to move to the appendix to reduce and focus the discussion.**

**Maybe it is just me, but I found there were a few places in the manuscript where the message the authors are trying to convey is difficult to follow because of overly vague wording or convoluted phrases. I have pointed places where I had a bit of trouble following the authors for one reason or another in the list of minor comments.**

Thank you for this feedback. We agree that there is a lot of information in the paper. As suggested, we have now moved the majority of the mixing height section to Appendix E6 and included a short summary of the main points in the main text (see below). To further shorten some of the main text, we also moved the evaluation of the WRF results to Appendix B, as this is prior work to this study carried out by one of the co-authors.

We have also revised Figure 9a to remove h$_{min}$ 10 and 100 m simulations from the time series. We agree that this was hard to interpret and it can more easily be interpreted from Figure 9b (diurnal cycles). See the updated figure below. We have also updated the appendix figures

E2 and E3 by removing $h_{min}$ = 10 m and 100 m from the time series, again they remain in the diurnal cycles.

[Figure]

**6.2.4 Sensitivity to Vertical Mixing**

[revised manuscript text omitted]

**Minor comments:**

**Line 5: 'Regional enhancements, simulated up to 200 m, are due to elevated power plant emissions above 50 m, with south-westerly pollutant outflow.', Coming in the abstract and dropped in with little context, it is very difficult for a new reader to understand what is being described. In fact, this section of the abstract is composed of a couple of disjointed statements that are dropped in and somewhat independent of each other, making it difficult to form a clear idea of the work that was done or the results. I would suggest a bit of minor reworking to give the abstract a clearer message.**

Thank you for this comment. We have revised the abstract to improve clarity as follows:

Abstract:
Lagrangian tracer simulations are deployed to investigate processes influencing vertical and horizontal dispersion of anthropogenic pollution in Fairbanks, Alaska, during the ALPACA-2022 field campaign. Simulations of carbon monoxide (CO), sulphur dioxide ($SO_2$) and nitrogen oxides ($NO_x$), including surface and elevated emissions, are highest at the surface under very cold stable conditions.  Inclusion of a novel power plant plume rise treatment that considers the presence of surface and elevated temperature inversion layers leads to improved agreement with observed CO and $NO_x$ plumes with discrepancies attributed to, for example, displacement of plumes by modelled winds. At the surface, model results show that observed CO variability is largely driven by meteorology and to a lesser extent by emissions, although simulated tracers are sensitive to modelled vertical dispersion. Modelled underestimation of surface $NO_x$ during very cold polluted conditions is considerably improved following the inclusion of substantial increases in diesel vehicle $NO_x$ emissions at cold temperatures (e.g. a factor of 6 at -30 °C). In contrast, overestimation of surface $SO_2$ is attributed  This study highlights the need for

improvements to local wintertime Arctic anthropogenic surface and elevated emissions and improved simulation of Arctic stable boundary layers.

Abstract:
Lagrangian tracer simulations are deployed to investigate processes influencing vertical and horizontal dispersion of anthropogenic pollution in Fairbanks, Alaska, during the ALPACA-2022 field campaign. Simulations of carbon monoxide (CO), sulphur dioxide ($SO_2$) and nitrogen oxides ($NO_x$), including surface and elevated **sources,** are highest at the surface under very cold stable conditions. **Pollution enhancements above the surface (50-300 m) are mainly attributed to elevated power plant emissions. Both surface and elevated sources are contributing to Fairbanks regional pollution that is transported downwind, primarily to the south-west, and may be contributing to wintertime Arctic haze.** Inclusion of a novel power plant plume rise treatment**,** that considers the presence of surface and elevated temperature inversion layers**,** leads to improved agreement with observed CO and $NO_x$ plumes with discrepancies attributed to, for example, displacement of plumes by modelled winds. At the surface, model results show that observed CO variability is largely driven by meteorology, and to a lesser extent by emissions, and simulated tracers are sensitive to modelled vertical dispersion. Modelled underestimation of surface $NO_x$ during very cold polluted conditions is considerably improved following the inclusion of substantial increases in diesel vehicle $NO_x$ emissions at cold temperatures (e.g. a factor of 6 at -30 °C). In contrast, overestimation of surface $SO_2$ is attributed **mainly to model deficiencies in vertical dispersion of elevated (5-18 m) space heating emissions.** This study highlights the need for improvements to local wintertime Arctic anthropogenic surface and elevated emissions and improved simulation of Arctic stable boundary layers.

**Line 130 – 131: The last 'and' in 'and solvent use, and are also emitted at the surface.' Seems out of place.**

Thank you for pointing out this typo.

Lines 129-131: On-road and non-road mobile sources take into account week-day and weekend differences and are emitted at the surface (0-4 m).

Line 129-131: On-road and non-road mobile sources take into account week-day and weekend differences and are emitted at the surface (0-4 m). **Likewise, non-point sources, including stationary fuel combustion, commercial cooking and solvent use, are also emitted between 0-4 m.**

**Lines 236 – 238: The exact relationship between particles and the emission mass / concentration is a bit difficult to understand. The authors write 'All tracers are assigned masses according to their emission mass at hourly time resolution. For each power plant facility, 5000 particles are released hourly for each tracer and diurnal variability is calculated from the diurnal cycle for each power plant stack.' Does that mean 5000 particles are released for each stack each hour, irrespective of the actual emissions in that hour, but that the mass of each particle is scaled by the emissions in that hour?**

You have mostly interpreted this correctly, however the total number of particles are scaled by the emission mass. The text has been revised to clarify this point.

Lines 236-238: All tracers are assigned masses according to their emission mass at hourly time resolution. For each power plant facility, 5000 particles are released hourly for each tracer, and diurnal variability is calculated from the diurnal cycle for each power plant stack.

Lines 236-238: **For each power plant stack, 5000 particles are released for each tracer, every hour, providing the stack was operational. The total number of particles is scaled by the emission mass, and distributed evenly between the particles. Diurnal variability is calculated from the diurnal cycle for each stack.**

**Lines 241 – 242: Somewhat similar to the question about lines 236 – 238 describing emissions from the power plant stacks, here the text states '80,000 particles are released hourly and weighted according to the emission mass in each 1.33 x 1.33 km2 grid cell, extending over the wider Fairbanks and North Pole area (Fig. 1).' Is the 80,000 particles released each hour the total over the entire model domain or are 80,000 particles released in each grid cell? And if the number of particles is constant for each hour, it is the mass of each particle that is adjusted to reflect the emission intensity?**

Thank you for pointing out the unclear phrasing. The number of particles in each grid cell depends on the emission mass, which is distributed evenly between particles. In fact, surface sources are emitted over an area that is smaller than the full domain, but extends over the FNSB region. The text has been revised to clarify this point.

Lines 241 – 242: 80,000 particles are released hourly and weighted according to the emission mass in each 1.33 x 1.33 km$^2$ grid cell, extending over the wider Fairbanks and North Pole area (Fig. 1).

Lines 241 – 242: **A total of 80,000 particles are released every hour, over the FNSB nonattainment area. The number of particles in each grid cell depends on the emission mass, which is distributed evenly between all particles.**

**Lines 290 – 291: 'For instance, days with strong stability in the surface layer (0-25 m) and weaker stability aloft (> 25 m), such as 25 January…' When I look at dT (23 – 3 m) from Figure 3c, it looks like January 25th had very weak stability with dT almost zero for at least part of the 25th.**

Thank you for pointing this out. There was a mistake, the text should have stated 24 January. On 24 January the surface stability is strong but this is not the case aloft (the surface is decoupled from large-scale influence), and by 25 January the surface stability is disrupted, and the large-scale conditions influence the surface resulting in lower pollution levels. We have revised the text accordingly. Note that EIs correspond to elevated inversions (already defined).

Lines 290-292: For instance, days with strong stability in the surface layer (0-25 m) and weaker stability aloft (> 25 m), such as 25 January, indicate a decoupling of the surface layer from EIs that are linked to large scale meteorology.

Lines 290-292: For instance, days with strong stability in the surface layer (0-25 m), and weaker stability aloft (> 25 m), such as **24 January,** indicate a decoupling of the surface layer from EIs aloft that are linked to large-scale meteorology. **By 25 January, the large-scale synoptic conditions influence the surface level, as shown by weak temperature gradients, and substantial reductions in surface pollution (e.g. NO$_x$, panel a).**

**Lines 298 – 299: It is not clear to me what is meant by 'The T-C period corresponds to the formation of a high-low pressure gradient disrupting anticyclonic conditions.' In particular, the phrase 'high-low pressure gradient'.**

Here, we refer to the high pressure and low pressure systems influencing meteorology over interior Alaska.

Lines 298 – 299: The T-C period corresponds to the formation of a high-low pressure gradient disrupting anticyclonic conditions.

Lines 298 – 299: **During the transient T-C period, the high pressure system that was positioned over interior Alaska during A-C is interrupted by the northward movement of the Aleutian low pressure system, resulting in a high-low pressure gradient.**

**Figure 3 caption: The caption says that panel (b) gives the wind speed at 3 m elevation, but the y-axis states that the wind is at 23 m. Assuming it is the wind at 23m, I wonder if the 3m windspeed would be more indicative of stability in the very lowest layers of the atmosphere? In particular, there are strong winds on 01/24, some of the highest windspeeds observed, that is labelled as a Strongly Stable period.**

Thank you for spotting this mistake. The caption has been corrected to say 23 m. We decided against using 3 m winds, since they can often be less accurate due to interferences from buildings and trees, as was the case at the CTC site in the Fairbanks downtown area. For this reason, we include 23 m winds, that still provide a good indication about near-surface conditions vs large-scale meteorology.

**Lines 332 – 333: I am having trouble understanding what is meant by 'Interestingly, concentrations are enhanced during WS compared to SS conditions above 200 m, when winds are east-northeast (WS) as opposed to south-east (SS).' Are there four cases being compared here, WS with east-northeast winds, SS with east-northeast winds, WS with south-east winds, SS with south-east winds? Or just two cases because winds are predominantly east-northeast under WS conditions and more south-easterly under SS conditions?**

We intended to make the point that during SS conditions, average winds are generally easterly below 200 m and weak (<3 m/s). But, between 200-300 m they are more south-easterly on average (and stronger, > 3 m/s). In contrast, during WS conditions, generally strong (>3 m/s) north/north easterly winds prevail below 200m, with strong easterly winds between 200-300 m, i.e. the winds over these 2 altitude ranges are more comparable during WS than SS conditions, with more stratification during SS conditions.

Lines 332-333: SO$_2$ is also influenced by power plant emissions at 200-300 m with enhancements up to 1-2 ppb.

to SS conditions above 200 m, when winds are east-northeast (WS) as opposed to south-east (SS). These results may also be due to stronger upward transport during WS conditions.

Lines 332-333: SO$_2$ is also influenced by power plant emissions at 200-300 m with enhancements up to 1-2 ppb. **Concentration enhancements are larger during WS conditions when winds are often north-easterly below 200 m, and stronger (> 3 m/s), compared to SS conditions when winds are from the east and weaker (< 3 m/s). At 200-300 m, wind speeds are strong in both WS and SS conditions. Weaker winds in the lower ABL in SS conditions, reflect increased stratification and limited vertical transport, with stronger winds and more vertical exchange during WS conditions.**

**Lines 336 – 337: I think I understand what the authors are trying to convey with the statement 'CO concentrations above 50 m relative to 0-10 m are inappreciable compared to SO$_2$ because…' but the phrasing is difficult to parse.**

The text has been revised to clarify this point. CO concentrations are much higher at the surface than aloft (e.g. up to 2.5 ppm vs 300 ppb, respectively, see Figure 9a and 6a). On the other hand, SO$_2$ concentrations are more comparable in magnitude at the surface and aloft (up to approx. 50 ppb for both, see Figure 5). This is because CO power plant emissions are relatively much lower than CO surface emissions, which is not the case for SO$_2$ emissions. The original explanation was misleading because we compared the ratios of power plant emissions (total) and surface emissions (per grid cell), in order to use quantitative information from Figure A1. Firstly, note that Figure A1 has now been updated as the surface calculated means were previously divided by the number of grid cells, and not by the number of grid cells multiplied by 1.33 x 1.33 km$^2$, to give the units in kg hour$^{-1}$ km$^{-2}$. This mistake was spotted after submission. We have also added in an additional subplot to Figure A1 (panel a, i), showing the total emission mass over all grid cells in the Fairbanks non-attainment area (kg/hour). This is included to improve the explanation in lines 336-340, by comparing the total power plant emissions (Figure 2a) to total surface emissions (Figure A1 a,i). The figure is included below, together with the updated caption and modifications to lines 747-749.

Lines 336-340: CO concentrations above 50 m relative to 0-10 m are inappreciable compared to SO$_2$

Lines 336-340: CO **enhancements** above 50 m, relative to 0-10 m, are inappreciable compared to SO$_2$, **because CO surface emissions are much larger relative to power plant emissions, than is the case for SO$_2$. For instance, CO surface emissions (campaign average) in the Fairbanks non-attainment area (approx. 950 kg/hour, Figure A1 a,i), are a factor of 10 higher than total CO power plant emissions (approx. 98 kg/hour, Figure 2a), whereas the emission masses for SO$_2$ are comparable in both cases (110-120 kg/hour).**

[Figure]

Figure A1. CO, SO2, NO and NO2 emissions (kg hour−1 km−2) averaged over the campaign, summed between 0-18 m altitude, and averaged per 1.33 km grid cell for a) Fairbanks Non-attainment area, b) North Pole Non-attainment area, c) Downtown, d) Hamilton Acres and e) UAF Farm. Panels c to e are for the EPA emissions grid cell (1.33 km) closest to the location. See Fig. 1 for details. **averaged over the campaign, summed between 0-18 m altitude in: (a, i) Fairbanks non-attainment area, sum of 129 grid cells (1.33 x 133 km$^2$) (kg hour$^{-1}$). In panels (a, ii) and (b-e), the emissions are averaged per grid cell (kg hour$^{-1}$ km$^{-2}$) for (a, ii) Fairbanks non-attainment area, b) North Pole nonattainment area, c) Downtown, d) Hamilton Acres and e) UAF Farm.** Panels c to e are for the EPA emissions grid cell closest to the location. See Figure 1 for details.

Lines 747-749: Figure A1 shows surface emissions for each sector averaged over the non-attainment areas, Downtown, Hamilton Acres (HA) and UAF Farm areas.

Lines 747-749: Figure A1 shows **campaign averaged surface emissions for each sector,** over the non-attainment areas, Downtown, Hamilton Acres (HA) and UAF Farm areas**.**

**Lines 388 – 389: Mention of run NO-CAP is a bit redundant here, as the authors return to this run at lines 394 – 396.**

This is a good point, the sentence in question has been removed.

Lines 387-391: Figure 6 shows the comparison of model results from CTRL and observed enhancements for each of the identified plumes for CO and NO$_x$ during the campaign when flights took place. The model is also run without the plume capping at the point of emission injection (run NO-CAP). CTRL generally captures plume presence aloft when compared with

observed δNOx and δCO above 30 m (Figure 6a), although there are some displacements that could be due to temporal biases in modelled wind speeds and directions or in the diagnosed injection height.

**Figure 7: While I can understand the need to compare the radiosonde temperatures with the observations from the helikite, why do the authors not compare the temperature profiles from WRF with the helikite observations. While the plume rise calculations based on the radiosonde data are clearly important, I would have thought the WRF simulation of the temperature structure would also be important for the vertical mixing simulated by FLEXPART..**

The reviewer is correct that the model simulations are dependent on the WRF meteorological fields, including winds and temperature. The reason for showing the radiosonde profiles in Figure 7 is because they are used in the simulated plume rise capping. For instance, they show EIs aloft that may have capped some of the power plant emissions. We have included an additional sentence to make this clearer.

Inclusion of WRF temperatures in this figure would require additional discussion about WRF compared to the Helikite temperature profiles, and weaken the focus of the evaluation of the model results. Such comparisons have been carried out elsewhere and are now noted in the revised manuscript (main text and Appendix B). The WRF model performs less well at the UAF Farm compared to CTC due to difficulties in capturing the local drainage flow. Note that the Fochesatto et al., 2023 reference has been updated since the manuscript is now submitted (2024).

Lines 406-409: Observed Helikite temperature profiles are shown together with radiosonde temperature profiles at 1500 and 0300 AKST for the days in question, for each case in Fig. 7a.

Lines 406-409: Observed Helikite temperature profiles are shown together with radiosonde temperature profiles at 1500 and 0300 AKST for the days in question, for each case in Figure 7a. **Radiosonde profiles are shown to provide information regarding the diagnosed SBIs and EIs used in the plume rise capping parameterisation.**

**Lines 574 – 575: 'Hence the cold temperature dependence for diesel NO$_x$ emissions may be too weak in MOVES3', is confusing. The use of the phrase 'too weak' seems to imply there is at least some temperature dependence, but at line 568 the authors state that for MOVES3 the 'cold-temperature dependencies for diesel vehicle cold starts for both CO and NO$_x$ are set to zero'.**

This is a good point. The text has been revised.

Lines 574-575: Hence the cold temperature dependence for diesel NO$_x$ emissions may be too weak in MOVES3, resulting in a substantial underestimate in modelled NO$_x$ during cold stable conditions.

Lines 574-575: **Hence, the lack of cold temperature dependence for diesel NO$_x$ emissions in MOVES3, may be contributing to the substantial underestimate in modelled NO$_x$ during cold conditions.**

**Figure 10 caption: Perhaps the authors could point out that the grey triangles are the model calculated mixing ratios for the default case with a zero temperature dependence on diesel $NO_x$ emissions. It may help the reader more quickly understand what is being shown in Figure 10, because the eye is too easily drawn to the top of the shaded region.**

The description about how the mixing ratios shown in Figure 10 were obtained was confusing and misinterpreted by all 3 reviewers. The purpose of using this approach is also clarified.

This figure is not showing model simulated mixing ratios, but estimated mixing ratios over the Downtown area calculated using EPA emission fluxes used in the model simulations, using an assumption about a mixed layer depth at the surface. Surface emission fluxes are used to estimate mixing ratios that can be compared to the observations at CTC as a function of temperature. During the cold period estimated mixing ratios slightly overestimate CO compared to the observations (+23 %), whereas $NO_x$ is underestimated (-43 %) (Figure 10b). Since traffic is the main source of both CO and $NO_x$, this suggests that the vehicle $NO_x$ emissions are underestimated in the inventory in the coldest conditions. Adding a temperature enhancement for $NO_x$ brings the inventory and observations closer as shown in the right panel of Figure 10b. More information has been added to lines 580-582 to make this clearer and the figure caption has been revised. The revisions also take into account comments from reviewers #1 and #3 (lines 584-589).

Lines 580-582: The possible contribution of temperature dependent diesel emissions to CO and $NO_x$ in Fairbanks is investigated first by summing the EPA emissions for each sector over the volume of the box covering 4 grid cells in Downtown and up to 10 m to estimate hourly concentrations that are compared to CTC observations.

Lines 580-582: The possible contribution of temperature dependent diesel emissions to CO and $NO_x$ **concentrations in Fairbanks is investigated based on the EPA surface emission fluxes that are used in the model simulations. In order to compare to surface observations at CTC, surface fluxes for each emission sector, in $kg/m^2/s$, are converted into hourly mixing ratios (in ppb) by taking into account the volume of each emission grid cell (1.3 $km^2$ x 10m (AGL) in vertical). These estimates are averaged over the 4 grid cells covering the Downtown area and shown in Figure 10.**

Lines 584-589: The observations show a clear increase in $NO_x$ at colder temperatures, especially below -23°C, but is much less distinct for observed CO. For CO, as noted earlier, a cold temperature dependence is already included for mobile (on-road and non-road) gasoline emissions in MOVES3, and there is good agreement between the CO observations and estimated mixing ratios during the cold polluted period (Fig. 10b). The poor agreement between $NO_x$ observations and estimated $NO_x$ supports the hypothesis that an increase in diesel $NO_x$ vehicle emissions due to a cold temperature dependence may be required. Estimated mixing ratios are underestimated in the warm polluted period compared to observations for both CO and $NO_x$ (Fig. 10c), indicating that a cold-temperature effect is not driving the discrepancy in this period.

Lines 584-589: **At intermediate temperatures (-13°C to -23°C), common during the Mixed period, estimated $NO_x$ mixing ratios are overestimated compared to observations. This is in part because meteorology and mixing are not considered, as also shown for CO. However, t**he observations show a clear increase in $NO_x$ at colder temperatures, especially

below -23°C, which is much less distinct for observed CO. For CO, as noted earlier, a cold temperature dependence is already included for mobile (on-road and non-road) gasoline emissions in MOVES3, **and there is better agreement between the CO observations and estimated mixing ratios during the cold polluted period (CPP) (Figure 10b**). The poor agreement between $NO_x$ observations and estimated $NO_x$ **during the CPP** supports the hypothesis that an increase in diesel $NO_x$ vehicle emissions due to a cold temperature dependence may be required. **Furthermore, estimated CO and $NO_x$ mixing ratios are both underestimated during T-W (Figure 10c**), indicating that a cold-temperature effect is not driving the discrepancy in this period.

Figure 10 caption: EPA surface emission mixing ratios calculated over the Downtown area for contributing emission sectors (grey lines) compared to observations at the CTC site (black lines) in ppb averaged over 3 °C temperature bins for (a) full campaign, b) 29 January to 2 February and c) 23 to 25 February, for CO (left) and $NO_x$ (right). Different emission sectors coloured are shown and given in the legend. The $NO_x$ increment from the linear temperature dependence is shown with '..' hatching. The mid-point of the 3 °C temperature bin is shown on the x-axis. The bold grey lines correspond to the total surface-emitted emissions listed in the legend with the $NO_x$ vehicle temperature increment. See text for details.

**Figure 10: Estimated mixing ratios in ppb, using surface EPA emission fluxes, averaged over the Downtown area, compared to observed mixing ratios in ppb at the CTC site (black circles), averaged over 3 °C temperature bins for a) full campaign, b) 29 January to 2 February, and c) 23 to 25 February, for CO (left) and $NO_x$ (right). The shaded colours correspond to the contributing emission sectors indicated in the legend (total of all sectors = grey diamonds), The mid-point of the 3 °C temperature bin is shown on the x-axis. The increment in $NO_x$ vehicle emissions at low temperatures, according to the linear temperature dependence is also shown with cyan shading and '..' hatching. See text for details.**

**Reviewer #3**

**The manuscript presents an investigation of the processes influencing dispersion of local anthropogenic pollutants in a sub-Arctic urban setting (Fairbanks, Alaska) during wintertime. Through some detailed analysis of the comparisons between model simulations (using a particle dispersion model) and observations during the ALPACA campaign, the authors explored how the unique local meteorology at Fairbanks in wintertime (dominated by cold, stably stratified conditions) influence the dispersion and transport of pollutants from both surface and elevated sources. Overall, the paper is good and worthy of eventual publication. I do have a few comments (both general and specific) which I hope the authors can address to improve the paper's clarity.**

**General comments:**

**1.Plume-rise calculation: The use of 12 hourly (03 and 15 local time) radiosonde data at Fairbanks airport for the plume rise (or injection height) calculation is problematic. Why not using the EPA-WRF hourly meteorology at stack locations for this since it is driving FLEXPART?**

Thank you for this suggestion. We did consider this approach, but we opted to calculate offline using radiosonde observations instead of online using WRF. Use of EPA-WRF fields would require detailed validation of WRF at each of the stack locations. This would entail evaluation against vertical profile observations that are not available. The WRF model was run with assimilated data including winds from the lidar at CTC (first 2 weeks of the campaign) and UAF Farm (last 3 weeks of the campaign). The results, discussed in Fochesatto et al. (2024), show that the WRF model performs better at the downtown area than UAF Farm due to difficulties in capturing the local drainage flow. These findings are discussed in Appendix B and in Fochesatto et al., 2024 (was 2023 - now submitted (2024), the reference has been updated in the manuscript).

It can also be noted that FLEXPART-WRF is not currently set up for online plume rise calculations, however we have been in discussion with a group working on doing this, using Briggs equations and WRF temperatures etc for buoyancy calculations at each time step.

While we appreciate the suggestion as an alternative approach, we prefer to rely on available observational constraints available from the radiosonde profiles. We acknowledge the alternative approach, as a potential for further work, in the Conclusions. This revision also considers the Reviewer's second comment on the plume rise algorithm.

See lines 696-699: The plume rise calculations could be improved further by using WRF temperatures and winds at the location of the power plant stacks, rather than using radiosondes at Fairbanks airport, allowing spatial differences to be better captured. Acquisition of more vertical profile observations (e.g. using drones) at, and downwind of, the power plant stacks would also be valuable.

Lines 696-699: The plume rise calculations could be improved further by using WRF temperatures and winds at the location of the power plant stacks, rather than using radiosondes at Fairbanks airport, allowing spatial differences to be better captured. **The treatment of vertical plume rise could be further improved by taking into account the**

**changes in the buoyancy force of the plume as it rises above the stack, for example, as in Akingunola et al. (2018).** Acquisition of more vertical profile observations (e.g. using drones) at, and downwind of, the power plant stacks would also be valuable.

**2. How does FLEXPART-WRF deal with possible plume recirculation (given the topographical feature of Fairbanks) as the model domain (shown in Figure 1) seems to be rather small? I may have missed it – how long is the forward tracer dispersion simulation after particle release?**

Thank you for these remarks. The particles in FLEXPART-WRF are lost only by deposition processes or when they leave the domain. The overall domain is larger (200x200 grid cells), than the area where the sources are emitted over the FNSB area in the centre of the domain. Therefore, recirculation within the large domain is included. However, we do not analyse recirculation specifically in this study. We have acknowledged pollution recirculation in the conclusion and perspectives. Note that the tracers are released continuously, every hour.

Additionally, we started to investigate this by using the age classes of the particles, and found that most of the particles within the FNSB domain are <2 days old after emission. These initial results suggest that most of the pollution is fresh, and the influence of recirculation on longer timescales is minimal. However, to investigate this in more detail, would require 3D chemical-aerosol simulations over a larger domain.

Lines 689-692: Pollution outflow to the south-west, due to dominating north-easterly winds up to 200 m suggests a possible regional influence

Lines 687-692: Pollution outflow to the south-west, due to dominating north-easterly winds up to 200 m suggests a possible regional influence **due to anthropogenic emissions from Fairbanks and North Pole, which requires further investigation, including exploration of recirculation**.

**3. Overprediction of surface $SO_2$: I understand that the authors alluded to vertical mixing being a main factor affecting the model simulation of surface $SO_2$ rather than chemical lifetime (oxidation) in the wintertime at Fairbanks. It might still be interesting to compare modelled $SO_2$ with observed $SO_2$+sulfate (in terms of total sulfur; I believe that sulfate was measured at CTC site as indicated in Simpson et al., 2024). Since the model does not consider sulfur oxidation, this can perhaps provide a check on whether modelled and observed total sulfur is comparable. If they are, this can then lead to questions such as whether oxidation pathways, other than the usual suspects of OH (in clear air) and $H_2O_2$, $O_3$ (in aqueous phase), may be at play or whether it could be an emission issue, e.g., a larger portion of sulfur (from space heating, for example) could be emitted as sulfate rather than $SO_2$ perhaps (with reference to Moon et al., ACS EST Air 2024, 1, 139-149). Just a thought.**

This is a nice suggestion, and we have explored this idea. The plot below shows modelled $SO_2$ ($\mu g/m^3$) against observed $SO_2$+$SO_4^{2-}$ ($\mu g/m^3$) at CTC. Observed $SO_4^{2-}$ is provided by Brice Temime-Roussel and Barbara D'Anna (LCE group) (co-authors). Hourly sulphate concentrations reach up to 15 $\mu g/m^3$ in the cold polluted period (end of A-C and T-C). However, the observed campaign average is only 1.7 $\mu g/m^3$, which is much lower than average $SO_2$ (28

µg/m$^3$). This supports our conclusion that vertical mixing in FLEXPART-WRF may be important for the overestimation of SO$_2$, notably during less stable periods. In addition, Moon et al. (2023) suggested that 60 % of SO$_4^{2-}$ is from primary emissions. We had already taken into account SO$_2$ oxidation using the sulphur oxidation ratio (SOR) from Moon et al. (2023). This lowers SO$_2$ when oxidation of SO$_2$ is increased, generally in February or during high O$_3$ weakly stable conditions. See section 6.2.3 for details. Since the sulphur chemistry is complex, a caveat has been added to the end of Section 6.2.4 to acknowledge this comment in the vertical mixing sensitivity. This includes updates from reviewer #2, since this section has been reduced.

[Figure]

New Lines X-X: Overall, the results suggest improvements are needed to the treatment of vertical mixing in FLEXPART-WRF during wintertime Arctic conditions. **However, we note that SO$_2$ overestimates may also be influenced by additional chemical processing not accounted for in Section 6.2.3, or by underestimation of dry or wet deposition.**

**4. I don't quite see the logic behind the constant emission test being used to illustrate the sensitivity of modelled tracers to meteorology. What the test illustrates is perhaps the impact of emission dependency on meteorology (e.g., dependency of CO emissions from diesel trucks on temperature)? Maybe I misunderstood the CONST-EM run. Were the campaign averaged emissions (or emission rates) applied to surface emissions only or applied to stack emissions also? If applied to stack emissions, were plume-rise calculation done the same way as CTRL run or done differently?**

We agree that the text describing the setup of this simulation was not particularly clear. By running with constant CO emissions, any variability in emissions including temperature dependence, diurnal variability and weekday/weekend variability, are removed. There is no explicit chemistry in the model for CO, because there is essentially no CO photochemical loss due to winter nighttime conditions (very low or zero OH) leading to a long photochemical lifetime. For this reason, variations in CO in the run with constant emissions are only due to meteorology. We have added in a sentence to lines 248 to 249, to state that no lifetime or explicit chemistry is included in the model setup for CO. We've also updated lines 549-559. Constant CO emissions from power plants were also simulated, but since they are very low compared to surface CO emissions the impact was negligible at the surface.

Lines 248-249: The influence of meteorology and emission treatments are explored in Section 6, together with atmospheric lifetimes (Appendix E5).

Lines 248-249: The influence of meteorology and emission treatments are explored in Section 6, together with atmospheric lifetimes (Appendix E5). **There is no explicit chemistry or atmospheric lifetime for CO included in the model setup.**

Lines 549-559: In order to explore the influence of meteorological variability on simulated tracers at the surface, the model is run with constant emissions  Note that $NO_x$ and $SO_2$ with constant emissions were also simulated (not shown), and are more comparable to the CTRL simulation. Differences in diurnal cycles for CTRL and CONST-EM CO ~~are shown in Fig. 9b). CTRL shows better agreement compared to the   observations for the whole campaign, AC-C and TC. Simulated CO mixing ratios are enhanced during the daytime due to the diurnal variability in the emissions, which are dominated by the on-road sector Downtown (Fig. A1). These results suggest that modelled CO biases can be explained partly by emission variability and by differences in modelled and observed meteorology influencing tracer transport and mixing as well as ABL stability. Discrepancies are linked in part to the EPA-WRF simulation as discussed above and also to treatments in FLEXPART-WRF. The sensitivity of results to the mixing height parameter in the FLEXPART-WRF BL scheme is examined further in Section 6.2.4.~~

Lines 549-559: In order to explore the influence of meteorological variability on simulated tracers at the surface, the model is run with constant emissions **(run CONST-EM, see Figure 9a). For this run, hourly emissions are averaged over the full campaign, removing the diurnal and weekday/week-end variability, and the effects of temperature on the emissions. Results are examined for CO since, due to its long photochemical lifetime in winter, simulated CO is only dependent on meteorology, i.e. it has no chemical loss.** Note that $NO_x$ and $SO_2$ with constant emissions were also simulated (not shown), and are more comparable to the CTRL simulation. Differences in diurnal cycles for CTRL and CONST-EM CO are **also shown in Figure 9b. In general, CONST-EM CO shows the same variability over time as CTRL. For example, CONST-EM CO is higher during the stable polluted AC-C period compared to the less stable Mixed period (Figures 9a and 9b). CONST-EM results also exhibit some diurnal variability, albeit less than in CTRL, and compared to the observations. These results highlight that variations in meteorological conditions, including diurnal effects, are an important factor controlling pollutants at the surface. Differences between CONST-EM and CTRL also show the importance of diurnal variations in CO emissions during cold pollution episodes. Surface CO emissions are dominated by the on-road sector Downtown (see Figure A1). Nevertheless, the CONST-CO negative biases are more pronounced in cold polluted periods than the full campaign, showing the cold temperature dependence of CO gasoline emissions is important. In summary, CO biases can be explained partly by emission variability and by differences in modelled and observed meteorology influencing tracer transport and mixing, as well as ABL stability. Discrepancies due to meteorology are linked to the EPA-WRF simulation as discussed above, and also to treatments of vertical mixing and turbulence in FLEXPART-WRF. The sensitivity of results to the mixing height parameter in the FLEXPART-WRF BL scheme is examined further in Section 6.2.4.**

**Specific comments**

**Line 17: Arctic haze occurs primarily during late winter and spring (highest occurrence in April and May) according to Shaw et al. (1995) rather than winter and early spring.**

Thanks for your comment, we have corrected this sentence accordingly.

**Line 17:** Arctic haze, with enhanced aerosols and trace gases, is formed in the lower troposphere during **late winter and early springtime (Shaw, 1995),** and is predominantly caused by low-level transport of pollution, driven by low pressure weather systems, originating from Northern Eurasia (Stohl, 2006; Bourgeois and Bey, 2011; Law et al., 2014).

**Line 158-160: The problem here is that the buoyancy force is only evaluated at the stack height while the atmospheric stability changes with height as in the case. It is particularly problematic under the strongly stratified situation or with elevated inversion layers. Akingunola et al. (2018) employed a methodology to evaluate the buoyancy force as the plume rises through the vertical column (model levels), which is more computational involved than the capping strategy used here.**

We thank the reviewer for this suggestion. This relates to the general comment regarding the use of WRF variables in treatment of power plant plume rise. We did consider this approach but several reasons justified using an approach based on observations for this study. It would also require additional validation of the WRF runs at the stack locations, but unfortunately meteorological data is not available at these locations. However, we have added the reviewer's suggestion to use a more elaborate approach using WRF in future studies in the Conclusions/future perspectives section..

Lines 696-699: The plume rise calculations could be improved further by using WRF temperatures and winds at the location of the power plant stacks, rather than using radiosondes at Fairbanks airport, allowing spatial differences to be better captured. Acquisition of more vertical profile observations (e.g. using drones) at, and downwind of, the power plant stacks would also be valuable.

Lines 696-699: The plume rise calculations could be improved further by using WRF temperatures and winds at the location of the power plant stacks, rather than using radiosondes at Fairbanks airport, allowing spatial differences to be better captured. **The treatment of vertical plume rise could be further improved by taking into account the changes in the buoyancy force of the plume as it rises above the stack, for example, as in Akingunola et al. (2018).** Acquisition of more vertical profile observations (e.g. using drones) at, and downwind of, the power plant stacks would also be valuable.

**Line 168: I don't understand why this is done, by removing negative temperature gradient.**

This sentence was worded incorrectly as also pointed out by Reviewer #1. We remove profiles with negative temperature gradients, i.e. those without surface or elevated inversions. We have corrected the text:

Line 170: Negative temperature gradients throughout the profile are removed as no inversions are observed.

Line 170: **Profiles with negative temperature gradients, i.e. no surface or elevated inversions detected, are removed from the analysis.**

**Line 178: The use of +/- 8% of plume height for plume width/thickness – what is the basis for this?**

This threshold was chosen to constrain the upper and lower limit for the plume thickness/width in FLEXPART-WRF simulation and based on observed plumes. Observed plume widths were best captured using +/- 8%, following initial analysis using 5%, 8% and 10 %. However, it can be noted that the observations may sometimes not sample the full plume depth, but this is not taken into account here.

Lines 177-179: Emission tracers are released between +/-8 % of the calculated plume rise height, to represent the plume thickness. **This threshold was chosen based on the optimal thickness compared to observed plumes in test simulations. In the case of plume capping, this also accounts for a** small fraction of the emissions **penetrating** the temperature inversion.

**Line 247-251: Were there fog and precipitation events during the ALPACA campaign?**

Yes, fog occurred during the cold polluted period (Lill et al., 2024), and there was precipitation in February. We have added a sentence stating this to the text.

Lines 249-251: Dry and wet deposition are included in CTRL only for $SO_2$ (see Appendix E3 for more details) since these losses are not important for CO, and considered to be very small for $NO_x$ (Liu et al., 1987). Runs with and without dry and wet deposition of $SO_2$ only had a very small influence on the results (not shown).

Lines 249-251: Dry and wet deposition are included in CTRL only for $SO_2$ (see Appendix E3 for more details) since these losses are not important for CO, and considered to be very small for $NO_x$ (Liu et al., 1987). **A fog event occurred from 29 January to 3 February (Lill et al., 2024), and precipitation events in February.** Runs with and without dry and wet deposition of $SO_2$ only had a very small influence on the results (not shown).

**Line 273-277: Are SS and WS regimes determined for the lowest 100 m only?**

They are determined for the lowest 25 m only, i.e. based on stability strengths towards the surface. This is why we include panel (e) in Figure 3.

Lines 273-276: At the surface **(0-25 m)**, strongly stable (SS) and weakly stable (WS) meteorological regimes are diagnosed based on observed temperature gradients (dT/dZ per 100 m), calculated using the 12-hourly FAI radiosonde data.

**Figure 5: Are the results shown from the runs conducted with only surface or power plant emissions separately? Also, it may be clearer to denote SS and WS using the notation of filled (solid) and unfilled (open) circles.**

Yes, they are run separately, as there's no particle interaction. We combine the results together in most of the figures, but here they're shown separately here in Figure 5. The Figure 5 caption has been revised.

Figure 5. a) Modelled (CTRL) $SO_2$ tracer as a function of altitude (m) and local time (AKST, hours) for i) total power plant emissions and (ii) total surface emissions at a) Downtown and b) UAF Farm.

Figure 5. a) Modelled (CTRL) $SO_2$ tracer as a function of altitude (m) and local time (AKST, hours) for i) total power plant tracer and (ii) total surface tracer at a) Downtown and b) UAF Farm. **The WS and SS surface stability regimes are indicated every 12 hours by filled (solid), and unfilled (open) circles, respectively.**

**Line 368-370: How is wintertime Arctic haze defined here? Are you really considering 0.012 – 0.1 ug/m3 of $SO_2$ and < 0.01 ug/m3 sulfate aerosols as Arctic haze? These are much lower levels compared to the concentration levels (particularly in terms of sulfate) observed during the springtime Arctic haze events at High Arctic sites (e.g., Utqiagvik and Alert).**

This is a good point and was also commented on by Reviewer 1. We agree that the text reads as though Arctic Haze has very low concentrations of $SO_2$ and sulphate, which is not the case. In fact, there was an error in our conversion from $\mu g/m^3$ to ppb based on $SO_2$ data reported by Tran et al., 2011 (Figure 5 data). We apologise for this oversight. We have instead included updated examples of $SO_2$ concentrations in the Arctic (Skov et al., 2023), and of observed $SO_4^{2-}$ aerosol concentrations in wintertime Arctic haze (Ioannidis et al., 2023). This point essentially highlights the large influence of local emissions in the Fairbanks area.

Lines 368-370: This regional pollution could be contributing to wintertime Arctic haze which has appreciably smaller concentrations of trace gases and aerosols. For example,

Lines 368-370: This regional pollution could be contributing to wintertime Arctic haze **which has lower concentrations** of trace gases and aerosols. **For example, simulated $SO_2$ concentrations at Villum in north-east Greenland range between 0.1 and 2.2 $\mu g/m^3$ (approx. 0.1-0.9 ppb) in 2018 and 2019 winter months (Skov et al., 2023). Sulphate concentrations between 0.1 to 0.8 $\mu g/m^3$ at Alert, Zeppelin and Villum in January and February 2014 were reported in Ioannidis et al. (2023), while sulphate in downtown Fairbanks ranged between 1-5 $\mu g/m^3$ during ALPACA-2022 (Moon et al., 2023).**

**Line 380-383: The explanation of how the background CO and NOx concentrations are determined is not very clear.**

We have revised these sentences. Reviewer #1 also asked for clarification of this point..

Lines 380-386: In this analysis, CO and $NO_x$ pollution plumes are identified in each of the Helikite flights when elevated concentrations are observed above the 90th percentile in the data of the flight. To compare to the model results, that are enhancements above background, a polluted background is assigned in each flight using the modal concentration of the observed

concentration distribution, and subtracted from observed plume mixing ratios. The resulting δCO and δNOₓ enhancements are compared with the model results. In order to evaluate power plant plumes only, this comparison only uses observations above 30 m, away from the influence of surface emissions. Some profiles of CO on 30 January and 10 February are removed due to issues with the CO sensor when the power was switched off to replace batteries and switched back on during the flight.

Lines 380-386: **Since model results are representative of enhancements above background, to facilitate comparison with Helikite data, a polluted background is assigned to Helikite observations of CO and NOₓ to determine pollution plume enhancements (δCO and δNOₓ) equivalent to the simulated quantities. First, the pollution plumes are assigned using the 90th percentile of the distribution of concentrations observed during each flight. A polluted background is assigned using the modal concentration of each flight, and subtracted from the identified plume to give the observed pollution plume enhancement (δCO and δNOₓ). In order to evaluate power plant plumes only, this comparison only uses observations above 30 m, away from the influence of surface emissions. Some profiles of CO on 30 January and 10 February are removed due to issues with the CO sensor.**

**Figure 8(c): Are these supposed to be time series of modelled plume height or the vertical extend of the power plant plumes?**

Thanks for highlighting this. We have edited the caption for Figure 8c to ensure it is clear that we are referring to the altitude where 95th percentile of tracers reside.

Figure 8. a) Vertical cross section of total simulated (CTRL) power plant tracer over several hours before, during and after each flight, with observations included as scatter points (as in Fig. 6). b) Hourly % contributions from different power plant stack throughout the vertical profile. c)  For panels a) to c), Cases 1-3 are shown from left to right. See text for more details.

Figure 8. a) Vertical cross section of total simulated (CTRL) power plant tracer over several hours before, during and after each flight, with observations included as scatter points (as in Figure 6). b) Hourly % contributions from different power plant stacks throughout the vertical profile. c) **The altitude (m) where the 95ᵗʰ percentile of tracers reside, for each contributing power plant stacks, as a function of time (hourly).** For panels a) to c), cases 1-3 are shown from left to right. See text for more details.

**Line 423-426: The difference between the radiosonde wind observation at Fairbanks airport and the LiDAR wind observation close to Aurora power plant would not necessarily affect the modelled dispersion of the Aurora power plant plume. The issue is whether the WRF simulated wind at the Aurora location is different from the LiDAR (close to Aurora) wind observation, since the dispersion calculation depends on the WRF simulated wind not the radiosonde observation at Fairbanks airport (I assume). Of course, the difference in temperature profiles between the airport and the power plant location could affect the plume injection height in this case. If there is a significant**

**vertical wind shear at the power plant location, that would impact the modelled dispersion if the plume were misplaced in the model.**

Thank you, we agree this is a good point. The WRF wind speeds are around 2 m/s around 50 m altitude at both downtown and the UAF Farm. However, we used radiosonde data for this specific figure because the plume rise parameterisation uses wind speeds at the approximate stack height (Equation 2). Since observed radiosonde wind speeds are much lower (1 m/s) than the lidar observations at the CTC site (4 m/s), this case potentially points to a deficiency in the use of the radiosonde data for the plume rise treatment. Nevertheless, as we have stated above, we use radiosonde profiles to give an observational constraint on the plume rise calculation. We have altered the text to clarify.

Lines 422-426: Therefore, the calculated emission injection height for Aurora until 0900 AKST is 150 m (midpoint) and is the same in CTRL and NO-CAP. Also, at 0300 AKST on 30 January (time of radiosonde), the observed LiDAR wind speeds at CTC (900 m south-east of Aurora) were up to 4 m/s at the Aurora stack height (48 m), while the radiosonde wind speeds were lower than 1m/s (> 5 km south-west of Aurora).

Lines 422-426: Therefore, the calculated emission injection height for Aurora until 0900 AKST is 150 m (midpoint) in both the CTRL and NO-CAP runs. Also, at 0300 AKST on 30 January (time of radiosonde), the observed LiDAR wind speeds at CTC (900 m south-east of Aurora) were up to 4 m/s at the Aurora stack height (48 m), while the radiosonde wind speeds were lower than 1 m/s (> 5 km south-west of Aurora). **Since radiosonde wind speeds are used to calculate plume rise, this suggests that the simulated altitude of the Aurora plume may be underestimated due to a lack of observed spatial coverage in the parameterisation.**

**Line 433: I can't see any agreement between the nighttime radio sounding and the Helikite profiling below 270 m from Fig. 7a.**

Thank you for this comment. You are correct that there isn't agreement below the EI. This is to do with the local drainage flow that influences the UAF Farm site, reducing both static and dynamic surface stability in this case. We didn't comment on this, but have now added in a comment to explain this.

Line 433- 434: In this case, the radiosonde-derived EI agrees with the observed Helikite EI and, even if the stratification is rather weak, a layer of trapped emissions, with observed CO and $NO_x$ enhancements, is evident.

Line 433- 434: In this case, the radiosonde-derived EI agrees with the observed Helikite EI and, even if the stratification is rather weak, a layer of trapped emissions, with observed CO and $NO_x$ enhancements, is evident. **However, below 270 m, the radiosonde temperature profile shows an SBI, in disagreement with the Helikite profile which has a negative temperature gradient, likely due to influence from the drainage flow at the UAF Farm, see Appendix B, and Fochesatto et al. 2024.**

**Line 436-437: The statement on the CTRL (with capping) being better than NO-CAP in this case is somewhat subjective. It could be argued that NO-CAP was a bit better than**

**CTRL in terms of plume centerline (or hceight of maximum concentration) and vertical distribution in Fig.7b (case 2).**

Thank you for this comment. The text has been revised to specify how CTRL improves the comparison to observations compared to NO-CAP.

Lines 436-437: This plume is attributed to UAF C. In this case, EI capping is applied and improves the modelled plume altitude compared to NO-CAP.

Lines 436-437: This plume is attributed to UAF C. In this case, EI capping is applied and improves the modelled plume altitude compared to NO-CAP **in terms of the plume altitude.**

**Line 444: what about the $NO_x$ plume at ~45 m?**

A sentence has been added about this. Note that this plume enhancement is small compared to the elevated plumes.

Lines 443-444: An elevated plume is observed between 85-120 m in $NO_x$ and 120-160 m in CO.

Lines 443-444: **A plume with relatively small $NO_x$ enhancements (< 5 ppb mean $\delta NO_x$) is observed at approximately 50 m, and an elevated plume is observed with increased enhancements in $NO_x$ between 85-120 m (5-10 ppb mean $\delta NO_x$) and 120-160 m in CO (25-30 ppb).**

**Line 451&455: "Fig.D3a" should be Fig.D2a?**

Thank you for noticing this typo, it has been changed accordingly. Note that line 451 should be Figure D2 panel a (CO) and lines 454-455 refer to Figure D2 panel b ($NO_x$).

Line 451: The UAF C stack contributes to $\delta$CO directly at the UAF farm as shown in **Figure D2a.**

Lines 454-455: This results in stronger transport to the south, displacing the simulated plumes slightly south of the UAF Farm **(Figure D2b)**, most likely explaining the underestimated **modelled $\delta NO_x$.**

**Line 459-462: Could perhaps compare the LiDAR aerosol plume with modelled SO2 plumes (given that the aerosols may be dominated by sulfate)?**

We agree that this is an important point to make, and have included a point about this in the text.

Lines 459-462: Appendix Figure D3 shows doppler wind LiDAR observations for cases 1 (CTC) and 2 (UAF Farm). In each case, plumes are identified by the wind LiDAR at a comparable altitude to the identified plumes at the Farm. Although the wind LiDAR is sensitive to aerosols, and not trace gases, the results suggest that power plants are also a source of aerosols over Fairbanks (more details in Appendix D2).

Lines 459-462: Appendix Figure D3 shows doppler wind LiDAR observations for cases 1 (CTC) and 2 (UAF Farm). In each case, plumes are identified by the wind LiDAR at a comparable altitude to the identified plumes at the Farm. Although the wind LiDAR is sensitive to aerosols, and not trace gases, **it is possible that sulphate aerosols are contributing to observed aerosols. The** results suggest that power plants are also a source of aerosols over Fairbanks (more details in Appendix D2).

**Line 599: Fig.10 shows that NO$_x$ is modestly overestimated in this temperature range (-13C to -23C) without the inclusion of the adjustment for cold temperature.**

We apologise that the explanation of this figure was confusing and misinterpreted by all 3 reviewers. This figure is not showing simulated mixing ratios, but estimated mixing ratios over downtown Fairbanks based on EPA emission fluxes that are also used as input to the model simulations, using an assumption about a mixed layer depth at the surface. More information has been added to lines 580-582 to make this more clear. The figure caption has also been revised. The changes also take into account the comments from Reviewers 1 and 2.

During the cold period estimated mixing ratios slightly overestimate CO compared to the observations (+23 %), whereas NO$_x$ is underestimated (-43 %) (Figure 10b). Since traffic is the main source of both CO and NO$_x$, this suggests that the vehicle NO$_x$ emissions are underestimated in the inventory in the coldest conditions. Adding a temperature enhancement for NO$_x$ brings the inventory and observations closer as shown in the right panel of Figure 10b. This method does not take into account meteorology or mixing leading to overestimated surface mixing ratios, in particular, during the Mixed period, when temperatures between -13C to -23C were common, and vertical exchange was more prevalent. In the T-W period, estimated mixing ratios of both CO and NO$_x$ are underestimated, indicating the cold temperature is not controlling the underestimate during this event.

Lines 580-582: The possible contribution of temperature dependent diesel emissions to CO and NO$_x$ in Fairbanks is investigated first by summing the EPA emissions for each sector over the volume of the box covering 4 grid cells in Downtown and up to 10 m to estimate hourly concentrations that are compared to CTC observations.

Lines 580-582: The possible contribution of temperature dependent diesel emissions to CO and NO$_x$ **concentrations in Fairbanks is investigated based on the EPA surface emission fluxes that are used in the model simulations. In order to compare to surface observations at CTC, surface fluxes for each emission sector, in kg/m$^2$/s, are converted into hourly mixing ratios (in ppb) by taking into account the volume of each emission grid cell (1.3 km$^2$ x 10m (AGL) in vertical). These estimates are averaged over the 4 grid cells covering the Downtown area and shown in Figure 10.**

Lines 584-589: The observations show a clear increase in NO$_x$ at colder temperatures, especially below -23°C, but is much less distinct for observed CO. For CO, as noted earlier, a cold temperature dependence is already included for mobile (on-road and non-road) gasoline emissions in MOVES3, and there is good agreement between the CO observations and estimated mixing ratios during the cold polluted period (Fig. 10b). The poor agreement between NO$_x$ observations and estimated NO$_x$ supports the hypothesis that an increase in diesel NO$_x$ vehicle emissions due to a cold temperature dependence may be required**.** Estimated mixing ratios are underestimated in the warm polluted period compared to

observations for both CO and NO$_x$ (Fig. 10c), indicating that a cold-temperature effect is not driving the discrepancy in this period.

Lines 584-589: **At intermediate temperatures (-13°C to -23°C), common during the Mixed period, estimated NO$_x$ mixing ratios are overestimated compared to observations. This is in part because meteorology and mixing are not considered, as also shown for CO. However, t**he observations show a clear increase in NO$_x$ at colder temperatures, especially below -23°C, which is much less distinct for observed CO. For CO, as noted earlier, a cold temperature dependence is already included for mobile (on-road and non-road) gasoline emissions in MOVES3, **and there is better agreement between the CO observations and estimated mixing ratios during the cold polluted period (CPP) (Figure 10b**). The poor agreement between NO$_x$ observations and estimated NO$_x$ **during the CPP** supports the hypothesis that an increase in diesel NO$_x$ vehicle emissions due to a cold temperature dependence may be required. **Furthermore, estimated CO and NO$_x$ mixing ratios are both underestimated during T-W (Figure 10c),** indicating that a cold-temperature effect is not driving the discrepancy in this period.

Figure 10 caption: EPA surface emission mixing ratios calculated over the Downtown area for contributing emission sectors (grey lines) compared to observations at the CTC site (black lines) in ppb averaged over 3 °C temperature bins for (a) full campaign, b) 29 January to 2 February and c) 23 to 25 February, for CO (left) and NO$_x$ (right). Different emission sectors coloured are shown and given in the legend. The NO$_x$ increment from the linear temperature dependence is shown with '..' hatching. The mid-point of the 3 °C temperature bin is shown on the x-axis. The bold grey lines correspond to the total surface-emitted emissions listed in the legend with the NO$_x$ vehicle temperature increment. See text for details.

**Figure 10: Estimated mixing ratios in ppb, using surface EPA emission fluxes, averaged over the Downtown area, compared to observed mixing ratios in ppb at the CTC site (black circles), averaged over 3 °C temperature bins for a) full campaign, b) 29 January to 2 February, and c) 23 to 25 February, for CO (left) and NO$_x$ (right). The shaded colours correspond to the contributing emission sectors indicated in the legend (total of all sectors = grey diamonds), The mid-point of the 3 °C temperature bin is shown on the x-axis. The increment in NO$_x$ vehicle emissions at low temperatures, according to the linear temperature dependence is also shown with cyan shading and '..' hatching. See text for details.**

**Line 605-606: Why should the cold temperature effect on NO$_x$ emission from diesel vehicles be limited to stably stratified conditions only? In another word, why would the vehicle emissions be dependent on atmospheric stability?**

We agree that the cold temperature effect is important during cold conditions, and not necessarily under stable conditions. We have corrected the sentence below.

Line 605-606: The results suggest an increase in NO$_x$ emissions from diesel vehicles is needed during stable periods with very cold temperatures, notably below -20 °C.

**Line 605-606: The results suggest an increase in NO$_x$ emissions from diesel vehicles is needed during periods with very cold temperatures, notably below -20 °C.**

**Line 662-668 and Line 727-731: The rather complicated sensitivity to the vertical mixing (via the hmin setting), i.e., conflicting results amongst different pollutants (with enhanced or suppressed vertical mixing), may also be an indication of compensating errors in the system.**

We agree that this could also be an explanation. We have added a sentence at the end of Section 6.2.4 (which includes updates from reviewers #1 and #2.

Lines 667-668: Overall, this suggests that improvements are needed to the treatment of vertical mixing in FLEXPART-WRF during wintertime Arctic conditions.

Lines 667-668: However, we note that $SO_2$ overestimates may also be influenced by additional chemical processing not accounted for in Section 6.2.3, or by underestimation of dry or wet deposition. **Variable results among pollutants could also indicate compensating errors in the model.**